# BaB-ND: Long-Horizon Motion Planning with Branch-and-Bound and Neural Dynamics

**Keyi Shen**[1*]**, Jiangwei Yu**[1*]**, Jose Barreiros**[2]**, Huan Zhang**[1]**, Yunzhu Li**[3]
[1]University of Illinois Urbana-Champaign    [2]Toyota Research Institute    [3]Columbia University
`{keyis2, jy79}@illinois.edu, jose.barreiros@tri.global,`
`huan@huan-zhang.com, yunzhu.li@columbia.edu`

## Abstract

Neural-network-based dynamics models learned from observational data have shown strong predictive capabilities for scene dynamics in robotic manipulation tasks. However, their inherent non-linearity presents significant challenges for effective planning. Current planning methods, often dependent on extensive sampling or local gradient descent, struggle with long-horizon motion planning tasks involving complex contact events. In this paper, we present a GPU-accelerated branch-and-bound (BaB) framework for motion planning in manipulation tasks that require trajectory optimization over neural dynamics models. Our approach employs a specialized branching heuristics to divide the search space into subdomains, and applies a modified bound propagation method, inspired by the state-of-the-art neural network verifier $\alpha,\beta$-CROWN, to efficiently estimate objective bounds within these subdomains. The branching process guides planning effectively, while the bounding process strategically reduces the search space. Our framework achieves superior planning performance, generating high-quality state-action trajectories and surpassing existing methods in challenging, contact-rich manipulation tasks such as non-prehensile planar pushing with obstacles, object sorting, and rope routing in both simulated and real-world settings. Furthermore, our framework supports various neural network architectures, ranging from simple multilayer perceptrons to advanced graph neural dynamics models, and scales efficiently with different model sizes. Project page: https://robopil.github.io/bab-nd/ .

## 1 Introduction

Learning-based predictive models using neural networks reduce the need for full-state estimation and have proven effective across a variety of robotics-related planning tasks in both simulations (Li et al., 2018; Hafner et al., 2019c; Schrittwieser et al., 2020; Seo et al., 2023) and real-world settings (Lenz et al., 2015; Finn & Levine, 2017; Tian et al., 2019; Lee et al., 2020; Manuelli et al., 2020; Nagabandi et al., 2020; Lin et al., 2021; Huang et al., 2022; Driess et al., 2023; Wu et al., 2023; Shi et al., 2023). While neural dynamics models can effectively predict scene evolution under varying initial conditions and input actions, their inherent non-linearity presents challenges for traditional model-based planning algorithms, particularly in long-horizon scenarios.

To address these challenges, the community has developed a range of approaches. Sampling-based methods such as the Cross-Entropy Method (CEM) (Rubinstein & Kroese, 2013) and Model Predictive Path Integral (MPPI) (Williams et al., 2017) have gained popularity in manipulation tasks (Lowrey et al., 2018; Manuelli et al., 2020; Nagabandi et al., 2020; Wang et al., 2023) due to their flexibility, compatibility with neural dynamics models, and strong GPU support. However, their performance in more complex, higher-dimensional planning problems is limited and requires further theoretical analysis (Yi et al., 2024). Alternatively, more principled optimization approaches, such as Mixed-Integer Programming (MIP), have been applied to planning problems using sparsified neural dynamics models with ReLU activations (Liu et al., 2023b). Despite achieving global optimality and better closed-loop control performance, MIP is inefficient and struggles to scale to large neural networks, limiting its ability to handle larger-scale planning problems.

In this work, we introduce a branch-and-bound (BaB) based framework that achieves stronger performance on complex planning problems than sampling-based methods, while also scaling to

---

*Equal contribution.

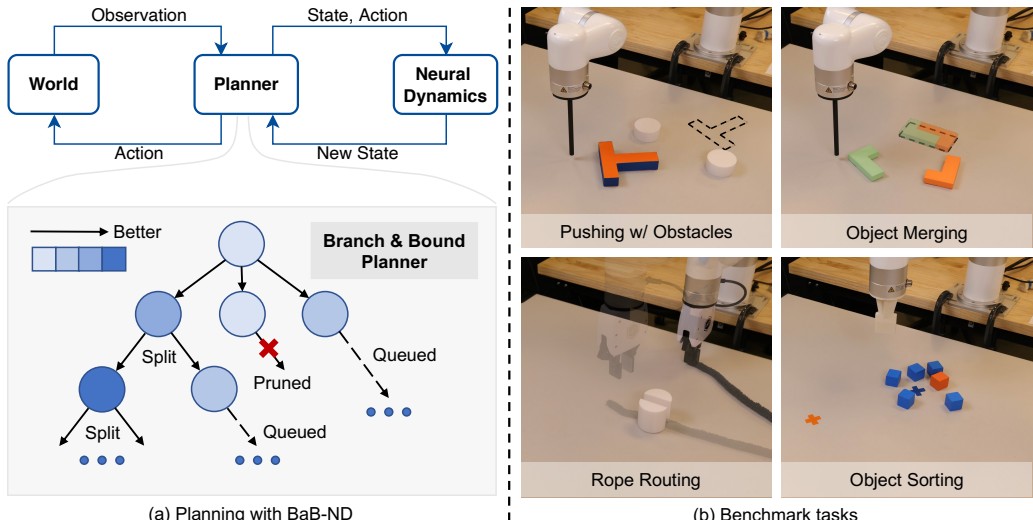

Figure 1: **Framework overview.** (a) Our framework takes scene observations and applies a branch-and-bound (BaB) method to generate robot trajectories using the neural dynamics model (ND). The BaB-ND planner constructs a search tree by branching the problem into sub-domains and then systematically searching only in promising sub-domains by evaluating nodes with a bounding procedure. (b) BaB-ND demonstrates superior long-horizon planning performance compared to existing sampling-based methods and achieves better closed-loop control in the real-world scenarios. We evaluate our framework on various complex planning tasks, including non-prehensile planar pushing with obstacles, object merging, rope routing, and object sorting.

large neural dynamics models that are intractable for MIP-based approaches. Our framework is inspired by the success of BaB in neural network verification (Bunel et al., 2018; 2020b; Palma et al., 2021), which tackles challenging optimization objectives involving neural networks. State-of-the-art neural network verifiers such as $\alpha,\beta$-CROWN (Xu et al., 2021; Wang et al., 2021; Zhang et al., 2022a), utilize BaB alongside bound propagation methods (Zhang et al., 2018; Salman et al., 2019), demonstrating impressive strength and scalability in verification tasks, far surpassing MIP-based approaches (Tjeng et al., 2019; Anderson et al., 2020). However, unlike neural network verification, which only requires finding a lower bound of the objective, model-based planning demands high-quality feasible solutions (i.e., planned state-action trajectories). Thus, significant adaptation and specialization are necessary for BaB-based approaches to effectively solve planning problems.

Our framework, BaB-ND (Figure 1.a), divides the action space into smaller subdomains through novel branching heuristics (*branching*), estimates objective bounds using a modified bound propagation procedure to prune subdomains that cannot yield better solutions (*bounding*), and focuses searching on the most promising subdomains (*searching*). We evaluate our approach on contact-rich manipulation tasks that require long-horizon planning with non-smooth objectives, non-convex feasible regions (with obstacles), long action sequences, and diverse neural dynamics model architectures (Figure 1.b). Our results demonstrate that BaB-ND consistently outperforms existing sampling-based methods by systematically and strategically exploring the action space, while also being significantly more efficient and scalable than MIP-based approaches by leveraging the inherent structure of neural networks and GPU support.

We make three key contributions: (1) We propose a general, widely applicable BaB-based framework for effective long-horizon motion planning over neural dynamics models. (2) Our framework introduces novel branching, bounding, and searching procedures, inspired by neural network verification algorithms but specifically adapted for planning over neural dynamics models. (3) We demonstrate the effectiveness, applicability, and scalability of our framework across a range of complex planning problems, including contact-rich manipulation tasks, the handling of deformable objects, and object piles, using diverse model architectures such as multilayer perceptrons and graph neural networks.

## 2 RELATED WORKS

**Neural dynamics model learning in manipulation.** Dynamics models, learned from observations in simulation or the real world using deep neural networks (DNNs), have been widely and successfully

applied to robotic manipulation tasks (Shi et al., 2023; Wang et al., 2023). Neural dynamics models can be learned directly from pixel space (Finn et al., 2016; Ebert et al., 2017; 2018; Yen-Chen et al., 2020; Suh & Tedrake, 2020) or low-dimensional latent space (Watter et al., 2015; Agrawal et al., 2016; Hafner et al., 2019b;a; Schrittwieser et al., 2020; Wu et al., 2023). Other approaches use more structured scene representations, such as keypoints (Kulkarni et al., 2019; Manuelli et al., 2020; Li et al., 2020), particles (Li et al., 2018; Shi et al., 2022; Zhang et al., 2024), and meshes (Huang et al., 2022). Our work employs keypoint or object-centric representations, and the proposed BaB-ND framework is compatible with various architectures, ranging from multilayer perceptrons (MLPs) to graph neural networks (GNNs) (Battaglia et al., 2016; Li et al., 2019).

**Model-based planning with neural dynamics models.** The highly non-linear and non-convex nature of neural dynamics models hinders the effective optimization of model-based planning problems. Previous works (Yen-Chen et al., 2020; Ebert et al., 2017; Nagabandi et al., 2020; Finn & Levine, 2017; Manuelli et al., 2020; Sacks et al., 2023; Han et al., 2024) utilize sampling-based algorithms like CEM (Rubinstein & Kroese, 2013) and MPPI (Williams et al., 2017) for online planning. Despite their flexibility and ability to leverage GPU support, these methods struggle with large input dimensions due to the exponential growth in the number of required samples. Previous work (Yin et al., 2022) improved MPPI by introducing dynamics model linearization and covariance control techniques, but their effectiveness when applied to neural dynamics models remains unclear. Other approaches (Li et al., 2018; 2019) have used gradient descent to optimize action sequences but encounter challenges with the local optima and non-smooth objective landscapes. Recently, methods inspired by neural network verification have been developed to achieve safe control and robust planning over systems involving neural networks (Wei & Liu, 2022; Liu et al., 2023b; Hu et al., 2024a; Wu et al., 2024; Hu et al., 2024b), but their scalability to more complex real-world manipulation tasks is still uncertain. Moreover, researchers are also exploring the promising direction of performing planning over graphs of convex sets (GCSs) for contact-rich manipulation tasks Marcucci (2024); Graesdal et al. (2024). However, these approaches do not incorporate neural networks.

**Neural network verification.** Neural network verification ensures the reliability and safety of neural networks (NNs) by formally proving their output properties. This process can be formulated as finding the *lower bound* of a minimization problem involving NNs, with early verifiers utilizing MIP (Tjeng et al., 2019) or linear programming (LP) (Bunel et al., 2018; Lu & Kumar., 2020). These approaches suffer from scalability issues (Salman et al., 2019; Zhang et al., 2022b; Liu et al., 2021) because they have limited parallelization capabilities and fail to fully exploit GPU resources. On the other hand, bound propagation methods such as CROWN (Zhang et al., 2018) can efficiently propagate bounds on NNs (Eric Wong, 2018; Singh et al., 2019; Wang et al., 2018; Gowal et al., 2019) in a layer-by-layer manner, with the ability to be accelerated on GPUs. Combining bound propagation with BaB leads to successful approaches in NN verification (Bunel et al., 2020a; De Palma et al., 2021; Kouvaros & Lomuscio, 2021; Ferrari et al., 2022), and notably, the $\alpha,\beta$-CROWN framework (Xu et al., 2021; Wang et al., 2021; Zhang et al., 2022a) achieved strong verification performance on large NNs (Bak et al., 2021; Müller et al., 2022). In our model-based planning setting, we utilize the lower bounds from verification, with modifications and specializations, to guide our systematic search procedure to find high-quality feasible solutions.

## 3 BRANCH-AND-BOUND FOR PLANNING WITH NEURAL DYNAMICS MODELS

**Formulation.** We formulate the planning problem as an optimization problem in Eq. 1, where $c$ is the cost function, $t_0$ is the current time step, and $H$ is the planning horizon. $\hat{x}_t$ is the (predicted) state at time step $t$, and the current state $\hat{x}_{t_0} = x_{t_0}$ is known. $u_t \in \{u \mid \underline{u} \le u \le \overline{u}\} \subset \mathbb{R}^k$ is the robot's action at each step. $f_{\text{dyn}}$ is the pre-trained neural dynamics model (Please refer to Section D.3 for details about learning the neural dynamics model.), which takes state and action at time $t$ and predicts the next state $\hat{x}_{t+1}$. **The goal of the planning problem** is to find a sequence of optimal actions $u_t$ that minimize the sum of step costs:

$$\min_{\{u_t \in \mathcal{U}\}} \sum_{t=t_0}^{t_0+H} c(\hat{x}_t, u_t) \quad \text{s.t.} \quad \hat{x}_{t+1} = f_{\text{dyn}}(\hat{x}_t, u_t) \qquad \Longrightarrow \qquad \min_{\boldsymbol{u} \in \mathcal{C}} f(\boldsymbol{u}). \tag{1}$$

This problem can be challenging due to its long planning horizon $H$, complex cost function $c$, and the non-linear neural dynamics model $f_{\text{dyn}}$ with recursive dependencies at every step. Existing sampling-based and gradient-based methods may converge to sub-optima without systematic searching, while MIP-based methods fail to scale up with the size of $f_{\text{dyn}}$ and the planning horizon $H$.

To simplify notation, we can substitute all constraints on $\hat{x}_{t+1}$ into the objective recursively, and further simplify the problem as a constrained optimization problem $\min_{\boldsymbol{u} \in \mathcal{C}} f(\boldsymbol{u})$ (Eq. 1). Here $f$ is our final objective, a scalar function that absorbs the neural network $f_{\text{dyn}}$ and the cost function summed in all $H$ steps. $\boldsymbol{u} = \{u_{t_0:t_0+H}\} \in \mathcal{C}$ is the action sequence and $\mathcal{C} \subset \mathbb{R}^d$ is the entire input space with dimension $d = kH$. We also flatten $\boldsymbol{u}$ as a vector containing actions for all time steps, and use $\boldsymbol{u}_j$ to denote a specific dimension. Our goal is to then find the optimal objective value $f^*$ and its corresponding optimal action sequence $\boldsymbol{u}^*$.

**Branch-and-bound on a 1D toy example.** Our work proposes to solve the planning problem Eq. 1 using branch-and-bound. Before diving into technical details, we first provide a toy case of a non-convex objective function $f(\boldsymbol{u})$ in 1D space ($k = H = 1, \mathcal{C} = [-1, 1]$) and illustrate how to use branch-and-bound to find $f^*$.

In Figure 2.1, we visualize the landscape of $f(\boldsymbol{u})$ with its optimal value $f^*$. Initially, we don't know $f^*$ but we can sample the function at a few different locations (orange points). Although sampling (*searching*) often fails to discover the optimal $f^*$ over $\mathcal{C} = [-1, 1]$, it gives an upper bound of $f^*$, since any orange point has an objective greater than or equal to $f^*$. We denote $\overline{f}^*$ as the current best upper bound (orange dotted line).

In Figure 2.2, we split $\mathcal{C}$ into two subdomains $\mathcal{C}_1$ and $\mathcal{C}_2$ (*branching*) and then estimate the lower bounds of the objective in $\mathcal{C}_1$ and $\mathcal{C}_2$ using linear functions (*bounding*). The key insight is if the lower bound in one subdomain is larger than $\overline{f}^*$, then sampling from that subdomain will not yield any better objective than $\overline{f}^*$, and we may discard that subdomain to reduce the search space. In the example, $\mathcal{C}_1$ is discarded in Figure 2.3.

Then, in Figure 2.4, we only perform sampling in $\mathcal{C}_2$ with the same number of samples. *Searching* in the reduced space is likely to obtain a better objective and therefore $\overline{f}^*$ can be improved.

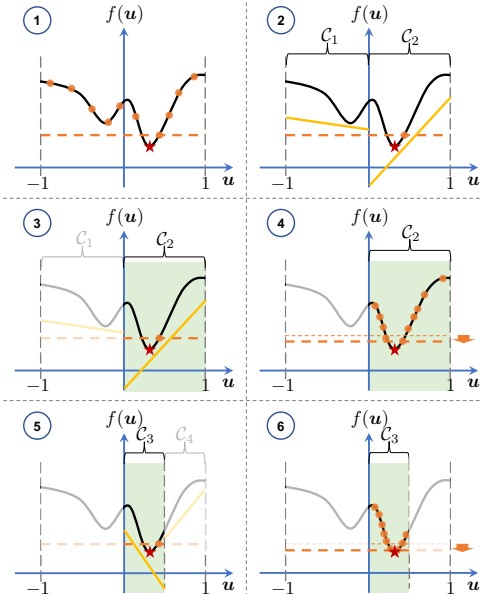

Figure 2: **Seeking $f^*$ with Branch-and-Bound.** ① Sample on input space $\mathcal{C}$. •: sampled points. ★: the optimal value $f^*$. – –: the current best upper bound of $f^*$ from sampling. ② Branch $\mathcal{C}$ into $\mathcal{C}_1$ and $\mathcal{C}_2$. —: the linear lower bounds of $f^*$ in subdomains. ③ Discard $\mathcal{C}_1$ since its lower bound is larger than $\overline{f}^*$. ▨: the remaining subdomain to be searched. ④ Search on only $\mathcal{C}_2$ and upper bound of $f^*$ is improved. - - - -: the previous upper bound. ⑤ Continue to branch $\mathcal{C}_2$ and bound on $\mathcal{C}_3$ and $\mathcal{C}_4$. ⑥ Search on $\mathcal{C}_3$. The upper bound approaches $f^*$.

We could repeat these procedures (*branching*, *bounding*, and *searching*) to reduce the sampling space and improve $\overline{f}^*$ as in Figure 2.5 and Figure 2.6. Finally, the value of $\overline{f}^*$ will converge to $f^*$. This branch-and-bound method systematically partitions the input space and iteratively improves the objective. In practice, heuristics for branching, along with methods for bound estimation and solution search, are critical to the performance of branch and bound.

**Methodology overview.** We now discuss how to use the branch-and-bound (BaB) method to find high-quality actions for the planning problem formulated as $\min_{\boldsymbol{u} \in \mathcal{C}} f(\boldsymbol{u})$ (Eq. 1). We define a *subproblem* $\min_{\boldsymbol{u} \in \mathcal{C}_i} f(\boldsymbol{u})$ as the task of minimizing $f(\boldsymbol{u})$ within a *subdomain* $\mathcal{C}_i$, where $\mathcal{C}_i \subseteq \mathcal{C}$. Our algorithm, BaB-ND, involves three components: *branching* (Figure 3.b, Section 3.1), *bounding* (Figure 3.c, Section 3.2), and *searching* (Figure 3.d, Section 3.3).

- *Branching* generates a partition $\{\mathcal{C}_i\}$ of some input space $\mathcal{C}$ such that $\bigcup_i \mathcal{C}_i = \mathcal{C}$, enabling systematic exploration of the input space.

- *Bounding* estimates the lower bound of $f(\boldsymbol{u})$ on each subdomain $\mathcal{C}_i$ (denoted as $\underline{f}^*_{\mathcal{C}_i}$). The lower bound can be used to prune unpromising domains and also guide the search for promising domains.

- *Searching* seeks good feasible solutions and outputs the best objective $\overline{f}^*_{\mathcal{C}_i}$ within each subdomain $\mathcal{C}_i$. $\overline{f}^*_{\mathcal{C}_i}$ is an upper bound of $f^*$, as any feasible solution provides an upper bound for the optimal minimization objective $f^*$.

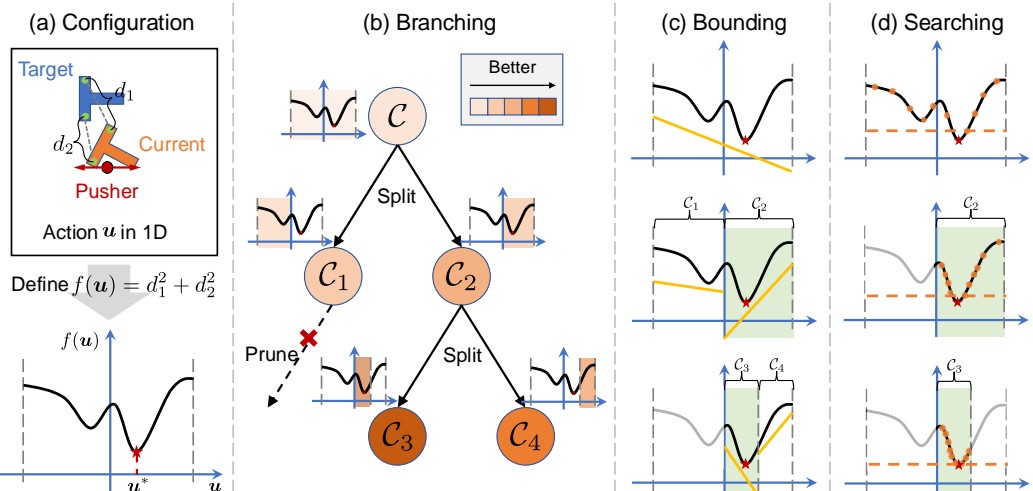

Figure 3: **Illustration of the branch-and-bound process.** (a) Configuration: we visualize a simplified case of pushing an object toward the target using a 1D action $\boldsymbol{u}$. We select two keypoints on the object and target and denote the distances as $d_1$ and $d_2$. Then we define our objective function $f(\boldsymbol{u})$ and seek $\boldsymbol{u}^*$ to minimize it. (b) **Branching:** we iteratively construct the search tree by splitting, queuing, and pruning nodes (subdomains). In every iteration, only the most promising nodes are prioritized for splitting, cooperating with bounding and searching. (c) **Bounding:** In every subdomain $\mathcal{C}_i$, we obtain the linear lower bound of $f^*$ ($\underline{f}^*$) via bound propagation. (d) **Searching:** we search better solutions with smaller objective ($\overline{f}^*$) on selected subdomains. indicates the most promising subdomain in every iteration. The search space progressively shrinks within the original input domain $\mathcal{C}$ with better solutions found and more subdomains pruned. A detailed illustration of our BaB-ND in a simplified robotic manipulation task is provided in Section A.1.

We can always prune subdomain $\mathcal{C}_j$ if its $\underline{f}^*_{\mathcal{C}_j} > \overline{f}^*$, where $\overline{f}^* := \min_i \overline{f}^*_{\mathcal{C}_i}$ is the best upper bound among all subdomains $\{\mathcal{C}_i\}$, since, in $\mathcal{C}_j$, there is no solution better than the current best objective $\overline{f}^*$. The above procedure can be repeated many times, and each time during *branching*, some previously produced subdomains $\mathcal{C}_i$ can be picked for *bounding*, and *searching*, while the remaining subdomains are stored in a set $\mathbb{P}$ for future *branching*. Our main algorithm is shown in Algorithm 1.

**Distinctions from neural network verification.** Although the generic BaB framework has been applied in neural network verification (Bunel et al., 2018; Wang et al., 2021), its goal is to *prove a sound lower bound* of $f(\boldsymbol{u})$ within $\mathcal{C}$, ensuring that $\underline{f}^*_{\mathcal{C}} \leq f^*$. In contrast, our BaB-ND aims to find a *concrete solution* $\tilde{\boldsymbol{u}}$—a near-optimal action sequence—for the objective-minimization problem $\min_{\boldsymbol{u} \in \mathcal{C}} f(\boldsymbol{u})$. These fundamental differences in objectives lead to distinct design choices.

We propose new *branching* heuristics that effectively guide the search for better solutions, extensively adapt the existing *bounding* algorithm CROWN (Zhang et al., 2018) to tackle its ineffectiveness and inefficiency issues under our complex planning setting and integrate a new *searching* component to find high-quality action sequences.

### 3.1 BRANCHING HEURISTICS FOR BAB-ND PLANNING

The efficiency of BaB relies heavily on the quality of branching. Thus, selecting promising subdomains and determining effective ways to split them are two critical aspects of BaB, corresponding to `batch_pick_out(`$\mathbb{P}, n$`)` and `batch_split(`$\{\mathcal{C}_i\}$`)` in Algorithm 1. Here, we introduce our specialized branching heuristics designed to select and split subdomains to obtain high-quality solutions.

**Heuristic for selecting subdomains to split.** The function `batch_pick_out(`$\mathbb{P}, n$`)` picks $n$ most promising subdomains for branching, based on their associated $\underline{f}^*_{\mathcal{C}_i}$ or $\overline{f}^*_{\mathcal{C}_i}$. The pickout process must balance exploitation (focusing on areas around good solutions) and exploration (investigating regions that have not been thoroughly explored). *First*, we sort subdomains $\mathcal{C}_i$ by $\overline{f}^*_{\mathcal{C}_i}$ in ascending order and select the first $n_1$ subdomains to form $\{\mathcal{C}^1_{\text{pick}}\}$. Subdomains with smaller $\overline{f}^*_{\mathcal{C}_i}$ are prioritized, as good solutions have been found there. *Then*, we form another promising set $\{\mathcal{C}^2_{\text{pick}}\}$ by sampling $n - n_1$

---

**Algorithm 1** Branch and bound for planning. Comments are in brown.

---

1: **Function**: bab_planning
2: **Inputs**: $f, \mathcal{C}, n$ (batch size), terminate (Termination condition)
3: $\{(\overline{f}^*, \tilde{\boldsymbol{u}})\} \leftarrow \texttt{batch\_search}\,(f, \{\mathcal{C}\})$        ▷ Initially search on the whole $\mathcal{C}$
4: $\{\underline{f}^*\} \leftarrow \texttt{batch\_bound}\,(f, \{\mathcal{C}\})$        ▷ Initially bound on the whole $\mathcal{C}$
5: $\mathbb{P} \leftarrow \{(\mathcal{C}, \underline{f}^*, \overline{f}^*, \tilde{\boldsymbol{u}})\}$        ▷ $\mathbb{P}$ is the set of all candidate subdomains
6: **while** $\text{length}(\mathbb{P}) > 0$ **and not** terminate **do**
7:      $\{(\mathcal{C}_i, \underline{f}^*_{\mathcal{C}_i}, \overline{f}^*_{\mathcal{C}_i}, \tilde{\boldsymbol{u}}_{\mathcal{C}_i})\} \leftarrow \texttt{batch\_pick\_out}\,(\mathbb{P}, n)$        ▷ Pick out subdomains to split from $\mathbb{P}$
8:      $\{\mathcal{C}_i^{\text{lo}}, \mathcal{C}_i^{\text{up}}\} \leftarrow \texttt{batch\_split}\,(\{\mathcal{C}_i\})$        ▷ Splits each $\mathcal{C}_i$ into two subdomains $\mathcal{C}_i^{\text{lo}}$ and $\mathcal{C}_i^{\text{up}}$
9:      $\{\underline{f}^*_{\mathcal{C}_i^{\text{lo}}}, \underline{f}^*_{\mathcal{C}_i^{\text{up}}}\} \leftarrow \texttt{batch\_bound}\,(f, \{\mathcal{C}_i^{\text{lo}}, \mathcal{C}_i^{\text{up}}\})$        ▷ Estimate lower bounds on new subdomains
10:     $\{(\overline{f}^*_{\mathcal{C}_i^{\text{lo}}}, \tilde{\boldsymbol{u}}_{\mathcal{C}_i^{\text{lo}}}), (\overline{f}^*_{\mathcal{C}_i^{\text{up}}}, \tilde{\boldsymbol{u}}_{\mathcal{C}_i^{\text{up}}})\} \leftarrow \texttt{batch\_search}\,(f, \{\mathcal{C}_i^{\text{lo}}, \mathcal{C}_i^{\text{up}}\})$        ▷ Search new solutions
11:     **if** $\min\left(\{\overline{f}^*_{\mathcal{C}_i^{\text{lo}}}, \overline{f}^*_{\mathcal{C}_i^{\text{up}}}\}\right) < \overline{f}^*$ **then**
12:        $\overline{f}^* \leftarrow \min\left(\{\overline{f}^*_{\mathcal{C}_i^{\text{lo}}}, \overline{f}^*_{\mathcal{C}_i^{\text{up}}}\}\right), \tilde{\boldsymbol{u}} \leftarrow \arg\min\left(\{\overline{f}^*_{\mathcal{C}_i^{\text{lo}}}, \overline{f}^*_{\mathcal{C}_i^{\text{up}}}\}\right)$        ▷ Update the best solution if needed
13:     $\mathbb{P} \leftarrow \mathbb{P} \bigcup \texttt{Pruner}\left(\overline{f}^*, \{(\mathcal{C}_i^{\text{lo}}, \underline{f}^*_{\mathcal{C}_i^{\text{lo}}}, \overline{f}^*_{\mathcal{C}_i^{\text{lo}}}), (\mathcal{C}_i^{\text{up}}, \underline{f}^*_{\mathcal{C}_i^{\text{up}}}, \overline{f}^*_{\mathcal{C}_i^{\text{up}}})\}\right)$        ▷ Prune bad domains using $\overline{f}^*$
14: **Outputs:** $\overline{f}^*, \tilde{\boldsymbol{u}}$

---

subdomains from the remaining $N$ ones, using $\text{softmax}$, with the probability $p_i$ defined in Eq. 2, where $T$ is the temperature and $\underline{f}^*_{\mathcal{C}_i,\text{scaled}}$ represents $\underline{f}^*_{\mathcal{C}_i}$ after min-max normalization for numerical stability. A smaller $\underline{f}^*_{\mathcal{C}_i}$ may indicate potentially better solutions in $\mathcal{C}_i$, which should be prioritized:

$$p_i = \frac{\exp(-\underline{f}^*_{\mathcal{C}_i,\text{scaled}}/T)}{\sum_{j=1}^{N} \exp(-\underline{f}^*_{\mathcal{C}_j,\text{scaled}}/T)}. \tag{2}$$

Note that this heuristic was not discussed in the neural network verification literature, which requires verifying all subdomains, making the order of subdomain selection less critical.

**Heuristic for splitting subdomains.** batch_split $(\{\mathcal{C}_i\})$ partitions every $\mathcal{C}_i$ to help search for good solutions. For a box-constrained subdomain $\mathcal{C}_i := \{\boldsymbol{u}_j \mid \underline{\boldsymbol{u}}_j \leq \boldsymbol{u}_j \leq \overline{\boldsymbol{u}}_j; j = 0, \ldots, d-1\}$ (subscript $i$ omitted for brevity), it is natural to split it into two subdomains $\mathcal{C}_i^{\text{lo}}$ and $\mathcal{C}_i^{\text{up}}$ along a dimension $j^*$ by bisection. Specifically, $\mathcal{C}_i^{\text{lo}} = \{\boldsymbol{u}_j \mid \underline{\boldsymbol{u}}_{j^*} \leq \boldsymbol{u}_{j^*} \leq \frac{\underline{\boldsymbol{u}}_{j^*}+\overline{\boldsymbol{u}}_{j^*}}{2}\}$, $\mathcal{C}_i^{\text{up}} = \{\boldsymbol{u}_j \mid \frac{\underline{\boldsymbol{u}}_{j^*}+\overline{\boldsymbol{u}}_{j^*}}{2} \leq \boldsymbol{u}_{j^*} \leq \overline{\boldsymbol{u}}_{j^*}\}$. In both $\mathcal{C}_i^{\text{lo}}$ and $\mathcal{C}_i^{\text{up}}$, $\underline{\boldsymbol{u}}_j \leq \boldsymbol{u}_j \leq \overline{\boldsymbol{u}}_j, \forall j \neq j^*$ holds.

One native way to select $j^*$ is to choose the dimension with the largest input range $\overline{\boldsymbol{u}}_j - \underline{\boldsymbol{u}}_j$. This efficient strategy can help identify promising solutions since dimensions with a larger range often indicate greater variability or uncertainty in $f$. However, it does not consider the specific landscape of $f$, which may indicate dimensions better suited for splitting.

We additionally consider the distribution of top $w\%$ samples with the best objectives from *searching* to partition $\mathcal{C}_i$ into promising subdomains worth further searching. Specifically, for every dimension $j$, we record the number of top samples satisfying $\underline{\boldsymbol{u}}_j \leq \boldsymbol{u}_j \leq \frac{\underline{\boldsymbol{u}}_j+\overline{\boldsymbol{u}}_j}{2}$ and $\frac{\underline{\boldsymbol{u}}_j+\overline{\boldsymbol{u}}_j}{2} \leq \boldsymbol{u}_j \leq \overline{\boldsymbol{u}}_j$ as $n_j^{\text{lo}}$ and $n_j^{\text{up}}$, respectively. Then, $|n_j^{\text{lo}} - n_j^{\text{up}}|$ indicates the distribution bias of top samples along a dimension $j$. A dimension with large $|n_j^{\text{lo}} - n_j^{\text{up}}|$ is critical to objective values in $\mathcal{C}_i$ and should be prioritized to split due to the imbalanced samples on two sides. In this case, it is often possible that one of the two subdomains ($\mathcal{C}_i^{\text{lo}}$ and $\mathcal{C}_i^{\text{up}}$) contains better solutions, whereas the other has a larger lower bound for the objective and can be pruned.

Based on the discussion above, we rank input dimensions descendingly by $(\overline{\boldsymbol{u}}_j - \underline{\boldsymbol{u}}_j) \cdot |n_j^{\text{lo}} - n_j^{\text{up}}|$, select the top one as $j^*$, and then split $\mathcal{C}_i$ into two subdomains evenly on dimension $j^*$. This heuristic is notably different from those discussed in neural network verification literature (Bunel et al., 2018; 2020b), since we aim to find better feasible solutions, not better lower bounds.

### 3.2 BOUNDING METHOD FOR BAB-ND PLANNING

Our bounding procedure aims to provide a tight lower bound for the objective function $f(\boldsymbol{u})$ in any subdomain, enabling the pruning of unpromising subdomains and the identification of promising

ones. While this component is crucial to the effectiveness of BaB, grasping this high-level idea is sufficient to understand our main algorithm.

To guide the search with tight bound estimation, an important insight is that in the planning problem, a strictly sound lower bound is not required, as the lower bound is used to evaluate the quality of subdomains rather than to provide a provable guarantee of $f(\boldsymbol{u})$, as in neural network verification. Based on this observation, we propose two approaches, *propagation early-stop* and *searching-integrated bounding*, to obtain an efficient estimation of the lower bound $\underline{f}^{*}_{\mathcal{C}_i}$, leveraging popular bound propagation-based algorithms like CROWN (Zhang et al., 2018).

**Approach 1: Propagation early-stop.** CROWN is a bound propagation algorithm that propagates a linear lower bound (inequality) through the neural network and has been successfully used in BaB-based neural network verifiers for the bounding step (Xu et al., 2021; Wang et al., 2021). In CROWN, the linear bound will be propagated backward from the output (in our case, $f(\boldsymbol{u})$) to the input of the network (in our case, $\boldsymbol{u}$), and be concretized to a concrete lower bound value using the constraints on inputs (in our case, $\mathcal{C}_i$).

However, this linear bound can become increasingly loose in deeper networks and may result in vacuous lower bounds. In our planning setting with the neural dynamics model $f_{\text{dyn}}$, the long planning horizon $H$ in Eq. 1 requires unrolling $f_{\text{dyn}}$ $H$ times to form $f(\boldsymbol{u})$, leading to excessively loose bounds that are ineffective for pruning unpromising domains during BaB.

To address this challenge, we stop the bound propagation process early to avoid the excessively loose bound when propagated through multiple layers to the input $\boldsymbol{u}$. The linear bound will be concretized using intermediate layer bounds, as discussed in Approach 2, rather than the constraints on the inputs. A more formal description of this technique (with technical details on how CROWN is modified) is presented in Appendix B.2 with an illustrative example.

**Approach 2: Search-integrated bounding.** In CROWN, the propagation process requires recursively computing intermediate layer bounds (often referred to as *pre-activation bounds*). These pre-activation bounds represent the lower and upper bounds for any intermediate layer that is followed by a nonlinear layer. The time complexity of this process is quadratic with respect to the number of layers. Directly applying the original CROWN-like bound propagation is both ineffective and inefficient for long-horizon planning, as the number of pre-activation bounds increases with the planning horizon. This results in overly loose lower bounds due to the accumulated relaxation errors and high execution times.

To quickly obtain the pre-activation bounds, we can utilize the by-product of extensive sampling during *searching* to form the empirical bounds instead of recursively using CROWN to calculate these bounds. Specifically, we denote the intermediate layer output for layer $v$ as $\mathbf{g}_v(\boldsymbol{u})$, and assume we have $M$ samples $\boldsymbol{u}^m$ ($m = 1, \ldots, M$) from the *searching* process. We calculate the pre-activation lower and upper bounds as $\min_m \mathbf{g}_v(\boldsymbol{u}^m)$ and $\max_m \mathbf{g}_v(\boldsymbol{u}^m)$ for each dimension of $\mathbf{g}_v(\boldsymbol{u})$. Although these empirical bounds may underestimate the actual bounds, they are sufficient for CROWN to get a good estimation of $\underline{f}^{*}$ to guide the search.

### 3.3 Searching Approach for BaB-ND Planning

Given an objective function $f$ and a batch of subdomains $\{\mathcal{C}_i\}$, `batch_search`($f, \{\mathcal{C}_i\}$) explores these subdomains to find solutions, returning the best objectives and associated inputs $\{(\overline{f}^{*}_{\mathcal{C}_i}, \tilde{\boldsymbol{u}}_{\mathcal{C}_i})\}$. A large variety of sampling-based methods can be utilized and we currently adopt CEM as the underlying method. Other existing methods, such as MPPI or projected gradient descent (PGD), can be alternatives. In typical neural network verification literature, searching is often ignored during BaB (Wang et al., 2021; Bunel et al., 2020b) since these approaches do not seek feasible solutions.

To cooperate with the *bounding* component, we need to additionally record the outputs of needed intermediate layer $v$, and obtain their bounds as described in Section 3.2. Since we require the lower bound of the optimal objective $\underline{f}^{*}_{\mathcal{C}_i}$ for every $\mathcal{C}_i$, the outputs of layer $v$ must be calculated for every $\mathcal{C}_i$, using the samples within the subdomain.

Considering that the subdomains $\{\mathcal{C}_i\}$ will become progressively smaller, it is expected that sampling-based methods could provide good solutions. Moreover, since we always record $\overline{f}^{*}_{\mathcal{C}_i}$ and its associated $\tilde{\boldsymbol{u}}_{\mathcal{C}_i}$, they can initialize future searches on at least one of the split subdomains ($\mathcal{C}_i^{\text{lo}}$ and $\mathcal{C}_i^{\text{up}}$) from $\mathcal{C}_i$.

## 4    EXPERIMENTAL RESULTS

In this section, we evaluate the performance of our BaB-ND on a range of complex robotic manipulation tasks. Our primary objective is to investigate three key questions through experiments: (1) How **effectively** does BaB-ND handle long-horizon planning? (2) Is BaB-ND **applicable** to diverse manipulation scenarios involving multi-object interactions and deformable objects? (3) How does the **scalability** of BaB-ND compare to existing methods?

**Synthetic example.**    Before deploying our BaB-ND on robotic manipulation tasks, we design a synthetic function to evaluate its capability for finding optimal solutions in a highly non-convex problem. We define $f(\boldsymbol{u}) = \Sigma_{i=0}^{d-1} 5\boldsymbol{u}_i^2 + \cos 50\boldsymbol{u}_i$, $\boldsymbol{u} \in [-1, 1]^d$ where $d$ is the input dimension. The optimal solution $f^* \approx -1.9803d$ and $f(\boldsymbol{u})$ has 16 local optima with 2 global optima along each dimension. Hence, optimizing $f(\boldsymbol{u})$ can be challenging as $d$ increases.

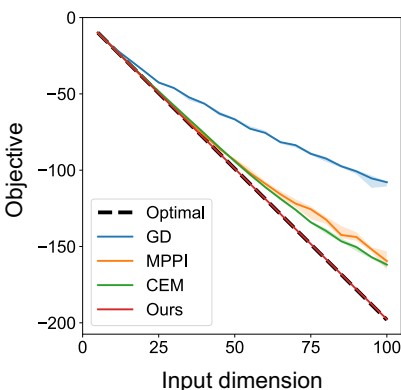

We compare our BaB-ND with three baselines: (1) **GD**: projected Gradient Descent on random samples with hyper-parameter tuning for step size; (2) **MPPI**: Model Predictive Path Integral with hyper-parameter tuning on noise level and reward temperature; (3) **CEM**: Decentralized Cross-Entropy Method (Zhang et al., 2022c) using an ensemble of independent CEM instances performing local improvements of their sampling distributions.

Figure 4: **Optimization result on a synthetic $f(\boldsymbol{u})$ over increasing dimensions $d$.** BaB-ND outperforms all baselines in terms of the optimized objective. We run all methods multiple times and visualize the median values with 25[th] and 75[th] percentiles in the shaded area.

In Figure 4, we visualize the best objective values found by different methods across different input dimensions up to $d = 100$. BaB-ND consistently outperforms all baselines which converge to sub-optimal values. For $d = 100$, BaB-ND can achieve optimality on 98 to 100 dimensions. This synthetic experiment demonstrates the potential of BaB-ND in planning problems involving neural dynamics.

**Experiment settings.**    We evaluate our BaB-ND on four complex robotic manipulation tasks involving non-smooth objectives, non-convex feasible regions, and requiring long action sequences. Different architectures of neural dynamics like MLP and GNN are leveraged for different scenarios. Please refer to Section D for more details about tasks, dynamics models, and cost functions.

   • **Pushing with Obstacles.** In Figure 5.a, this task involves using a pusher to manipulate a "T"-shaped object to reach a target pose while avoiding collisions with obstacles. An MLP neural dynamics model is trained with interactions between the pusher and object without obstacles. Obstacles are modeled in the cost function, resulting in non-smooth landscapes and non-convex feasible regions.

   • **Object Merging.** In Figure 5.c, two "L"-shaped objects are merged into a rectangle at a specific target pose, which requires a long action sequence with multiple contact mode switches.

   • **Rope Routing.** As shown in Figure 5.b, the goal is to route a deformable rope into a tight-fitting slot (modeled in the cost function) in the 3D action space. Instead of greedily approaching the target in initial steps, the robot needs to find the trajectory to finally reach the target.

   • **Object Sorting.** In Figure 5.d, a pusher interacts with a cluster of objects to sort one outlier object out of the central zone to the target while keeping others closely gathered. We use GNN to predict multi-object interactions. Every long-range action may significantly change the state. Additional constraints on actions are considered in the cost to avoid crashes between the robot and objects.

We evaluate baselines and BaB-ND on the open-loop planning performance (the best objective of Eq. 1 found) in simulation. Then, we select the best two baselines to evaluate their real-world closed-loop control performance (the final step cost or success rate of executions).

In real-world experiments, we first perform long-horizon planning to generate reference state trajectories and leverage MPC (Camacho & Bordons Alba, 2013) to efficiently track the trajectories in two tasks: *Pushing with Obstacles* and *Object Merging*. In the *Rope Routing* task, we directly execute the planned long-horizon action sequence due to its small sim-to-real gap. In the *Object Sorting* task, since the observations can change greatly after each push, we use MPC to replan after every action.

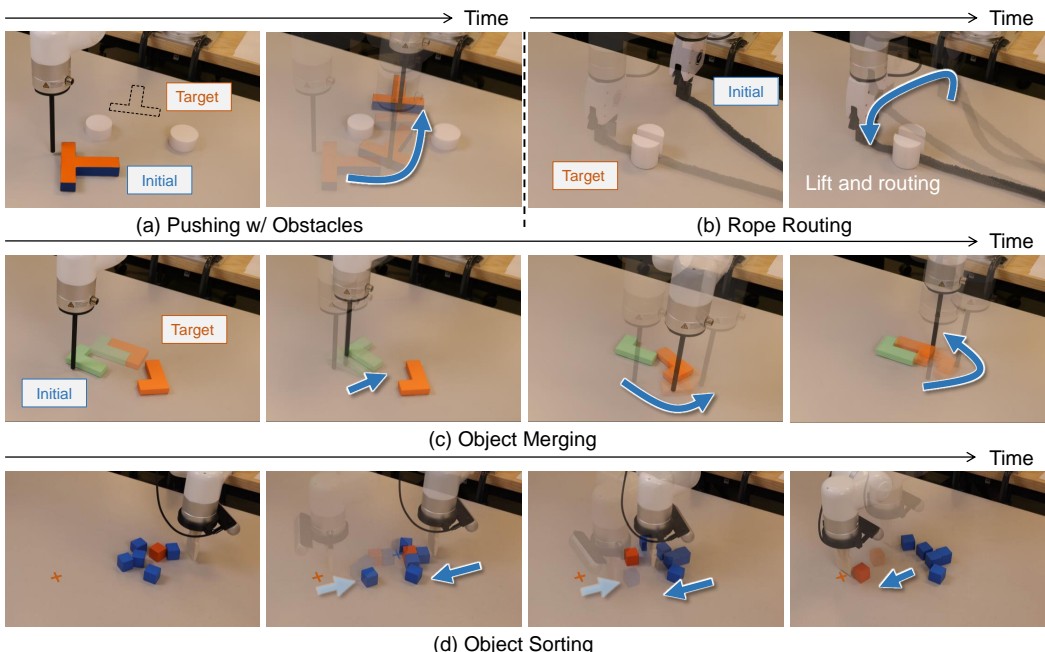

Figure 5: **Qualitative results on real-world manipulation tasks.** We evaluate our BaB-ND across four complex robotic manipulation tasks, involving non-convex feasible regions, requiring long-horizon planning, and interactions between multiple objects and the deformable rope. For each task, we visualize the initial and target configurations and one successful trajectory. Please refer to our **project page** for video demonstrations.

**Effectiveness.** We first evaluate the effectiveness of BaB-ND on *Pushing with Obstacles* and *Object Merging* tasks which are contact-rich and require strong long-horizon planning performance. The quantitative results of open-loop and closed loop performance for these tasks is presented in Figure 6.

In both tasks, our BaB-ND effectively optimizes the objective of Eq. 1 and gives better open-loop performance than all baselines. The well-planned trajectories can yield improved closed-loop performance in the real world with efficient tracking. Specifically, in the *Pushing with Obstacles* task, GD offers much worse trajectories than others, often resulting in the "T"-shaped object stuck at one obstacle. MPPI and CEM can offer trajectories passing through the obstacles but with poor alignment

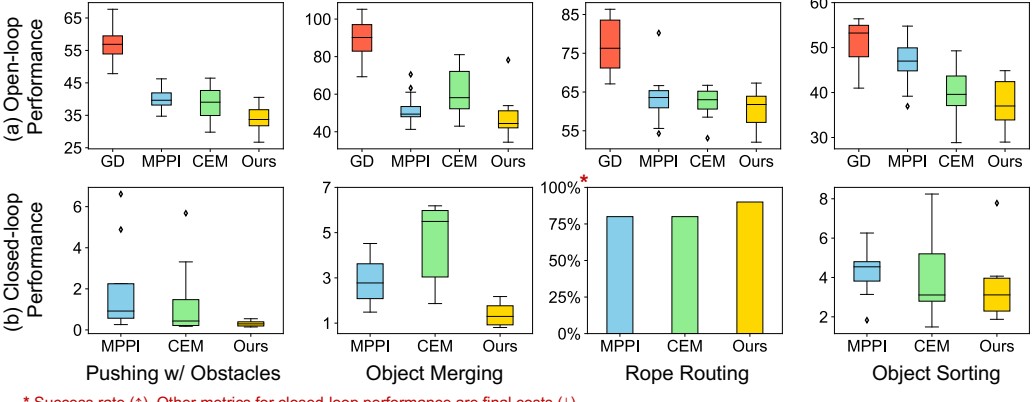

Figure 6: **Quantitative analysis of planning performance and execution performance in real world.** BaB-ND consistently outperforms baselines on open-loop performance leading to better closed-loop performance. (a) The open-loop performance of all tasks in simulation. We report the best objective of Eq. 1 found by all methods across different test cases. (b) The closed-loop performance in real world. GD is excluded from testing due to its poor open-loop performance. We report the success rate for *Rope Routing*, since a greedy trajectory that horizontally routes the rope may achieve a low final cost but fails to place it into the slot, and we report final step costs for all other tasks.

with the target. In contrast, BaB-ND can not only pass through obstacles successfully, but also often perfectly align with the final target.

**Applicability.** We assess the applicability of BaB-ND on *Rope Routing* and *Object Sorting* tasks involving the manipulation of deformable objects and interactions between multiple objects modeled by GNNs. The quantitative results in Figure 6 demonstrate our applicability to these tasks.

In the *Rope Routing* task, MPPI, CEM, and our method achieve comparable open-loop performance, while GD may struggle with suboptimal trajectories, often routing the rope horizontally and getting stuck outside the slot. In the *Object Sorting* task, CEM outperforms MPPI, as MPPI is better suited for planning continuous action sequences, whereas this task involves discrete actions. Our method outperforms CEM, achieving a similar median performance with noticeably lower variance.

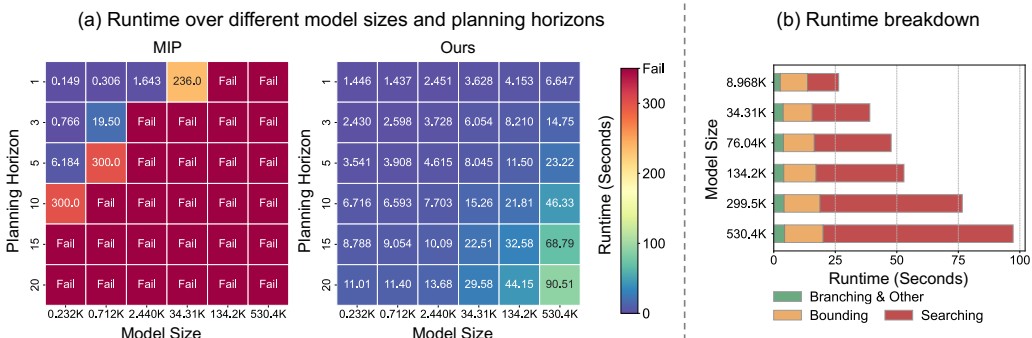

Figure 7: **Quantitative analysis of runtime and scalability.** (a) The runtime of MIP and BaB-ND for solving simple planning problems with varying model sizes and planning horizons. BaB-ND can handle significantly larger problems than MIP. ("Fail" indicates MIP fails to find any solution within 300 seconds.) (b) Runtime breakdown of our components on large and complex planning problems with $H = 20$. Runtimes for components other than searching increase slightly as model size grows, indicating the strong scalability of BaB-ND.

**Scalability.** We evaluate the scalability of our BaB-ND on *Pushing with Obstacles* task with varying model sizes and planning horizons on multiple test cases, in comparison to MIP (Liu et al., 2023b). We train the neural dynamics models with different sizes and the same architecture, and use the number of parameters in the single neural dynamics model $f_{\text{dyn}}$ to indicate the model size. Moreover, to accommodate MIP, we remove all obstacle-related components in cost and define the objective as the step cost after planning horizon $H$, $c(\hat{x}_{t_0+H}, u_{t_0+H})$ instead of the accumulated cost.

In Figure 7.a, we visualize the average runtime of MIP and ours on test cases with different model sizes and planning horizons. The results show that MIP only handles small problems. Among all 36 settings, it provides optimal solutions for 6, sub-optimal solutions for 3, and fails to find any solution for the remaining within 300 seconds. In contrast, our BaB-ND scales up well to large problems with planning horizon $H = 20$ and a model containing over 500K parameters.

In Figure 7.b, we evaluate the runtime of each primary component of our BaB-ND across various model sizes, ranging from approximately 9K to over 500K, using the original objective for the *Pushing with Obstacles* tasks (containing items to model obstacles and accumulated cost among all steps) over a planning horizon of $H = 20$. The breakdown bar chart illustrates that the runtimes for the *branching* and *bounding* components grow relatively slowly across model sizes, which increase more than 50-fold. Our improved bounding procedure, as discussed in Section 3.2, scales well with growing model size. In addition, the *searching* runtime scales proportionally to neural network size since the majority of searching time is spent on sampling the model with a large batch size on GPUs.

## 5 CONCLUSION

In this paper, we propose a branch-and-bound framework for long-horizon motion planning with neural dynamics in robotic manipulation tasks. We leverage specialized branching heuristics for systematic search and adapt the bound propagation algorithm from neural network verification to estimate tight objective bounds efficiently and reduce search space. Our framework demonstrates superior planning performance in complex, contact-rich manipulation tasks, as well as scalability and adaptability to various model architectures. The limitations and future directions are discussed in Section A.4.

## 6 ACKNOWLEDGMENT

This work is supported by the Toyota Research Institute (TRI). We thank Hongkai Dai, Aykut Onol, and Mengchao Zhang for their constructive suggestions on the paper manuscript. We also thank Mingtong Zhang, Haozhe Chen, Baoyu Li, and Binghao Huang for their help with real-world experiments and simulation environment development. This article reflects solely the opinions and conclusions of its authors, not those of TRI or any other Toyota entity.

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

# A    EXTENDED FORMULATION AND METHOD OVERVIEW

## A.1    ILLUSTRATION OF BAB-ND ON A SIMPLIFIED TASK

We refer to Figure 3 to introduce theoretical concepts in Section 3 and to illustrate BaB-ND on a simplified robotic manipulation task.

**Configuration.**    In Figure 3.a, we first define the configuration of the task, where the robot moves left or right to push an object toward the target.

The 1D *action* $u \in \mathcal{C}$ in this case represents the movement of the robot pusher, with $\mathcal{C} = [-l, l]$ as its domain, where $l$ is the maximum movement distance (e.g., 1 cm in practice). A value of $u < 0$ means the robot moves left, while $u > 0$ means the robot moves right.

The *objective* $f(u)$ measures the distance between the object and the target under a specific action $u$. In this case, $f(u) = d_1^2 + d_2^2$, where $d_1$ is the distance between a keypoint ($P_1$) on the object and the corresponding keypoint ($P_{1,T}$) on the target, and $d_2$ is the distance between another keypoint pair ($P_2$ and $P_{2,T}$). For example, if the robot moves left ($u < 0$), $d_2$ decreases while $d_1$ increases.

The values of $d_1$ and $d_2$ depend on a neural network *dynamics model* $f_{\mathrm{dyn}}$. This model takes as input the current positions of $P_1$ and $P_2$ relative to the pusher, along with an action $u$, to predict the next positions of $P_1$ and $P_2$. Based on these predictions, $d_1$ and $d_2$ are updated accordingly, and $f(u)$ may exhibit non-convex behavior.

**Formulation of BaB.**    Our goal in planning is to find the optimal action $u^*$ that minimizes $f(u)$. To achieve this, we propose a branch-and-bound-based method. In Figure 3.b, c, and d, we illustrate three components of our method. We first introduce some concepts below.

A *subdomain* $\mathcal{C}_i \in \mathcal{C}$ is a subset of the entire input domain $\mathcal{C}$. For example, in Figure 3.b, we initially split $\mathcal{C} = [-l, l]$ into two subdomains: $\mathcal{C}_1 = [-l, 0]$ and $\mathcal{C}_2 = [0, l]$, to separately analyze left and right movements.

Each subdomain $\mathcal{C}_i$ has associated *lower and upper bounds of the best objective* in it: $\underline{f}^*_{\mathcal{C}_i}$ and $\overline{f}^*_{\mathcal{C}_i}$. These represent the bounds of the best objective in $\mathcal{C}_i$ ($f^*_{\mathcal{C}_i} := \min_{u \in \mathcal{C}_i} f(u)$). For example, if the optimal objective (the sum of the squared distances between keypoint pairs, $d_1^2 + d_2^2$) given by the best action in $\mathcal{C}_i$ is 2, we might estimate $\underline{f}^*_{\mathcal{C}_i} = 1$ and $\overline{f}^*_{\mathcal{C}_i} = 3$ ($1 \leq \min_{u \in \mathcal{C}_i} d_1^2 + d_2^2 = 2 \leq 3$). Intuitively, $\underline{f}^*_{\mathcal{C}_i}$ underestimates the minimum objective in $\mathcal{C}_i$, while $\overline{f}^*_{\mathcal{C}_i}$ overestimates it.

We split the original domain $\mathcal{C}$ into multiple subdomains $\mathcal{C}_i$ with *branching*, compute $\underline{f}^*_{\mathcal{C}_i}$ using *bounding* (Figure 3.c), and $\overline{f}^*_{\mathcal{C}_i}$ using *searching* (Figure 3.d). These bounds allow us to determine whether a subdomain $\mathcal{C}_i$ is promising for containing the optimal action $u^*$ or whether it can be pruned as unpromising. For instance, in Figure 3, assume $\underline{f}^*_{\mathcal{C}_1} = 4$ and $\overline{f}^*_{\mathcal{C}_2} = 3$. This means that no objective better than 4 can be achieved in $\mathcal{C}_1$, while no objective worse than 3 can occur in $\mathcal{C}_2$. In this case, we can directly prune $\mathcal{C}_1$ without further exploration.

**Branching.**    In Figure 3.b, we visualize the *branching* process, which constructs a search tree iteratively. We first split $\mathcal{C} = [-l, l]$ into two subdomains: $\mathcal{C}_1 = [-l, 0]$ and $\mathcal{C}_2 = [0, l]$, allowing us to consider left and right movements separately. We can iteratively split any subdomain into smaller subdomains. For example, $\mathcal{C}_2$ can be further split into $\mathcal{C}_3$ and $\mathcal{C}_4$.

Naively, we could search every subdomain and select the best action among all subdomains as our final best action. However, this approach is computationally expensive, especially when $\mathcal{C}$ is divided into many small subdomains. Therefore, we need to prune unpromising subdomains to reduce the search space and computational overhead.

**Bounding.**    Pruning relies on the *bounding* component (Figure 3.c), which provides $\underline{f}^*$, the lower bound of $f(u)$ within a given input domain. In our simplified case, $\underline{f}^*$ represents the lower bound of the sum of the squared distances between keypoint pairs.

This bounding process is performed for every subdomain. Within a specific subdomain, such as $\mathcal{C}_1$, we estimate a linear function $g(u)$ that is always smaller than or equal to $f(\boldsymbol{u})$ in $\mathcal{C}_1$ (i.e., $g(u) \leq f(u), \forall u \in \mathcal{C}_1$). We then use the minimum value of $g(u)$ in $\mathcal{C}_1$ as the lower bound of $f(\boldsymbol{u})$ in $\mathcal{C}_1$ (i.e., $\underline{f}^*_{\mathcal{C}_1} := \min_{\boldsymbol{u} \in \mathcal{C}_1} g(u)$). This estimation is based on CROWN and our adaptations.

Intuitively, subdomains with large lower bounds can be treated as unpromising, while those with small lower bounds are considered promising. Using these lower bounds, we can prioritize the promising subdomains and prune unpromising subdomains whose lower bounds exceed $\overline{f}^*$, the best objective found so far.

**Searching.** The best objective found, $\overline{f}^* := \min_i \overline{f}^*_{\mathcal{C}_i}$, is the best objective among all subdomains, where $\overline{f}^*_{\mathcal{C}_i}$ represents the upper bound of the best objective in $\mathcal{C}_i$, obtained through the *searching* process using sampling-based methods, as shown in Figure 3.d.

Specifically, $\overline{f}^*_{\mathcal{C}_i} := \min_{\boldsymbol{u}_k \in \mathcal{C}_i} f(\boldsymbol{u}_k)$ is the best objective among all input samples $\boldsymbol{u}_k$ in $\mathcal{C}_i$. This is valid because $\forall \boldsymbol{u}_k \in \mathcal{C}_i, \overline{f}^* \leq f(\boldsymbol{u}_k)$ holds. Thus, any $f(\boldsymbol{u}_k)$ can serve as an upper bound for $\overline{f}^*_{\mathcal{C}_i}$, but we select the best one to achieve a tighter bound on $\overline{f}^*_{\mathcal{C}_i}$.

With more subdomains being pruned in the branch-and-bound process, sampling-based methods can be applied to progressively smaller input spaces, enabling the discovery of better objectives. This process may ultimately converge to the actual optimal value $f^*$ and identify the optimal action $\boldsymbol{u}^*$.

## A.2 Algorithm of BaB-ND

The BaB-ND algorithm Algorithm 2 takes an objective function $f$ with neural networks, a domain $\mathcal{C}$ as the input space, and a termination condition if necessary. The sub-procedure `batch_search` seeks better solutions on domains $\{\mathcal{C}_i\}$. It returns the best objectives $\{\overline{f}^*_{\mathcal{C}_i}\}$ and the corresponding solutions $\{\tilde{\boldsymbol{u}}_{\mathcal{C}_i}\}$ for $n$ selected subdomains simultaneously. The sub-procedure `batch_bound` computes the lower bounds of $f^*$ on domains $\{\mathcal{C}_i\}$ as described in Section 3.2. It operates in a batch and returns the lower bounds $\{\underline{f}^*_{\mathcal{C}_i}\}$.

In the algorithm, we maintain $\overline{f}^*$ and $\tilde{\boldsymbol{u}}$ as the best objective and solution we can find. We also maintain a global set $\mathbb{P}$ storing all the candidate subdomains where $\underline{f}^*_{\mathcal{C}_i} \geq \overline{f}^*$. Initially, we only have the whole input domain $\mathcal{C}$, so we perform `batch_search` and `batch_bound` on $\mathcal{C}$ and initialize the current $\overline{f}^*$, $\tilde{\boldsymbol{u}}$, and $\mathbb{P}$ (Lines 2-4).

Then we utilize the power of GPUs to branch, search, and bound subdomains in parallel while maintaining $\mathbb{P}$ (Lines 6-11). Specifically, `batch_pick_out` selects $n$ (batch size) promising subdomains from $\mathbb{P}$. If the length of $\mathbb{P}$ is less than $n$, then $n$ is reduced to the length of $\mathbb{P}$. `batch_split` splits each selected $\mathcal{C}_i$ into two subdomains $\mathcal{C}^{\text{lo}}_i$ and $\mathcal{C}^{\text{up}}_i$ according to a branch heuristic in parallel. `Pruner` filters out bad subdomains (proven with $\underline{f}^*_{\mathcal{C}_i} > \overline{f}^*$), and the remaining ones are inserted into $\mathbb{P}$.

The loop breaks if there are no subdomains left in $\mathbb{P}$ or if some other pre-defined termination conditions, such as timeout or finding a good enough objective $\overline{f}^* \leq f_{th}$, are satisfied (Line 5). We finally return the best objective $\overline{f}^*$ and the corresponding solution $\tilde{\boldsymbol{u}}$.

## A.3 Distinctions between BaB-ND and neural network verification algorithms

**Goals.** BaB-ND aims to optimize an objective function involving neural dynamics models to solve challenging planning problems, seeking a *concrete solution* $\tilde{\boldsymbol{u}}$ (a near-optimal action sequence) to an objective-minimization problem $\min_{\boldsymbol{u} \in \mathcal{C}} f(\boldsymbol{u})$. In contrast, neural network verification focuses on *proving a sound lower bound* of $f(\boldsymbol{u})$ in the space $\mathcal{C}$, where a *concrete solution* $\tilde{\boldsymbol{u}}$ is not needed. These fundamental distinctions in goals lead to different algorithm design choices.

**Branching Heuristics.** In BaB-ND, branching heuristics are designed to effectively guide the search for better concrete solutions, considering both the lower and upper bounds of the best objective. In neural network verification, branching heuristics focus solely on improving the lower bounds.

---

**Algorithm 2** Branch and bound for planning. Comments are in brown.

---

1: **Inputs**: $f$, $\mathcal{C}$, $n$ (batch size), `terminate` (Termination condition)
2: $\{(\overline{f}^*, \tilde{\boldsymbol{u}})\} \leftarrow$ `batch_search` $(f, \{\mathcal{C}\})$ ▷ Initially search on the whole $\mathcal{C}$
3: $\{\underline{f}^*\} \leftarrow$ `batch_bound` $(f, \{\mathcal{C}\})$ ▷ Initially bound on the whole $\mathcal{C}$
4: $\mathbb{P} \leftarrow \{(\mathcal{C}, \underline{f}^*, \overline{f}^*, \tilde{\boldsymbol{u}})\}$ ▷ $\mathbb{P}$ is the set of all candidate subdomains
5: **while** `length`($\mathbb{P}$) $> 0$ **and not** `terminate` **do**
6:     $\{(\mathcal{C}_i, \underline{f}^*_{\mathcal{C}_i}, \overline{f}^*_{\mathcal{C}_i}, \tilde{\boldsymbol{u}}_{\mathcal{C}_i})\} \leftarrow$ `batch_pick_out` $(\mathbb{P}, n)$ ▷ Pick subdomains to split and remove them from $\mathbb{P}$
7:     $\{\mathcal{C}^{\text{lo}}_i, \mathcal{C}^{\text{up}}_i\} \leftarrow$ `batch_split` $(\{\mathcal{C}_i\})$ ▷ Each $\mathcal{C}_i$ splits into two subdomains $\mathcal{C}^{\text{lo}}_i$ and $\mathcal{C}^{\text{up}}_i$
8:     $\{(\overline{f}^*_{\mathcal{C}^{\text{lo}}_i}, \tilde{\boldsymbol{u}}_{\mathcal{C}^{\text{lo}}_i}), (\overline{f}^*_{\mathcal{C}^{\text{up}}_i}, \tilde{\boldsymbol{u}}_{\mathcal{C}^{\text{up}}_i})\} \leftarrow$ `batch_search` $(f, \{\mathcal{C}^{\text{lo}}_i, \mathcal{C}^{\text{up}}_i\})$ ▷ Search new solutions
9:     $\{\underline{f}^*_{\mathcal{C}^{\text{lo}}_i}, \underline{f}^*_{\mathcal{C}^{\text{up}}_i}\} \leftarrow$ `batch_bound` $(f, \{\mathcal{C}^{\text{lo}}_i, \mathcal{C}^{\text{up}}_i\})$ ▷ Compute lower bounds on new subdomains
10:     **if** $\min\left(\{\overline{f}^*_{\mathcal{C}^{\text{lo}}_i}, \overline{f}^*_{\mathcal{C}^{\text{up}}_i}\}\right) < \overline{f}^*$ **then**
11:         $\overline{f}^* \leftarrow \min\left(\{\overline{f}^*_{\mathcal{C}^{\text{lo}}_i}, \overline{f}^*_{\mathcal{C}^{\text{up}}_i}\}\right)$, $\tilde{\boldsymbol{u}} \leftarrow \arg\min\left(\{\overline{f}^*_{\mathcal{C}^{\text{lo}}_i}, \overline{f}^*_{\mathcal{C}^{\text{up}}_i}\}\right)$ ▷ Update the best solution if needed
12:     $\mathbb{P} \leftarrow \mathbb{P} \bigcup$ `Pruner` $\left(\overline{f}^*, \{(\mathcal{C}^{\text{lo}}_i, \underline{f}^*_{\mathcal{C}^{\text{lo}}_i}, \overline{f}^*_{\mathcal{C}^{\text{lo}}_i}), (\mathcal{C}^{\text{up}}_i, \underline{f}^*_{\mathcal{C}^{\text{up}}_i}, \overline{f}^*_{\mathcal{C}^{\text{up}}_i})\}\right)$ ▷ Prune bad domains using $\overline{f}^*$
13: **Outputs**: $\overline{f}^*$, $\tilde{\boldsymbol{u}}$

---

**Bounding Approaches.** While existing bounding approaches, such as CROWN from neural network verification, can provide provable lower bounds for objectives, they are neither effective nor efficient for planning problems. To address this, we adapt the CROWN algorithm with propagation early-stop and search-integrated bounding to efficiently obtain tight bound estimations.

**Searching Components.** BaB-ND includes an additional searching component in the branch-and-bound procedure to find the optimal solution to planning problems. Neural network verifiers typically do not have this component, as they focus solely on obtaining lower bounds of objectives over an input space rather than identifying objective values for specific inputs. We further adapt the searching component to benefit from the BaB procedure while also guiding BaB for improved searching.

### A.4 LIMITATIONS AND FUTURE DIRECTIONS

In this section, we discuss a few limitations of our work and potential directions for future work.

- *Planning performance depends on the prediction errors of neural dynamics models.*

The neural dynamics model may not perfectly match the real world. As a result, our optimization framework, BaB-ND, may achieve a low objective as predicted by the learned dynamics model but still miss the target (e.g., the model predicts that a certain action reaches the target, but in reality, the pushing action overshoots). While improving model accuracy is not the primary focus of this paper, future research could explore more robust formulations that account for potential errors in neural dynamics models to improve overall performance and reliability.

- *Optimality of our solution may be influenced by the underlying searching algorithms.*

The planning performance of BaB-ND is inherently influenced by the underlying sampling-based searching algorithms (e.g., sampling-based methods may over-exploit or over-explore the objective landscape, resulting in suboptimal solutions in certain domains). Although our branch-and-bound procedure can mitigate this issue by systematically exploring the input space and efficiently guiding the search, incorporating advanced sampling-based searching algorithms with proper parameter scheduling into BaB-ND could improve its ability to tackle more challenging planning problems.

- *Improved branching heuristics and strategies are needed for more efficiently guiding the search for more challenging settings.*

There is still room for improving the branching heuristics and bounding strategies to generalize across diverse tasks (e.g., our current strategy may not always find the optimal axis to branch). Future efforts could focus on developing more generalizable strategies for broader applications, potentially leveraging reinforcement learning approaches.

## B MORE DETAILS ABOUT BOUNDING

### B.1 PROOFS OF CROWN BOUNDING

In this section, we first share the background of neural network verification including its formulation and a efficient linear bound propagation method CROWN (Zhang et al., 2018) to calculate bounds over neural networks. We take the Multilayer perceptron (MLP) with ReLU activation as the example and CROWN is a general framework which is suitable to different activations and model architectures.

**Definition.** We define the input of a neural network as $x \in \mathbb{R}^{d_0}$, and define the weights and biases of an $L$-layer neural network as $\mathbf{W}^{(i)} \in \mathbb{R}^{d_i \times d_{i-1}}$ and $\mathbf{b}^{(i)} \in \mathbb{R}^{d_i}$ ($i \in \{1, \cdots, L\}$) respectively. The neural network function $f : \mathbb{R}^{d_0} \to \mathbb{R}$ is defined as $f(x) = z^{(L)}(x)$, where $z^{(i)}(x) = \mathbf{W}^{(i)}\hat{z}^{(i-1)}(x) + \mathbf{b}^{(i)}$, $\hat{z}^{(i)}(x) = \sigma(z^{(i)}(x))$ and $\hat{z}^{(0)}(x) = x$. $\sigma$ is the activation function and we use ReLU throughout this paper. When the context is clear, we omit $\cdot(x)$ and use $z_j^{(i)}$ and $\hat{z}_j^{(i)}$ to represent the *pre-activation* and *post-activation* values of the $j$-th neuron in the $i$-th layer. Neural network verification seeks the solution of the optimization problem in Eq. 3:

$$\min f(x) := z^{(L)} \quad \text{s.t.} \quad z^{(i)} = \mathbf{W}^{(i)}\hat{z}^{(i-1)} + \mathbf{b}^{(i)}, \hat{z}^{(i)} = \sigma(z^{(i)}), x \in \mathcal{C}, i \in \{1, \cdots, L-1\} \quad (3)$$

The set $\mathcal{C}$ defines the allowed input region and our aim is to find the minimum of $f(x)$ for $x \in \mathcal{C}$, and throughout this paper we consider $\mathcal{C}$ as an $\ell_p$ ball around a data example $x_0$: $\mathcal{C} = \{x \mid \|x - x_0\|_p \leq \epsilon\}$.

First, let we consider the neural network with only linear layers. in this case, it is easily to get a linear relationship between $x$ and $f(x)$ that $f(x) = \mathbf{W}x + \mathbf{b}$ no matter what is the value of $L$ and derive the closed form of $f^* = \min f(x)$ for $x \in \mathcal{C}$. With this idea in our mind, for neural networks with non-linear activation layers, if we could bound them with some linear functions, then it is still possible to bound $f(x)$ with linear functions.

Then, we show that the non-linear activation ReLU layer $\hat{z} = \text{ReLU}(z)$ can be bounded by two linear functions in three cases according to the range of pre-activation bounds $\mathbf{l} \leq z \leq \mathbf{u}$: active ($\mathbf{l} \geq 0$), inactive ($\mathbf{u} \leq 0$) and unstable ($\mathbf{l} < 0 < \mathbf{u}$) in Lemma B.1.

**Lemma B.1** (Relaxation of a ReLU layer in CROWN). *Given pre-activation vector $z \in \mathbb{R}^d, \mathbf{l} \leq z \leq \mathbf{u}$ (element-wise), $\hat{z} = \text{ReLU}(z)$, we have*

$$\underline{\mathbf{D}}z + \underline{\mathbf{b}} \leq \hat{z} \leq \overline{\mathbf{D}}z + \overline{\mathbf{b}},$$

*where $\underline{\mathbf{D}}, \overline{\mathbf{D}} \in \mathbb{R}^{d \times d}$ are diagonal matrices defined as:*

$$\underline{\mathbf{D}}_{j,j} = \begin{cases} 1, & \text{if } \mathbf{l}_j \geq 0 \\ 0, & \text{if } \mathbf{u}_j \leq 0 \\ \boldsymbol{\alpha}_j, & \text{if } \mathbf{u}_j > 0 > \mathbf{l}_j \end{cases} \qquad \overline{\mathbf{D}}_{j,j} = \begin{cases} 1, & \text{if } \mathbf{l}_j \geq 0 \\ 0, & \text{if } \mathbf{u}_j \leq 0 \\ \frac{\mathbf{u}_j}{\mathbf{u}_j - \mathbf{l}_j}, & \text{if } \mathbf{u}_j > 0 > \mathbf{l}_j \end{cases} \qquad (4)$$

*$\boldsymbol{\alpha} \in \mathbb{R}^d$ is a free vector s.t., $0 \leq \boldsymbol{\alpha} \leq 1$. $\underline{\mathbf{b}}, \overline{\mathbf{b}} \in \mathbb{R}^d$ are defined as*

$$\underline{\mathbf{b}}_j = \begin{cases} 0, & \text{if } \mathbf{l}_j > 0 \text{ or } \mathbf{u}_j \leq 0 \\ 0, & \text{if } \mathbf{u}_j > 0 > \mathbf{l}_j. \end{cases} \qquad \overline{\mathbf{b}}_j = \begin{cases} 0, & \text{if } \mathbf{l}_j > 0 \text{ or } \mathbf{u}_j \leq 0 \\ -\frac{\mathbf{u}_j \mathbf{l}_j}{\mathbf{u}_j - \mathbf{l}_j}, & \text{if } \mathbf{u}_j > 0 > \mathbf{l}_j. \end{cases} \qquad (5)$$

*Proof.* For the $j$-th ReLU neuron, if $\mathbf{l}_j \geq 0$, then $\text{ReLU}(z_j) = z_j$; if $\mathbf{u}_j < 0$, then $\text{ReLU}(z_j) = 0$. For the case of $\mathbf{l}_j < 0 < \mathbf{u}_j$, the ReLU function can be linearly upper and lower bounded within this range:

$$\boldsymbol{\alpha}_j z_j \leq \text{ReLU}(z_j) \leq \frac{\mathbf{u}_j}{\mathbf{u}_j - \mathbf{l}_j}(z_j - \mathbf{l}_j) \quad \forall \mathbf{l}_j \leq z_j \leq \mathbf{u}_j$$

where $0 \leq \boldsymbol{\alpha}_j \leq 1$ is a free variable - any value between 0 and 1 produces a valid lower bound. $\square$

Next we apply the linear relaxation of ReLU to the $L$-layer neural network $f(x)$ to further derive the linear lower bound of $f(x)$. The idea is to propagate a weight matrix $\widetilde{\mathbf{W}}$ and bias vector $\widetilde{\mathbf{b}}$ from the $L$-th layer to 1-th layer. Specifically, when propagate through ReLU layer, we should greedily select upper bound of $\hat{z}_j$ when $\widetilde{\mathbf{W}}_{i,j}$ is negative and select lower bound of $\hat{z}_j$ when $\widetilde{\mathbf{W}}_{i,j}$ is positive to calculate the lower bound of $f(x)$. When propagate through linear layer, we do not need to do such selection since there is no relaxation on linear layer.

**Theorem B.2** (CROWN bound propagation on neural network). *Given the $L$-layer neural network $f(x)$ as defined in Eq. 3, we could find a linear function with respect to input $x$.*

$$f(x) := z^{(L)} \geq \widetilde{\underline{\mathbf{W}}}^{(1)} x + \widetilde{\underline{\mathbf{b}}}^{(1)} \tag{6}$$

*where $\widetilde{\underline{\mathbf{W}}}$ and $\widetilde{\underline{\mathbf{b}}}$ are recursively defined as following:*

$$\widetilde{\underline{\mathbf{W}}}^{(l)} = \underline{\mathbf{A}}^{(l)} \mathbf{W}^{(l)}, \widetilde{\underline{\mathbf{b}}}^{(l)} = \underline{\mathbf{A}}^{(l)} \mathbf{b}^{(l)} + \underline{\mathbf{d}}^{(l)}, \forall l = 1 \dots L \tag{7}$$

$$\underline{\mathbf{A}}^{(L)} = \mathbf{I} \in \mathbb{R}^{d_L \times d_L}, \widetilde{\underline{\mathbf{b}}}^{(L)} = 0 \tag{8}$$

$$\underline{\mathbf{A}}^{(l)} = \widetilde{\underline{\mathbf{W}}}_{\geq 0}^{(l+1)} \underline{\mathbf{D}}^{(l)} + \widetilde{\underline{\mathbf{W}}}_{<0}^{(l+1)} \overline{\mathbf{D}}^{(l)} \in \mathbb{R}^{d_{l+1} \times d_l}, \forall l = 1 \dots L - 1 \tag{9}$$

$$\underline{\mathbf{d}}^{(l)} = \widetilde{\underline{\mathbf{W}}}_{\geq 0}^{(l+1)} \underline{\mathbf{b}}^{(l)} + \widetilde{\underline{\mathbf{W}}}_{<0}^{(l+1)} \overline{\mathbf{b}}^{(l)} + \widetilde{\underline{\mathbf{b}}}^{(l)}, \forall l = 1 \dots L - 1 \tag{10}$$

*where $\forall l = 1 \dots L - 1, \underline{\mathbf{D}}^{(l)}, \overline{\mathbf{D}}^{(l)} \in \mathbb{R}^{d_l \times d_l}$ and $\underline{\mathbf{b}}^{(l)}, \overline{\mathbf{b}}^{(l)} \in \mathbb{R}^{d_l}$ are defined as in Lemma B.1. And subscript "$\geq 0$" stands for taking positive elements from the matrix while setting other elements to zero, and vice versa for subscript "$< 0$".*

*Proof.* First we have

$$\begin{aligned} f(x) := z^{(L)} &= \underline{\mathbf{A}}^{(L)} z^{(L)} + \underline{\mathbf{d}}^{(L)} \\ &= \underline{\mathbf{A}}^{(L)} \mathbf{W}^{(L)} \hat{z}^{(L-1)} + \underline{\mathbf{A}}^{(L)} \mathbf{b}^{(L)} + \underline{\mathbf{d}}^{(L)} \\ &= \widetilde{\underline{\mathbf{W}}}^{(L)} \hat{z}^{(L-1)} + \widetilde{\underline{\mathbf{b}}}^{(L)} \end{aligned} \tag{11}$$

Refer to Lemma B.1, we have

$$\underline{\mathbf{D}}^{(L-1)} z^{(L-1)} + \underline{\mathbf{b}}^{(L-1)} \leq \hat{z}^{(L-1)} \leq \overline{\mathbf{D}}^{(L-1)} z^{(L-1)} + \overline{\mathbf{b}}^{(L-1)} \tag{12}$$

Then we can form the lower bound of $z^{(L)}$ element by element: we greedily select the upper bound $\hat{z}_j^{(L-1)} \leq \overline{\mathbf{D}}_{j,j}^{(L-1)} z_j^{(L-1)} + \overline{\mathbf{b}}_j^{(L-1)}$ when $\widetilde{\underline{\mathbf{W}}}_{i,j}^{(L)}$ is negative, and select the lower bound $\hat{z}_j^{(L-1)} \geq \underline{\mathbf{D}}_{j,j}^{(L-1)} z_j^{(L-1)} + \underline{\mathbf{b}}_j^{(L-1)}$ otherwise. It can be formatted as

$$\widetilde{\underline{\mathbf{W}}}^{(L)} \hat{z}^{(L-1)} + \widetilde{\underline{\mathbf{b}}}^{(L)} \geq \underline{\mathbf{A}}^{(L-1)} z^{(L-1)} + \underline{\mathbf{d}}^{(L-1)} \tag{13}$$

where $\underline{\mathbf{A}}^{(L-1)} \in \mathbb{R}^{d_L \times d_{L-1}}$ is defined as

$$\underline{\mathbf{A}}_{i,j}^{(L-1)} = \begin{cases} \widetilde{\underline{\mathbf{W}}}_{i,j}^{(L)} \overline{\mathbf{D}}_{j,j}^{(L-1)}, & \text{if } \widetilde{\underline{\mathbf{W}}}_{i,j}^{(L)} < 0 \\ \widetilde{\underline{\mathbf{W}}}_{i,j}^{(L)} \underline{\mathbf{D}}_{j,j}^{(L-1)}, & \text{if } \widetilde{\underline{\mathbf{W}}}_{i,j}^{(L)} \geq 0 \end{cases} \tag{14}$$

for simplicity, we rewrite it in matrix form as

$$\underline{\mathbf{A}}^{(L-1)} = \widetilde{\underline{\mathbf{W}}}_{\geq 0}^{(L)} \underline{\mathbf{D}}^{(L-1)} + \widetilde{\underline{\mathbf{W}}}_{<0}^{(L)} \overline{\mathbf{D}}^{(L-1)} \tag{15}$$

And $\underline{\mathbf{d}}^{(L-1)} \in \mathbb{R}^{d_L}$ is similarly defined as

$$\underline{\mathbf{d}}^{(L-1)} = \widetilde{\underline{\mathbf{W}}}_{\geq 0}^{(L)} \underline{\mathbf{b}}^{(L-1)} + \widetilde{\underline{\mathbf{W}}}_{<0}^{(L)} \overline{\mathbf{b}}^{(L-1)} + \widetilde{\underline{\mathbf{b}}}^{(L)} \tag{16}$$

Then we continue to replace $z^{(L-1)}$ in Eq. 13 as $\mathbf{W}^{(L-1)} \hat{z}^{(L-2)} + \mathbf{b}^{(L-1)}$

$$\begin{aligned} \widetilde{\underline{\mathbf{W}}}^{(L)} \hat{z}^{(L-1)} + \widetilde{\underline{\mathbf{b}}}^{(L)} &\geq (\underline{\mathbf{A}}^{(L-1)} \mathbf{W}^{(L-1)}) \hat{z}^{(L-2)} + \underline{\mathbf{A}}^{(L-1)} \mathbf{b}^{(L-1)} + \underline{\mathbf{d}}^{(L-1)} \\ &= \widetilde{\underline{\mathbf{W}}}^{(L-1)} \hat{z}^{(L-2)} + \widetilde{\underline{\mathbf{b}}}^{(L-1)} \end{aligned} \tag{17}$$

By continuing to propagate the linear inequality to the first layer, we get

$$f(x) \geq \widetilde{\underline{\mathbf{W}}}^{(1)} \hat{z}^{(0)} + \widetilde{\underline{\mathbf{b}}}^{(1)} = \widetilde{\underline{\mathbf{W}}}^{(1)} x + \widetilde{\underline{\mathbf{b}}}^{(1)} \tag{18}$$

$\square$

After getting the linear lower bound of $f(x)$, and given $x \in \mathcal{C}$, we could solve the linear lower bound in closed form as in Theorem B.3. It is given by the Hölder's inequality.

**Theorem B.3** (Bound Concretization under $\ell_p$ ball Perturbations). *Given the L-layer neural network $f(x)$ as defined in Eq. 3, and input $x \in \mathcal{C} = \mathbb{B}_p(x_0, \epsilon) = \{x \mid \|x - x_0\|_p \leq \epsilon\}$, we could find concrete lower bound of $f(x)$ by solving the optimization problem $\min_{x \in \mathcal{C}} \underline{\widetilde{\mathbf{W}}}^{(1)} x + \underline{\widetilde{\mathbf{b}}}^{(1)}$ and its solution gives*

$$\min_{x \in \mathcal{C}} f(x) \geq \min_{x \in \mathcal{C}} \underline{\widetilde{\mathbf{W}}}^{(1)} x + \underline{\widetilde{\mathbf{b}}}^{(1)} \geq -\epsilon \|\underline{\widetilde{\mathbf{W}}}^{(1)}\|_q + \underline{\widetilde{\mathbf{W}}}^{(1)} x_0 + \underline{\widetilde{\mathbf{b}}}^{(1)} \tag{19}$$

*where $\frac{1}{p} + \frac{1}{q} = 1$ and $\| \cdot \|_q$ denotes taking $\ell_q$-norm for each row in the matrix and the result makes up a vector.*

*Proof.*

$$\min_{x \in \mathcal{C}} \underline{\widetilde{\mathbf{W}}}^{(1)} x + \underline{\widetilde{\mathbf{b}}}^{(1)} \tag{20}$$

$$= \min_{\lambda \in \mathbb{B}_p(0,1)} \underline{\widetilde{\mathbf{W}}}^{(1)}(x_0 + \epsilon \lambda) + \underline{\widetilde{\mathbf{b}}}^{(1)} \tag{21}$$

$$= \epsilon \left( \min_{\lambda \in \mathbb{B}_p(0,1)} \underline{\widetilde{\mathbf{W}}}^{(1)} \lambda \right) + \underline{\widetilde{\mathbf{W}}}^{(1)} x_0 + \underline{\widetilde{\mathbf{b}}}^{(1)} \tag{22}$$

$$= -\epsilon \left( \max_{\lambda \in \mathbb{B}_p(0,1)} -\underline{\widetilde{\mathbf{W}}}^{(1)} \lambda \right) + \underline{\widetilde{\mathbf{W}}}^{(1)} x_0 + \underline{\widetilde{\mathbf{b}}}^{(1)} \tag{23}$$

$$\geq -\epsilon \left( \max_{\lambda \in \mathbb{B}_p(0,1)} |\underline{\widetilde{\mathbf{W}}}^{(1)} \lambda| \right) + \underline{\widetilde{\mathbf{W}}}^{(1)} x_0 + \underline{\widetilde{\mathbf{b}}}^{(1)} \tag{24}$$

$$\geq -\epsilon \left( \max_{\lambda \in \mathbb{B}_p(0,1)} \|\underline{\widetilde{\mathbf{W}}}^{(1)}\|_q \|\lambda\|_p \right) + \underline{\widetilde{\mathbf{W}}}^{(1)} x_0 + \underline{\widetilde{\mathbf{b}}}^{(1)} \text{ (Hölder's inequality)} \tag{25}$$

$$= -\epsilon \|\underline{\widetilde{\mathbf{W}}}^{(1)}\|_q + \underline{\widetilde{\mathbf{W}}}^{(1)} x_0 + \underline{\widetilde{\mathbf{b}}}^{(1)} \tag{26}$$

$\square$

## B.2 DETAILS ABOUT BOUND PROPAGATION EARLY-STOP

We parse the objective function $f$ into a computational graph $\mathcal{G} = (\mathbf{V}, \mathbf{E})$, where $\mathbf{V}$ and $\mathbf{E}$ are the sets of nodes and edges, respectively. This process can be accomplished using popular deep learning frameworks, such as PyTorch, which support not only neural networks but also more general functions. In the graph $\mathcal{G}$, any mathematical operation is represented as a node $v \in \mathbf{V}$, and the edges $e = (w, v) \in \mathbf{E}$ define the flow of computation. The input $\boldsymbol{u}$, constant values, and model parameters constitute the input nodes of $\mathcal{G}$, forming the input set $\mathcal{I} = \{v \mid \text{In}(v) = \emptyset\}$, where $\text{In}(v) = \{w \mid (w, v) \in \mathbf{E}\}$ denotes the set of input nodes for a node $v$. Any arithmetic operation, such as ReLU, which requires input operands, is also represented as a node in $\mathcal{G}$ but with a non-empty input set. The node $o$ is the sole output node of $\mathcal{G}$ and provides the scalar objective value $f$ in our case.

---

**Algorithm 3** Bound Propagation w/ Early-stop.

1: **Function**: `compute_bound`
2: **Inputs**: computational graph $\mathcal{G}$, output node $o$, early-stop set $\mathcal{S}$
3: `CROWN_init`$(\mathcal{G}, o)$
4: $Q \leftarrow$ `Queue`$(), Q.\text{push}(o)$
5: **while** `length`$(Q) > 0$ **do**
6:     $v \leftarrow Q.\text{pop}()$
7:     **for** $w \in \text{In}(v)$ **do**
8:         $d_w \mathrel{-}= 1$
9:         **if** $d_w = 0$ and $w \notin \mathcal{I}$ **then**
10:           $Q.\text{push}(w)$
11:     **if** $v \in \mathcal{S}$ **then**
12:         **continue**
13:     `CROWN_prop`$(v)$
14: $\underline{f}^* \leftarrow$ `CROWN_concretize`$(\mathcal{I}, \mathcal{S})$
15: **Outputs:** $\underline{f}^*$

---

Our method (Algorithm 3) takes as input the graph $\mathcal{G}$ of $f$, the output node $o$ to bound, and a set of early-stop nodes $\mathcal{S} \subset \mathbf{V}$. It outputs the lower bound of the value of $o$, i.e., $\underline{f}^*$. It first performs `CROWN_init` to initialize $d_v$ for all nodes $v$, representing the number of output nodes of $v$ that have not yet been visited.

The algorithm maintains a queue $Q$ of nodes to visit and performs a Breadth First Search (BFS) on $\mathcal{G}$, starting from $o$. When visiting a node $v$, it traverses all input nodes $w$ of $v$, decrementing $d_w$. If all

output nodes of $w$ have been visited and $w$ is not an input node of $\mathcal{G}$, $w$ is added to $Q$ for propagation (Lines 7–10). The key modification occurs in Lines 11–12, where bound propagation from $v$ to all its input nodes is stopped if $v \in \mathcal{S}$.

Finally, the algorithm concretizes the output bound $\underline{f}^*$ at nodes $v \in \mathcal{I} \cup \mathcal{S}$ based on their lower and upper bounds $\mathbf{l}_v$ and $\mathbf{u}_v$. We assume $\mathbf{l}_v$ and $\mathbf{u}_v$ are known for $v \in \mathcal{I}$ since the input range of $\mathbb{N}_{\boldsymbol{u}}$, as well as all constant values and model parameters, is known. For any $v \in \mathcal{S}$, its $\mathbf{l}_v$ and $\mathbf{u}_v$ are given by $\min_m \mathbf{g}_v(\boldsymbol{u}^m)$ and $\max_m \mathbf{g}_v(\boldsymbol{u}^m)$ from *Search-integrated bounding* in Section 3.2.

**An illustrative example for bounding.** Assume $f(\boldsymbol{u})$ is an $L$-layer MLP. We illustrate how to estimate its lower bound $\underline{f}^*$ with early stopping at the last ReLU layer in Figure 8. For simplicity, we denote the output $f(\boldsymbol{u})$, the $L$-th linear layer, the ReLU layer, the $L-1$-th linear layer, and the input $\boldsymbol{u}$ as nodes $\mathbb{N}_f$, $\mathbb{N}_L$, $\mathbb{N}_R$, $\mathbb{N}_{L-1}$, and $\mathbb{N}_{\boldsymbol{u}}$, respectively. Additionally, we denote $f(\boldsymbol{u}) = z^{(L)}(\boldsymbol{u})$ as the output value of node $\mathbb{N}_L$, $\hat{z}^{(L-1)}(\boldsymbol{u})$ as the input of $\mathbb{N}_L$ and output of $\mathbb{N}_R$, and $z^{(L-1)}(\boldsymbol{u})$ as the input of $\mathbb{N}_R$ and output of $\mathbb{N}_{L-1}$.

In **Step 1**, we initialize CROWN and the queue $Q$ for traversal, starting with the output node $\mathbb{N}_f$. In **Step 2**, we update the out-degree of node $\mathbb{N}_L$ which is the input of $\mathbb{N}_f$, and propagate from $\mathbb{N}_f$ to $\mathbb{N}_L$. Since $d_L = 0$ indicates that all its outputs (in this case, only $\mathbb{N}_f$) have been visited, node $\mathbb{N}_L$ is added to $Q$. In **Step 3**, we continue propagation to the input of $\mathbb{N}_L$, which is the node $\mathbb{N}_R$. Then $\mathbb{N}_R$ is added to $Q$. In **Step 4**, we visit $\mathbb{N}_R$, which is defined as an early-stop node. The backward flow stops propagating to its input node $\mathbb{N}_{L-1}$, and $\mathbb{N}_{L-1}$ is not added to $Q$ because it is not an input node. Since $Q$ is now empty, the bound propagation is complete.

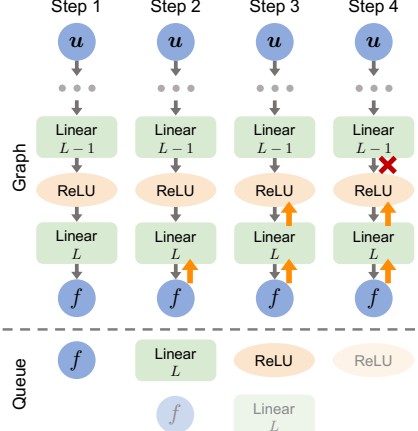

Figure 8: **Bound propagation with early-stop on an $L$-layer MLP $f(\boldsymbol{u})$.** Bound propagation starts from the node of output $f$ and then backwards layer by layer to the $L-1$-th linear layer. The backward flow is highlighted as ⬆ and stop at ✖. In the queue, the node popped at every step is semitransparent.

**Finally**, we require the lower and upper bounds of $\hat{z}^{(L-1)}(\boldsymbol{u})$ (the input value of $\mathbb{N}_R$ and the output value of $\mathbb{N}_{L-1}$) to compute $\underline{f}^*$. Using our *Search-integrated bounding* approach, these bounds are obtained empirically from samples during the searching process.

**A deeper look at the illustrative example.** We now connect the CROWN theorem in Section B.1 to our illustrative example to better understand the behaviors of `CROWN_prop` and `CROWN_concretize`. Here, the input $x$ in Section B.1 corresponds to $\boldsymbol{u}$.

In **Step 2**, since $v = L$ is a linear layer, calling `CROWN_prop` corresponds to the propagation in Eq. 11. Note that no relaxation is introduced when propagating through the linear layer.

In **Step 3**, $v = R$ is a non-linear ReLU layer, and calling `CROWN_prop` corresponds to the propagation in Eq. 13. This step requires a linear estimation of the non-linear layer as described in Eq. 12, which is obtained from the lower and upper bounds of the input to $\mathbb{N}_R$ (i.e., $\hat{z}^{(L-1)}(\boldsymbol{u})$) using Lemma B.1. At this stage, linear relaxation is introduced for the non-linear layer, potentially loosening the final lower bound of $f(\boldsymbol{u})$.

The lower and upper bounds of $\hat{z}^{(L-1)}(\boldsymbol{u})$ are referred to as intermediate layer bounds or *pre-activation bounds* in Section 3.2. However, these bounds are initially unknown in practice. In the original CROWN algorithm, computing these bounds requires recursively calling `compute_bound` with $o = L - 1$. In our approach, these bounds are instead estimated empirically from samples during the searching process, as they serve as the input bounds for the early-stop node $\mathbb{N}_R$.

Now assume we have obtained the intermediate layer bounds and propagated the linear relation through the non-linear node $\mathbb{N}_R$. With the early-stop mechanism, we stop further propagation to $\mathbb{N}_{L-1}$ and subsequently to the input $\mathbb{N}_{\boldsymbol{u}}$. At this point, `CROWN_concretize` is called to compute $\underline{f}^*$ using the intermediate layer bounds and the relaxed linear relation between $\mathbb{N}_R$ and $\mathbb{N}_f$ obtained from propagation. Specifically, this can be achieved by replacing $x$ with $\hat{z}^{(L-1)}(\boldsymbol{u})$ in Theorem B.3.

In contrast, the original CROWN algorithm continues propagating through $N_{L-1}$ and eventually to the input $N_u$, then calls `CROWN_concretize` using the linear relation between $N_u$ and $N_f$ and the lower and upper bounds of $N_u$, as described in Theorem B.3.

**Improvement of our approaches.**    Here, we discuss why our bounding approaches (*Propagation early-stop* and *Search-integrated bounding*) achieve much tighter bound estimations and greater efficiency compared to the original CROWN.

*Efficiency:* The original CROWN performs bound propagation through every layer and recursively computes each intermediate layer bound by propagating it back to the input. This process results in a quadratic time complexity with respect to the number of layers. In contrast, our method conducts bound propagation only from $N_f$ to a few early-stop nodes and derives the input bounds of these nodes from prior sampling-based searching without recursively calling CROWN. As a result, the time complexity of our approach can be linear with respect to the number of layers and even constant under certain configurations of early-stop nodes.

*Effectiveness:* As introduced earlier, the looseness in bound estimation stems from the linear relaxation of non-linear layers. In the original CROWN, the number of linear relaxations is quadratic with respect to the number of non-linear layers. In our approach, the bounding procedure involves far fewer linear relaxations. Furthermore, the empirical bounds obtained from searching, which may slightly underestimate the actual bounds, contribute to further tightening the bound estimation.

## C ADDITIONAL EXPERIMENT RESULTS

### C.1 SCALABILITY ANALYSIS

**Comparison with sampling-based methods.** We conducted an experiment to compare the scalability of our BaB-ND with sampling-based methods on complex planning problems. We used the same model sizes and planning horizons as in Figure 7 (a), optimizing the complex objective function applied in the *Pushing w/ Obstacles* task. Parameters for all methods were adjusted to ensure similar runtimes for the largest problems.

The results in Figure 9 show that the runtime of our BaB-ND is less sensitive to the increasing complexity of planning problems compared to sampling-based methods. While BaB-ND incurs additional overhead from initializing $\alpha,\beta$-CROWN and performing branching and bounding, it is less efficient than sampling-based methods for small problems.

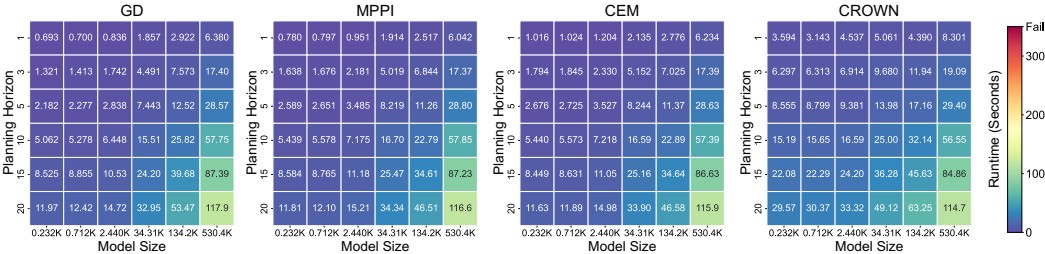

Figure 9: **Comparison of runtime with sampling-based methods.** Although our BaB-ND is less efficient on small planning problems than baselines, it achieves similar efficiency on larger planning problems.

We also report the average objectives for all methods on the largest four planning problems to evaluate their effectiveness in Table 1. Overall, the performance gaps between our BaB-ND and the baselines increase with the size of the problem, highlighting the ineffectiveness of sampling-based methods for large, complex planning problems.

Table 1: Comparison of planning performance across different configurations

| Method | Planning Problem Size | | | |
|---|---|---|---|---|
| | (134.2K,15) | (134.2K,20) | (530.2K,15) | (530.2K,20) |
| GD | 57.2768 | 64.4789 | 54.7078 | 60.2575 |
| MPPI | 47.4451 | 53.7356 | 45.1371 | 45.6338 |
| CEM | 47.0403 | 47.6487 | 43.8235 | 38.8712 |
| Ours | **46.0296** | **46.1938** | **41.6218** | **34.6972** |

Additionally, we evaluate the planning performance of sampling-based methods and our approach on the same simple synthetic planning problems as those in Figure 7. We report only the six cases that MIP can solve optimally within 300 seconds. The results in Table 2 show that, under these much simpler settings compared to those of our main experiments, all methods perform similarly. Sampling-based methods (MPPI, CEM, and ours) achieve a gap under the order of $1 \times 10^{-4}$ compared to MIP with an optimality guarantee.

Table 2: Comparison of planning performance on simple synthetic planning problems

| Method | Planning Problem Size | | | | | |
|---|---|---|---|---|---|---|
| | (0.232K,1) | (0.712K,1) | (2.440K,1) | (0.232K,3) | (0.712K,3) | (0.232K,5) |
| MIP | 30.3592 | 32.9750 | 33.5496 | 22.1539 | 28.0832 | 15.6069 |
| GD | 30.3622 | 32.9750 | 33.5496 | 22.3242 | 28.1404 | 17.0681 |
| MPPI | 30.3592 | 32.9750 | 33.5496 | 22.1539 | 28.0832 | 15.6069 |
| CEM | 30.3592 | 32.9750 | 33.5496 | 22.1539 | 28.0832 | 15.6069 |
| Ours | 30.3592 | 32.9750 | 33.5496 | 22.1539 | 28.0832 | 15.6069 |

**Comparison with CPU version.** We evaluate the performance improvement from CPU to GPU in Figure 10. We use the same test cases as in Figure 7 and report "NaN" if the process does not terminate within 300 seconds.

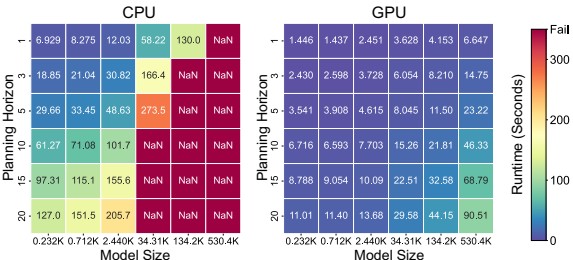

The results clearly demonstrate that our implementation benefits significantly from GPU acceleration, achieving over 10x speedup compared to the CPU version, even for small planning problems.

Figure 10: **Comparison of runtime on CPU and GPU.** GPU acceleration improves the scalability of BaB-ND much.

## C.2    COMPARISON WITH CONVENTIONAL MOTION PLANNING APPROACHES

We conduct an additional experiment on the task Pushing with Obstacle to compare the planning performance of our sampling-based baselines, our BaB-ND, and two conventional motion planning approaches: 1. Rapidly-exploring Random Tree (RRT); 2. Probabilistic Roadmap (PRM). In Table 3. Since RRT and PRM do not optimize the objective as we did in sampling-based methods and our BaB-ND, we only report the step cost at planning horizon $H$ as the final step cost instead of the planning objective.

Table 3: Comparison of planning performance with RRT and PRM

|  | **GD** | **MPPI** | **CEM** | **RRT** | **PRM** | **Ours** |
|---|---|---|---|---|---|---|
| Final step cost ($\downarrow$) | 4.1238 | 1.5082 | 1.0427 | 10.6472 | 1.6784 | **0.2339** |

The results demonstrate that our method significantly outperforms all other approaches. Implementation details for RRT and PRM have been included in Appendix D. The main reasons for the performance gap are as follows: 1. The search space in our task is complex and continuous, making it challenging for discrete sampling methods like RRT and PRM to achieve effective coverage. 2. These methods are prone to getting stuck on obstacles, often failing to reach the target state.

## C.3    ABLATION STUDY AND HYPER-PARAMETER ANALYSIS

**Ablation Study.** We conduct an additional ablation study on the *Pushing w/ Obstacles* and *Object Sorting* tasks to evaluate how different design choices impact planning performance in Table 4.

Table 4: Ablation study on branching and bounding components

| **(a) Heuristics for Selecting subdomains to Split** | | |
|---|---|---|
|  | $\underline{f}^*_{\mathcal{C}_i}$ and $\overline{f}^*_{\mathcal{C}_i}$ | $\overline{f}^*_{\mathcal{C}_i}$ only | $\underline{f}^*_{\mathcal{C}_i}$ only |
| Pushing w/ Obstacles | **31.9839** | 32.2777 | 32.6112 |
| Object Sorting | **31.0482** | 32.1249 | 33.2462 |
| **(b) Heuristics for Splitting subdomains** | | |
|  | $(\overline{\boldsymbol{u}}_j - \underline{\boldsymbol{u}}_j) \cdot \|n_j^{\text{lo}} - n_j^{\text{up}}\|$ | $(\overline{\boldsymbol{u}}_j - \underline{\boldsymbol{u}}_j)$ | $\|n_j^{\text{lo}} - n_j^{\text{up}}\|$ |
| Pushing w/ Obstacles | **31.9839** | 32.3869 | 32.6989 |
| Object Sorting | **31.0482** | 34.5114 | 32.8438 |
| **(c) Bounding Component** | | |
|  | Ours | Zero $\underline{f}^*_{\mathcal{C}_i}$ | Zero $\underline{f}^*_{\mathcal{C}_i} + f^*_{\mathcal{C}_i}$ only |
| Pushing w/ Obstacles | **31.9839** | 32.3419 | 34.6227 |
| Object Sorting | **31.0482** | 33.6110 | 34.4535 |

*(a) Heuristics for selecting subdomains to split:* **1.** Select based on both lower and upper bounds $\underline{f}^*_{\mathcal{C}_i}$ and $\overline{f}^*_{\mathcal{C}_i}$. **2.** Select based only on $\overline{f}^*_{\mathcal{C}_i}$. **3.** Select based only on $\underline{f}^*_{\mathcal{C}_i}$. Among these heuristics, selecting

promising subdomains based on both $\underline{f}^*_{\mathcal{C}_i}$ and $\overline{f}^*_{\mathcal{C}_i}$ achieves better planning performance by balancing exploitation and exploration effectively compared to the other strategies.

*(b) Heuristics for splitting subdomains:* **1.** Split based on the largest $(\overline{u}_j - \underline{u}_j) \cdot |n^{\text{lo}}_j - n^{\text{up}}_j|$. **2.** Split based on the largest $(\overline{u}_j - \underline{u}_j)$. **3.** Split based on the largest $|n^{\text{lo}}_j - n^{\text{up}}_j|$. Our heuristic demonstrates superior planning performance by effectively identifying important input dimensions to split.

*(c) Bounding components:* **1.** Use our bounding approach with propagation early-stop and search-integrated bounding. **2.** Use constant zero as trivial lower bounds to disable the bounding component. **3.** Disable both the bounding component and the heuristic for selecting subdomains to split. Our bounding component improves planning performance by obtaining tight bound estimations, helping prune unpromising subdomains to reduce the search space, and prioritizing promising subdomains for searching.

**Hyper-parameter Analysis.** We adjust three hyper-parameters in BaB-ND for the tasks *Pushing w/ Obstacles* and *Object Sorting* to evaluate its hyper-parameter sensitivity:

• $\eta = \frac{n_1}{n} \in [0, 1]$, the ratio of the number of subdomains picked with the best upper bounds ($n_1$) to the number of all picked subdomains ($n$) in the heuristic used for selecting subdomains to split. A larger $\eta$ promotes exploitation, while a smaller $\eta$ encourages exploration.

• $T \in \mathbb{R}$, the temperature of $\mathrm{softmax}$ sampling in the heuristic for subdomain selection. A larger $T$ results in more uniform and random sampling, whereas a smaller $T$ leads to more deterministic selection of subdomains with smaller lower bounds.

• $w \in (0, 100]$, the percentage of top samples used in the heuristic for splitting subdomains. A larger $w$ results in more conservative decisions by considering more samples, while a smaller $w$ leads to more aggressive splitting.

We report the mean objectives under different hyper-parameter configurations in Table 5. The base hyper-parameter configuration is $\eta = 0.75$, $T = 0.05$, and $w = 1$. For benchmarking, we vary at most one hyper-parameter at a time while keeping the others fixed at the base configuration.

Table 5: Planning performance under different hyper-parameter configurations

**(a) hyper-parameter $\eta$**

|  | $\eta = 0.25$ | $\eta = 0.50$ | $\eta = 0.75$ |
|---|---|---|---|
| Pushing w/ Obstacles | **31.8574** | 31.9828 | 31.9839 |
| Object Sorting | **30.1760** | 30.2795 | 31.0482 |

**(b) hyper-parameter $T$**

|  | $T = 0.05$ | $T = 1$ | $T = 20$ |
|---|---|---|---|
| Pushing w/ Obstacles | **31.9839** | 32.3990 | 32.1267 |
| Object Sorting | **31.0482** | 31.2366 | 31.8263 |

**(c) hyper-parameter $w$**

|  | $w = 0.1$ | $w = 1$ | $w = 10$ |
|---|---|---|---|
| Pushing w/ Obstacles | 32.0068 | **31.9839** | 32.0599 |
| Object Sorting | **30.5953** | 31.0482 | 31.1545 |

The results show that different hyper-parameter configurations produce slight variations in objectives, but the gaps are relatively small. This indicates that our BaB-ND is not highly sensitive to these hyper-parameters. Consequently, it is feasible in practice to use a fixed hyper-parameter configuration that delivers reasonable performance across different test cases and tasks.

## C.4 QUANTITATIVE ANALYSIS ON SEARCH SPACE

We conducted an experiment to measure the normalized space size of pruned subdomains over iterations. In Table 6, we report three metrics over the branch-and-bound iterations: **1.** the normalized space size of pruned subdomains, **2.** the size of the selected subdomains, and **3.** the improvement in the objective value.

With increasing iterations, the average and best total space size of pruned subdomains increases rapidly and then converges, demonstrating the effectiveness of our bounding methods. Once the pruned space size reaches a plateau, the total space size of selected promising subdomains continues to decrease, indicating that the estimated lower bounds remain effective in identifying promising subdomains. The decreasing objective over iterations further confirms that BaB-ND focuses on the most promising subdomains, reducing space size to the magnitude of $1 \times 10^{-4}$.

Table 6: Performance Metrics Over Iterations

| Metric | Iterations | | | | | |
|---|---|---|---|---|---|---|
| | 0 | 4 | 8 | 12 | 16 | 20 |
| Pruned space size (Avg, ↑) | 0.0000 | 0.7000 | 0.8623 | 0.8725 | 0.8744 | 0.8749 |
| Pruned space size (Best, ↑) | 0.0000 | 0.8750 | 0.9921 | 0.9951 | 0.9951 | 0.9952 |
| Selected space size (Avg, ↓) | 1.0000 | 0.3000 | 0.0412 | 0.0048 | 0.0005 | 0.0003 |
| Best objective (Avg, ↓) | 41.1222 | 36.0511 | 35.5091 | 34.8024 | 33.8991 | 33.3265 |

## C.5 Performance Change with Varying Input Discontinuities

We conducted a follow-up experiment by removing the obstacles (non-feasible regions) in the problem of *Pushing w/ Obstacles*, simplifying the objective function. Below, we report the performance of different methods on the simplified objective function (w/o obstacles) and the original objective function (w/ obstacles) in Table 7.

The results show that in simple cases, although our BaB-ND consistently outperforms baselines, MPPI and CEM provide competitive performance. In contrast, in complex cases, BaB-ND significantly outperforms the baselines, demonstrating its effectiveness in handling discontinuities and constraints.

Table 7: Performance comparison varying input discontinuities

| (a) Objective **w/o obstacles** | | | | |
|---|---|---|---|---|
| | (134.2K,15) | (134.2K,20) | (530.2K,15) | (530.2K,20) |
| GD | 64.5308 | 64.2956 | 63.0130 | 60.6300 |
| MPPI | 34.4295 | 26.9970 | 33.8077 | 26.1204 |
| CEM | 34.3864 | 26.7688 | 33.6669 | 25.9599 |
| Ours | **34.2347** | **26.4841** | **33.6144** | **25.6603** |
| (b) Objective **w/ obstacles** (Table 1) | | | | |
| | (134.2K,15) | (134.2K,20) | (530.2K,15) | (530.2K,20) |
| GD | 57.2768 | 64.4789 | 54.7078 | 60.2575 |
| MPPI | 47.4451 | 53.7356 | 45.1371 | 45.6338 |
| CEM | 47.0403 | 47.6487 | 43.8235 | 38.8712 |
| Ours | **46.0296** | **46.1938** | **41.6218** | **34.6972** |

## C.6 Further Scalability Analysis on the Synthetic Example

We extend our experiment on the synthetic example shown in Figure 4, as this allows us to easily scale up the input dimension while knowing the optimal objectives. We vary the input dimension $N$ from 50 to 300 and compare our BaB-ND with MPPI and CEM.

Although this synthetic example is simpler than practical cases, it provides valuable insights into the expected computational cost and solution quality as we scale to high-dimensional problems. It demonstrates the potential of BaB-ND in handling complex scenarios such as 3D tasks. We report the gaps between the best objective found by different baseline methods and the optimal objective value below.

The results in Table 8 show that our BaB-ND much outperforms baselines when the input dimension increases. These results are expected since existing sampling-based methods search for solutions across the entire input space, requiring an exponentially increasing number of samples to achieve

sufficient coverage. In contrast, our BaB-ND strategically splits and prunes unpromising regions of the input space, guiding and improving the effectiveness of existing sampling-based methods.

Table 8: Performance comparison across different input dimensions (Metric: Gap to $f^*$, $\downarrow$)

| Method | Input dimension $N$ | | | | | |
|---|---|---|---|---|---|---|
| | 50 | 100 | 150 | 200 | 250 | 300 |
| MPPI | 7.4467 | 45.1795 | 105.1584 | 181.1274 | 259.1044 | 357.3273 |
| CEM | 5.1569 | 15.6328 | 26.3735 | 39.3862 | 61.6739 | 92.4286 |
| Ours | **0.0727** | **0.2345** | **0.4210** | **0.6976** | **1.2824** | **1.7992** |

We further report the following metrics about our BaB-ND in Table 9 to better understand the behavior of BaB-ND under high-dimensional cases: **1.** The gap between the best objective found and the optimal objective value as above, **2.** The normalized space size of pruned subdomains at the last iteration, **3.** The normalized space size of selected subdomains at the last iteration, and **4.** The total runtime.

The results demonstrate that our BaB-ND effectively focuses on small regions to search for better objectives, while the runtime increases approximately linearly with input dimension under GPU acceleration.

Table 9: Performance metrics across different input dimensions $N$

| Metric | Input Dimensions $N$ | | | | | |
|---|---|---|---|---|---|---|
| | 50 | 100 | 150 | 200 | 250 | 300 |
| Gap to $f^*$ ($\downarrow$) | 0.0727 | 0.2345 | 0.4210 | 0.6976 | 1.2824 | 1.7992 |
| Selected Space Size ($\downarrow$) | 0.0002 | 0.0017 | 0.0026 | 0.0042 | 0.0064 | 0.0040 |
| Pruned Space Size ($\uparrow$) | 0.8515 | 0.6073 | 0.3543 | 0.1762 | 0.0579 | 0.0113 |
| Runtime ($\downarrow$) | 4.2239 | 6.5880 | 9.5357 | 11.6504 | 13.7430 | 15.8053 |

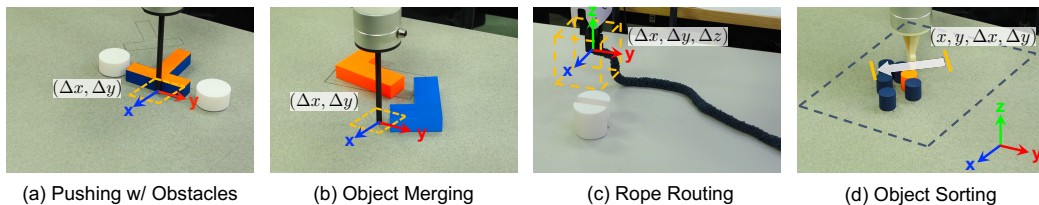

|                           |                        |                   |                    |
| (a) Pushing w/ Obstacles  | (b) Object Merging     | (c) Rope Routing  | (d) Object Sorting |

Figure 11: **Visualization of real-world experiments with illustration of action spaces in different tasks.** In tasks *Pushing w/ Obstacles* and *Object Merging*, the actions are defined as the movement of the pusher in 2D space. In the task *Rope Routing*, the action is defined as the movement of the gripper in 3D space. In the task *Object Sorting*, the 4 DoF action is defined as the 2D starting position and 2D movement of the pusher.

## D  EXPERIMENT DETAILS

### D.1  DEFINITION OF ACTIONS AND STATES

We consider actions with 2–4 degrees of freedom (DoF) in Cartesian coordinates for our tasks. Across all tasks, we adopt a general design principle of using 3D points as the state representation, as they effectively capture both geometric and semantic information and facilitate seamless transfer between simulation and the real world. The specific definitions of actions and states for each task are detailed below:

**Pushing w/ Obstacles.**    The action is defined as a 2D movement of the pusher, $(\Delta x, \Delta y)$, within a single step, as illustrated in Figure 11.a. Four keypoints are assigned to the "T"-shaped object to capture its geometric features, as visualized in Figure 12.a.

**Object Merging.**    The action is similarly defined as a 2D movement of the pusher, $(\Delta x, \Delta y)$, within one step, as shown in Figure 11.b. The orientation of the pusher rotates around the gravity direction and is always perpendicular to the direction of pushing $(\Delta x, \Delta y)$. The state is represented by six keypoints on the "L"-shaped objects (three keypoints per object), as depicted in Figure 12.b.

**Rope Routing.**    This task involves 3 DoF actions, defined as movements of the robot gripper along the Cartesian axes, $(\Delta x, \Delta y, \Delta z)$, as illustrated in Figure 11.c. To better represent the rope's geometric configuration, multiple keypoints (e.g., 10) are sampled along the rope, as shown in Figure 12.c.

**Object Sorting.**    This task requires a series of independent long-range pushing actions, where the initial position of the pusher for each step is independent of its final position from the previous step. The action is defined by the 2D initial position of the pusher, $(x, y)$, and its subsequent 2D movement, $(\Delta x, \Delta y)$, as visualized in Figure 11.d. For the state, we use an object-centric representation, defining the state as the center points of all objects, as illustrated in Figure 12.d. Interactions between objects are modeled as a graph using graph neural networks.

For all tasks, we do not explicitly specify the contact points between the robot and objects. Instead, our BaB-ND framework generates a sequence of end-effector positions for the robot to follow, implicitly determining aspects such as which side of the "T"-shaped object is being pushed.

### D.2  DATA COLLECTION

For training the dynamics model, we randomly collect interaction data from simulators. For *Pushing w/ Obstacles*, *Object Merging*, and *Object Sorting* tasks, we use Pymunk (Blomqvist, 2022) to collect data, and for the *Rope Routing* task, we use NVIDIA FleX (Macklin et al., 2014) to generate data. In the following paragraphs, we introduce the data generation process for different tasks in detail.

**Pushing w/ Obstacles.**    As shown in Figure 12.a, the pusher is simulated as a 5mm cylinder. The stem of the "T"-shaped object has a length of 90mm and a width of 30mm, while the bar has a length of 120mm and a width of 30mm. The pushing action along the x-y axis is limited to 30mm. We don't add explicit obstacles in the data generation process, while the obstacles are added as penalty terms during planning. We generated 32,000 episodes, each containing 30 pushing actions between the pusher and the "T"-shaped object.

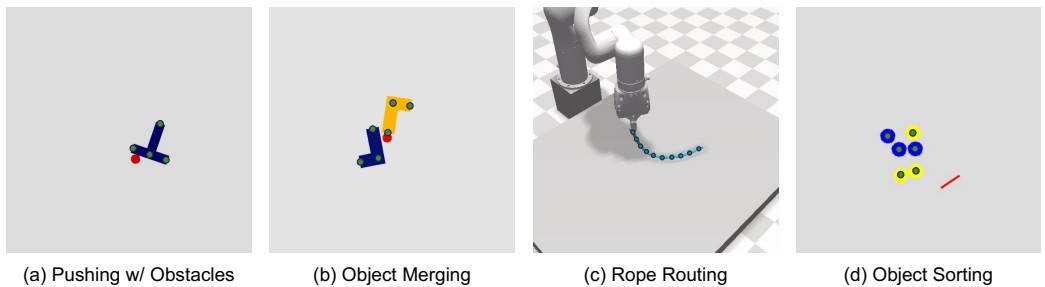

(a) Pushing w/ Obstacles      (b) Object Merging      (c) Rope Routing      (d) Object Sorting

Figure 12: **Simulation environments used for data collection.** We use Pymunk to simulate environments involving only rigid body interactions. For manipulating the deformable rope, we utilize NVIDIA FleX to simulate the interactions between the rope and the robot gripper.

**Object Merging.** As shown in Figure 12.b, the pusher is simulated as a 5mm cylinder. The leg of the "L"-shaped object has a length of 30mm and a width of 30mm, while the foot has a length of 90mm and a width of 30mm. The pushing action along the x-y axis is limited to 30mm. We generated 64,000 episodes, each containing 40 pushing actions between the pusher and the two "L"-shaped objects.

**Rope Routing.** As shown in Figure 12.c, we use an xArm6 robot with a gripper to interact with the rope. The rope has a length of 30cm and a radius of 0.03cm. One end of the rope is fixed while the gripper grasps the other end. We randomly sample actions in 3D space, with the action bound set to 30cm. The constraint is that the distance between the gripper position and the fixed end of the rope cannot exceed the rope length. We generated 15,000 episodes, each containing 6 random actions. For this task, we post-process the dataset and split each action into 2cm sections.

**Object Sorting.** As shown in Figure 12.d, the pusher is simulated as a rectangle measuring 45mm by 3.5mm. The radius of the object pieces is set to 15mm. For this task, we use long push as our action representation, which generates the start position and pushing action length along the x-y axis. The pushing action length is bounded between -100mm and 100mm. We generated 32,000 episodes, each containing 12 pushing actions between the pusher and the object pieces.

### D.3 DETAILS OF NEURAL DYNAMICS MODEL LEARNING

We learn the neural dynamics model from the state-action pairs collected from interactions with the environment. Let the state and action at time $t$ be denoted as $x_t$ and $u_t$. Our goal is to learn a predictive model $f_{\text{dyn}}$, instantiated as a neural network, that takes a short sequence of states and actions with $l$-step history and predicts the next state at time $t + 1$:

$$\hat{x}_{t+1} = f_{\text{dyn}}(x_t, u_t). \tag{27}$$

To train the dynamics model for better long-term prediction, we iteratively predict future states over a time horizon $T_h$ and optimize the neural network parameters by minimizing the mean squared error (MSE) between the predictions and the ground truth future states:

$$\mathcal{L} = \frac{1}{T_h} \sum_{t=l+1}^{l+T_h} \|x_{t+1} - f_{\text{dyn}}(\hat{x}_t, u_t)\|_2^2. \tag{28}$$

For different tasks, we choose different types of model architecture and design different inputs and outputs. For *Pushing w/ Obstacles*, *Object Merging*, and *Rope Routing* tasks, we use an MLP as our dynamics model. For the *Object Sorting* task, we utilize a GNN as the dynamics model, since the pieces are naturally modeled as a graph. Below is the detailed information for each task.

**Pushing w/ Obstacles.** We use a four-layer MLP with [128, 256, 256, 128] neurons in each respective layer. The model is trained with an Adam optimizer for 7 epochs, using a learning rate of 0.001. A cosine learning rate scheduler is applied to regularize the learning rate. For the model input, we select four keypoints on the object and calculate their relative coordinates to the current pusher position. These coordinates are concatenated with the current pusher action (resulting in an input dimension of 10) and input into the model. For the loss function, given the current state and action sequence, the model predicts the next 6 states, and we compute the MSE loss with the ground truth.

**Object Merging.** We use the same architecture, optimizer, training epochs, and learning rate scheduler as in the *Pushing w/ Obstacles* setup. For the model input, we select three keypoints for each object and calculate their relative coordinates to the current pusher positions. These coordinates are then concatenated with the current pusher action (resulting in a state dimension of 12) and input into the model. We also use the same loss function as in the *Pushing w/ Obstacles* setup.

**Rope Routing.** We use a two-layer MLP with 128 neurons in each layer. The model is trained with an Adam optimizer for 50 epochs, with a learning rate of 0.001, and a cosine learning rate scheduler to adjust the learning rate. For the model input, we use farthest point sampling to select 10 points on the rope, reordered from closest to farthest from the gripper. We then calculate their relative coordinates to both the current and next gripper positions, concatenate these coordinates, and input them into the model. For the loss function, given the current state and action sequence, the model predicts the next 8 states, and we compute the MSE loss with the ground truth.

**Object Sorting.** We use the same architecture as DPI-Net (Li et al., 2018). The model is trained with an Adam optimizer for 15 epochs, with a learning rate of 0.001, and a cosine learning rate scheduler to adjust the learning rate. For the model input, we construct a fully connected graph neural network using the center position of each piece. We then calculate their relative coordinates to the current and next pusher positions. These coordinates are concatenated as the node embedding and input into the model. For the loss function, given the current state and action sequence, the model predicts the next 6 states, and we compute the MSE loss with the ground truth.

### D.4 Definition of Cost Functions

In this section, we introduce our cost functions for model-based planning Eq. 1 across different tasks. For every task, we assume the initial and target states $x_0$ and $x_{\text{target}}$ are given. We denote the position of the end-effector at time $t$ as $p_t$. In tasks involving continuous actions like *Pushing w/ Obstacles*, *Object Merging*, and *Rope Routing*, the action $u_t$ is defined as the movement of the end-effector, $p_t = p_{t-1} + u_t$, and $p_0$ is given by the initial configuration. In the task of *Object Sorting* involving discrete pushing, $p_t$ is given by the action $a_i$ as the pusher position before pushing. In settings with obstacles, we set the set of obstacles as $O$. Each $o \in O$ has its associated static position and size as $p_o$ and $s_o$. Our cost functions are designed to handle discontinuities and constraints introduced by obstacles, and BaB-ND can work effectively on these complex cost functions.

**Pushing w/ Obstacles.** As introduced before, we formalize the obstacles as penalty terms rather than explicitly introducing them in the dynamics model. Our cost function is defined by a cost to the goal position plus a penalty cost indicating whether the object or pusher collides with the obstacle. The detailed cost is listed in Eq. 29.

$$c_t = c(x_t, u_t) = w_t \left\| x_t - x_{\text{target}} \right\|$$
$$+ \lambda \sum_{o \in O} \left( \text{ReLU}(s_o - \left\| p_t - p_o \right\|) + \text{ReLU}(s_o - \left\| x_t - p_o \right\|) \right) \tag{29}$$

where $\left\| x_t - x_{\text{target}} \right\|$ gives the difference between the state at time $t$ and the target. $\left\| p_t - p_o \right\|$ and $\left\| x_t - p_o \right\|$ give the distances between the obstacle $o$ and the end-effector and the object, respectively. Two ReLU items yield positive values (penalties) when the pusher or object is located within the obstacle $o$. $w_t$ is a weight increasing with time $t$ to encourage alignment to the target. $\lambda$ is a large constant value to avoid any collision. In implementation, $x_t$ is a concatenation of positions of keypoints, and $\left\| x_t - p_o \right\|$ is calculated keypoint-wise. Ideally, $c_T$ can be optimized to 0 by a strong planner with the proper problem configuration.

**Object Merging.** In this task requiring long-horizon planning to manipulate two objects, we do not set obstacles and only consider the difference between the state at every time step and the target. The cost is shown in Eq. 30.

$$c_t = w_t \left\| x_t - x_{\text{target}} \right\| \tag{30}$$

**Rope Routing.** In this task involving a deformable rope, we sample keypoints using Farthest Point Sampling (FPS). $x_{\text{target}}$ is defined as the target positions of sampled keypoints. The cost is defined in Eq. 31, similar to the one in *Pushing w/ Obstacles*. Here, two obstacles are introduced to form the tight-fitting slot. In implementation, naively applying such a cost does not always achieve our

target of routing the rope into the slot since a trajectory greedily translating in the $z$-direction without lifting may achieve optimality. Hence, we modify the formulation by assigning different weights for different directions $(x, y, z)$ when calculating $\|x_t - x_{\text{target}}\|$ to ensure the desirable trajectory yields the lowest cost.

$$c_t = w_t \|x_t - x_{\text{target}}\| + \lambda \sum_{o \in O} \left( \text{ReLU}(s_o - \|p_t - p_o\|) + \text{ReLU}(s_o - \|x_t - p_o\|) \right) \quad (31)$$

**Object Sorting.** In this task, a pusher interacts with a cluster of object pieces belonging to different classes. We set $x_{\text{target}}$ as the target position for every class. Additionally, for safety concerns to prevent the pusher from pressing on the object pieces, we introduce obstacles defined as the object pieces in the cost as Eq. 32. For each object piece $o$, its size $s_o$ is set as larger than the actual size in the cost, and its position $p_o$ is given by $x_t$. The definition of the penalty is similar to that in *Pushing w/ Obstacles*.

$$c_t = w_t \|x_t - x_{\text{target}}\| + \lambda \sum_{o \in O} \text{ReLU}(s_o - \|p_t - p_o\|) \quad (32)$$

### D.5 DETAILS OF REAL WORLD DEPLOYMENT

We have four cameras observing the environment from the corners of the workspace. We implemented task-specific perception modules to determine the object states from the multi-view RGB-D images.

**Pushing w/ Obstacles and Object Merging.** We use a two-level planning framework in these two tasks, involving both long-horizon and short-horizon planning. First, given the initial state $(s_0)$ and pusher position, we perform long-horizon open-loop planning to obtain a reference trajectory $(s_0, a_0, s_1, a_1, \ldots, s_N)$. Next, an MPPI planner is used as a local controller to efficiently track this trajectory. Since the local planning horizon is relatively short, the local controller operates at a higher frequency. In the local planning phase, the reference trajectory is treated as a queue of subgoals. Initially, we set $s_1$ as the subgoal and use the local controller to plan a local trajectory. Once $s_1$ is reached, $s_2$ is set as the next subgoal. By iterating this process, we ultimately reach the final goal state.

For perception, we filter the point cloud based on color from four cameras, and align it with iterative closest point (ICP) to determine the object states.

**Rope Routing.** For the *Rope Routing* task, we observe that the sim-to-real gap is relatively small. Therefore, the long-horizon planned trajectory is executed directly in an open-loop manner.

For perception, we begin by using Grounding DINO (Liu et al., 2023a) and Segment Anything Model (SAM) (Kirillov et al., 2023) to generate the mask for the rope and extract its corresponding point cloud. Subsequently, we apply farthest point sampling to identify 10 key points on the rope, representing its object state.

**Object Sorting.** There are relatively large observation changes after each pushing action. This creates a noticeable sim-to-real gap for the planned long-horizon trajectory. As a result, we replan the trajectory after each action.

For perception, we filter the point cloud based on color from four cameras and use K-means clustering to separate different object pieces.

### D.6 HYPERPARAMETERS OF BASELINES

We follow previous works (Li et al., 2018; 2019) to implement gradient descent (GD) for planning and enhance it through hyperparameter tuning of the step size. Specifically, we define a base step size and multiply it by varying ratios to launch independent instances with different step sizes. The best result among all instances is recorded for performance comparison. Similarly, we adapt MPPI (Williams et al., 2017) by tuning its noise level and reward temperature. We also implement the Decentralized Cross-Entropy Method (CEM) (Zhang et al., 2022c), which enhances CEM using an ensemble of independent instances. The specific hyperparameter settings for these methods are provided in Table 10.

Table 10: Hyperparameter setting for baselines

**(a) Hyperparameter setting for GD**

| Hyperparameters | Pushing w/ Obstacles | Object Merging | Rope Routing | Object Sorting |
|---|---|---|---|---|
| Number of samples | 320000 | 160000 | 50000 | 24000 |
| Number of iterations | 16 | 18 | 16 | 15 |
| Base step size | 0.01 | 0.01 | 0.01 | 0.01 |
| Step size ratios | [0.05, 0.1, 0.25, 0.5, 0.75, 1, 1.5, 2, 5, 10] | | | |

**(b) Hyperparameter setting for MPPI**

| Hyperparameters | Pushing w/ Obstacles | Object Merging | Rope Routing | Object Sorting |
|---|---|---|---|---|
| Number of samples | 320000 | 160000 | 64000 | 32000 |
| Number of iterations | 20 | 22 | 50 | 18 |
| Base temperature | 20 | 20 | 20 | 50 |
| Temperature ratios | [0.1, 0.5, 1, 1.5, 2] | [0.1, 0.5, 1, 1.5, 2] | [0.1, 0.5, 1, 5] | [0.1, 1, 5] |
| Base noise std | 0.15 | 0.15 | 0.12 | 0.3 |
| Noise ratios | [0.1, 0.5, 1, 1.5, 2] | [0.1, 0.5, 1, 1.5, 2] | [0.1, 0.5, 1, 2, 5] | [0.5, 1, 1.5] |

**(c) Hyperparameter setting for CEM**

| Hyperparameters | Pushing w/ Obstacles | Object Merging | Rope Routing | Object Sorting |
|---|---|---|---|---|
| Number of samples | 320000 | 160000 | 64000 | 32000 |
| Number of iterations | 20 | 55 | 50 | 16 |
| Jitter item | 0.001 | 0.0005 | 0.001 | 0.0005 |
| Number of elites | 10 | 10 | 10 | 10 |
| Number of agents | 10 | 10 | 10 | 10 |

## D.7 IMPLEMENTATION DETAILS OF CONVENTIONAL MOTION PLANNING APPROACHES

**RRT.** As shown in Algorithm 4, in each step, we sample a target state and find the nearest node in the RRT tree. We sample 1000 actions and use the dynamics model to predict 1000 future states. We select the state that is closest to the sampled target and does not collide with obstacles, then add it to the tree. We allow it to plan for 60 seconds, during which it can expand a tree with about 4000 nodes ($N_{max} = 4000$). To avoid getting stuck in local minima, we randomly sample target states 50% of the time, and for the other 50%, we select the goal state as the target.

**PRM.** As shown in Algorithm 5, the PRM construction algorithm generates a probabilistic roadmap by sampling $N$ pairs of object states and pusher positions from the search space, adding these pairs as nodes, and connecting nodes within a defined threshold $\delta$. Here, we set $N = 100K$ and $\delta = 0.15$. The roadmap is represented as a graph $G = (V, E)$, where $V$ includes the sampled nodes, and $E$ contains edges representing feasible connections. The planning over PRM Algorithm 6 uses this roadmap to find a path from the initial state to the goal state. It first integrates the initial state into the graph and connects it to nearby nodes, removes any nodes and edges colliding with obstacles, and applies A* search to find an optimal path.

---

**Algorithm 4** Rapidly-Exploring Random Tree (RRT)

---

1: **Input:** Initial state $x_0$, goal state $x_{\text{goal}}$, search space $\mathcal{X}$ ($\mathcal{X}$ is object state space), maximum iterations $N_{\text{max}}$, action upper and lower bounds $\{\overline{u}, \underline{u}\}$, threshold $\delta$
2: **Output:** A path from $x_0$ to $x_{\text{goal}}$ or failure
3: Initialize tree $T$ with root node $x_0$
4: **for** $i = 1$ to $N_{\text{max}}$ **do**
5:     Sample a random state $x_{\text{rand}}$ from $\mathcal{X}$
6:     Find the nearest node $x_{\text{near}}$ in $T$ to $x_{\text{rand}}$
7:     Sample 1000 actions within $\{\overline{u}, \underline{u}\}$ as a set $\mathcal{U}$, and compute the corresponding next states $\mathcal{X}_{\text{new}} = f_{\text{dyn}}(x_{\text{near}}, \mathcal{U})$
8:     Select the nearest, collision-free next state from $\mathcal{X}_{\text{new}}$ as $x_{\text{new}}$
9:     Add $x_{\text{new}}$ to $T$ with an edge from $x_{\text{near}}$
10:     **if** $x_{\text{new}}$ is within $\delta$ of $x_{\text{goal}}$ **then**
11:         Add $x_{\text{goal}}$ to $T$ with an edge from $x_{\text{new}}$
12:         **return** Path from $x_0$ to $x_{\text{goal}}$ in $T$
13: **return** Failure (No valid path found within $N_{\text{max}}$ iterations)

---

**Algorithm 5** Probabilistic Roadmap (PRM) Construction

---

1: **Input:** Search space $\{\mathcal{X}, \mathcal{P}\}$ ($\mathcal{X}$: object state space, $\mathcal{P}$: pusher position space), number of nodes $N$, connection threshold $\delta$
2: **Output:** A constructed PRM $G = (V, E)$
3: Initialize the roadmap $G = (V, E)$ with $V = \emptyset$ and $E = \emptyset$
4: Randomly sample $N$ pairs $(x, p)$ from $\{\mathcal{X}, \mathcal{P}\}$
5: Add the sampled pairs as nodes in $G$: $V = \{v_i \mid i \leq N\}$, where $v_i = (x_i, p_i)$
6: **for** $i = 1$ to $N$ **do**
7:     **for** $j = 1$ to $N$ **do**
8:         **if** $i \neq j$ **then**
9:             Compute action $u = p_j - p_i$
10:             Predict the next state $x_{\text{new}} = f_{\text{dyn}}(x_i, u)$
11:             **if** distance$(x_{\text{new}}, x_j) < \delta$ **then**
12:                 Add an edge $e_{ij} = \{v_i \rightarrow v_j\}$ to $E$
13: **return** $G = (V, E)$

---

**Algorithm 6** Planning over PRM

---

1: **Input:** Initial state $x_{\text{init}}$, initial pusher position $p_{\text{init}}$, goal state $x_{\text{goal}}$, obstacle space $\{\mathcal{X}_{\text{obs}}, \mathcal{P}_{\text{obs}}\}$, constructed PRM $G = (V, E)$, connection threshold $\delta$
2: **Output:** A path from $x_{\text{init}}$ to $x_{\text{goal}}$ or failure
3: Add $(x_{\text{init}}, p_{\text{init}})$ to $V$
4: **for** each node $v_j \in V$ **do**
5:     Compute action $u = p_j - p_{\text{init}}$
6:     Predict the next state $x_{\text{new}} = f_{\text{dyn}}(x_{\text{init}}, u)$
7:     **if** distance$(x_{\text{new}}, x_j) < \delta$ **then**
8:         Add an edge $\{v_{\text{init}} \rightarrow v_j\}$ to $E$
9: Remove all edges and nodes in $G$ that collide with obstacles $\{\mathcal{X}_{\text{obs}}, \mathcal{P}_{\text{obs}}\}$
10: Use the A* search algorithm to find a path from $v_{\text{init}} = (x_{\text{init}}, p_{\text{init}})$ to $x_{\text{goal}}$ in $G$
11: Identify the node $v_{\text{nearest}}$ closest to $x_{\text{goal}}$
12: Extract the path from $v_{\text{init}}$ to $v_{\text{nearest}}$
13: **return** the extracted path

---

