# OpenReview forum: "BaB-ND: Long-Horizon Motion Planning with Branch-and-Bound and Neural Dynamics"
_ICLR.cc/2025/Conference — ICLR 2025 Poster_

### Official Review · Reviewer_U2wt · 2024-11-02

**Soundness:** 3
**Presentation:** 3
**Contribution:** 2
**Rating:** 5
**Confidence:** 4

**Summary:**

This work introduces a divide-and-conquer approach for planning long-horizon manipulation trajectories. The method relies on a dynamics function trained on simulated data and operates through branching, bounding, and searching.

**Strengths:**

The writing is clear and well-structured, and the appendix contents are helpful.

The concepts and formulation are well-defined, and the 1D toy example effectively validates the idea.

The method includes extensive experiments, both simulated and real-world, with comparisons to several baseline methods.

**Weaknesses:**

The specific novelty and distinctions of the proposed method compared to neural network verifiers are unclear, beyond its focus on model-based trajectory planning.

The robot experiments lack clarity, with important details omitted.

The limitations and potential future directions of this work are not discussed.

The learned dynamics function relies on a well-designed simulator, which may limit the scalability of this method to a wide variety of tasks.

The method assumes static initial poses for obstacles and goal regions, making it challenging to adapt to dynamic changes.

**Questions:**

What are the definitions of open-loop and closed-loop performance? They are unclear, making the results in Figure 5 difficult to interpret.

For data collection to train the dynamics function, thousands of episodes are gathered in the simulator. How is action sampling handled for each task? Are actions randomly generated? Is there a sim-to-real gap? Why not train an RL agent in the simulator instead?

Is it possible to extend the current method to scenarios where obstacles or goal regions are dynamic?

What is the action space or control signal for most tasks? It appears to be limited to translations in the 2D plane.

With a 20-step planning horizon, how many actions does the controller execute? What is the planner’s operating frequency?

It appears that much of the processing time is consumed by searching. It will dramatically increase computation time and workload when the action space includes both 3D translations and 3D rotations. Would this limit scalability to a full 3D action space?

---

> ### Author Response · Authors · 2024-11-24
> **Author Response to Reviewer U2wt (Part 1)**
>
> Thank you for acknowledging our well-structured writing, well-defined formulation, and the extensive experiments we have conducted. We greatly appreciate your thoughtful comments and constructive suggestions. In response, we have clarified the novelty and distinctions of our BaB-ND compared to neural network verification techniques and provided additional details regarding our real-world experiments in the revised paper. Furthermore, we have included a **new real-world experiment** demonstrating the application of our BaB-ND in dynamic scenarios.
>
> > The specific novelty and distinctions of the proposed method compared to neural network verifiers are unclear, beyond its focus on model-based trajectory planning.
>
> We have revised our paper to include a discussion of the distinctions between our BaB-ND and existing neural network verification algorithms in the method section and appendix, summarized as follows:
>
> 1. **Goals**: BaB-ND aims to optimize an objective function involving neural dynamics models to solve challenging planning problems, seeking a **concrete solution** $\tilde{\boldsymbol{u}}$ (a near-optimal action sequence) in space $\mathcal{C}$ to an objective-function minimization  problem $\min_{{\boldsymbol{u}} \in \mathcal{C}}{f({\boldsymbol{u}})}$. In contrast, neural network verification focuses on **proving a sound lower bound** of $f({\boldsymbol{u}})$ in the space $\mathcal{C}$, and a **concrete solution** $\tilde{\boldsymbol{u}}$ is not needed. These fundamental distinctions in goals lead to different algorithm design choices.
>
> 2. **Branching Techniques**: In BaB-ND, branching techniques are designed to effectively guide the search for better concrete solutions, considering both the lower and upper bounds of the best objective. In neural network verification, branching techniques focus solely on improving the lower bounds.
>
> 3. **Bounding Approaches**: While existing bounding approaches, such as CROWN from neural network verification, can provide provable lower bounds for objectives, they are neither effective nor efficient for planning problems. To address this, we adapt the CROWN algorithm with propagation early-stop and search-integrated bounding to efficiently obtain tight bound estimations.
>
> 4. **Searching Components**: BaB-ND introduces a searching component within the branch-and-bound procedure to identify optimal solutions for planning problems. This component not only benefits from the BaB framework but also guides it to enhance the search process. In contrast, neural network verifiers typically lack the searching component [1, 2], as their primary focus is on obtaining lower bounds for the objective rather than determining objective values for specific inputs.
>
> [1] S. Wang et al., ‘Efficient bound propagation with per-neuron split constraints for complete and incomplete neural network verification’, NeurIPS, 2021.
> [2] H. Zhang et al., ‘General Cutting Planes for Bound-Propagation-Based Neural Network Verification’, NeurIPS, 2022.
>
> > The robot experiments lack clarity, with important details omitted.
>
> We answer your specific questions regarding the details of robot experiments as below. And we provide a more detailed description in our revised Appendix D.4.
>
> > What are the definitions of open-loop and closed-loop performance? They are unclear, making the results in Figure 5 difficult to interpret.
>
> The key difference between open-loop and closed-loop is whether feedback from the environment is used.
>
> In the open-loop experiment, the trajectory is planned solely based on the initial and goal states, without incorporating any feedback from the environment during execution. Performance is evaluated by summing the planned costs over all steps.
>
> In the closed-loop experiment, feedback from the environment is continuously integrated by the controller to adjust its current plan, effectively compensating for discrepancies between model predictions and reality. Performance is assessed based on the final cost to the target.
>
> Our results demonstrate that, with our effective BaB planning framework, we achieve noticeably better performance in both open-loop and closed-loop settings.

---

> ### Author Response · Authors · 2024-11-24
> **Author Response to Reviewer U2wt (Part 2)**
>
> > For data collection to train the dynamics function, thousands of episodes are gathered in the simulator. How is action sampling handled for each task? Are actions randomly generated? Is there a sim-to-real gap? Why not train an RL agent in the simulator instead?
>
> During data collection, we randomly initialize the object's state and the end effector's position. Actions are then randomly sampled to encourage interactions between the end effector and the object, ensuring sufficient coverage of scene variations that the neural dynamics model may encounter during testing.
>
> Yes, there can be a sim-to-real gap. To address this, we use a closed-loop planning framework for real-world deployment that incorporates feedback from the environment. This approach effectively compensates for model errors and enables success across a diverse set of challenging manipulation tasks.
>
> While it is possible to train an RL agent in the simulator, that is not the focus of our work. This paper addresses the trajectory optimization problem involving a pre-trained dynamics model. Our key contribution is demonstrating how to effectively perform planning with learned dynamics models, rather than training improved RL agents. The study on effective neural dynamics model training is an important future work for the robot learning community.
>
>
> > What is the action space or control signal for most tasks? It appears to be limited to translations in the 2D plane.
>
> We have both 2D and 3D tasks. For the `Pushing with Obstacles` and `Object Merging` tasks, the action space is the movement of the pusher in the xy-plane, with a movement limit of [-3, 3] cm along each axis per step. For the `Object Sorting` task, the action space includes the position of the pusher and its movement. The pusher's position must be within the 2D workspace, and its movement is limited to [-20, 20] cm in the xy-plane. For the `Rope Routing` task, the action space is the movement of the gripper in 3D space, with the same movement limit of [-3, 3] cm along each axis.
>
> We agree that incorporating more complex 3D tasks could further enhance the significance of this work. As part of future extensions, we plan to apply our framework to tasks from [3, 4], such as dexterous and cloth manipulation. We are confident in the potential results, as these tasks typically rely on CEM/MPPI for planning, and we have demonstrated that our method consistently outperforms CEM/MPPI in both efficiency and performance.
>
> [3] A. Nagabandi et al., ‘Deep dynamics models for learning dexterous manipulation’, CoRL, 2020.
> [4] X. Lin et al., ‘Learning Visible Connectivity Dynamics for Cloth Smoothing’, CoRL, 2021.
>
> > With a 20-step planning horizon, how many actions does the controller execute? What is the planner’s operating frequency?
>
> In the closed-loop planning, the controller executes around 140 actions for a 20-step planning horizon. This is because we use a smaller action bound during execution to provide more precise control. The planner operates at a frequency of about 4 fps. We have included more details about our real-world experiment setting in Appendix D.4.
>
> > The limitations and potential future directions of this work are not discussed.
>
> We included additional discussions on the limitations and potential future directions in our revised version in Appendix A.4.
>
> We acknowledge that although our branch and bound procedure can help systematically explore the input space and efficiently guide searching, its planning performance can be limited by the underlying sampling-based searching algorithms. Developing and integrating stronger searching algorithms into our BaB-ND can extend its capability on more challenging planning problems. Adaptively adjusting the branching heuristics and bounding estimation can also further improve the convergence on more general cases.

---

> ### Author Response · Authors · 2024-11-24
> **Author Response to Reviewer U2wt (Part 3)**
>
> > The learned dynamics function relies on a well-designed simulator, which may limit the scalability of this method to a wide variety of tasks.
>
> Learning neural dynamics models for planning is an emerging technique, and we acknowledge the challenges of building effective models for diverse scenarios. Prior work ([3-5] and others discussed in the related work section) has explored various types of neural dynamics models to address limitations in their design and learning. Some of these models are trained using well-designed simulators, but when such simulators are unavailable, many are trained directly through real-world interactions [6].
>
> That said, addressing these challenges is not the focus of our work. Instead, our paper emphasizes **effective planning using pre-trained neural dynamics models** for robotic manipulation tasks. Our method is compatible with MLPs and GNNs, which are widely used in prior work [3-6], and can be seamlessly integrated with advancements in training methods for these models, further broadening their potential applications.
>
> [3] A. Nagabandi et al., ‘Deep dynamics models for learning dexterous manipulation’, CoRL, 2020.
> [4] X. Lin et al., ‘Learning Visible Connectivity Dynamics for Cloth Smoothing’, CoRL, 2021.
> [5] Z. Liu et al., ‘Model-Based Control with Sparse Neural Dynamics’, NeurIPS, 2023.
> [6] H. Shi et al., ‘RoboCook: Long-Horizon Elasto-Plastic Object Manipulation with Diverse Tools’, CoRL, 2023.
>
> > The method assumes static initial poses for obstacles and goal regions, making it challenging to adapt to dynamic changes. … Is it possible to extend the current method to scenarios where obstacles or goal regions are dynamic?
>
> We conducted an additional experiment on varied conditions, like the configuration of the target or obstacles may change during the execution. Please refer to the anonymous video links: [Demo video for adding a new obstacle](https://streamable.com/rla3ud) and [Demo video for changing target state](https://streamable.com/852ug2). The results show that our BaB-ND can effectively provide new long-horizon action sequences to reach the target when the environment changes.
>
> > It appears that much of the processing time is consumed by searching. It will dramatically increase computation time and workload when the action space includes both 3D translations and 3D rotations. Would this limit scalability to a full 3D action space?
>
> Our method exactly targets providing a significantly more scalable alternative to popular existing sampling/searching-based methods (e.g., MPPI and CEM). As you noted, existing sampling-based methods face scalability challenges with large input dimensions (e.g., long-horizon action sequences in 3D space) because they **search for solutions across the entire input space**, requiring an exponentially increasing number of samples to achieve sufficient coverage.
> In contrast, our BaB-ND strategically splits and prunes unpromising regions of the input space, guiding and improving the effectiveness of existing sampling-based methods. Our experiments (Figure 6(b) in the paper) demonstrated that **our BaB-ND produces noticeably better results and scales more effectively to larger input dimensions** compared to existing sampling/searching-based methods.
>
> We have attempted to address all your questions and made the corresponding changes to the paper. We believe the paper is now much stronger thanks to your suggestions. Please let us know if you have any additional concerns.

---

> > ### Comment · Reviewer_U2wt · 2024-11-26
> >
> > Thank you for addressing my questions—some of my concerns have been resolved.
> >
> > The distinctions between BaB-ND and related work are now clearer.
> >
> > With the clarification of your closed-loop setting, it’s evident that the method can address dynamic scenes to some extent. The new video helps demonstrate this capability.
> >
> > For the rope task, could you clarify if the action space includes 3 degrees of freedom (DoF)? Were there any tasks that required incorporating a rotation action?
> >
> > Regarding the dynamic model training in simulation, what does the current observation space include?
> >
> > I appreciate the discussion on limitations and future direction. However, expanding on specific content and including more corner cases would strengthen the section.

---

> ### Comment · Area_Chair_sADg · 2024-11-25
>
> Dear Reviewer,
>
> Please provide feedback to the authors before the end of the discussion period, and in case of additional concerns, give them a chance to respond.
>
> Timeline: As a reminder, the review timeline is as follows:
>
> November 26: Last day for reviewers to ask questions to authors.
>
> November 27: Last day for authors to respond to reviewers.

---

> ### Author Response · Authors · 2024-11-29
> **Follow-Up Response to Reviewer U2wt**
>
> Thank you for your feedback. We greatly appreciate your follow-up questions and the opportunity to provide further clarification on our work. We have **revised our paper** (Appendix D.1) to include a detailed description of input and observation spaces, and also **expanded the limitations section** based on your suggestion.
>
> > For the rope task, could you clarify if the action space includes 3 degrees of freedom (DoF)? Were there any tasks that required incorporating a rotation action?
>
> In our `Rope Routing` task, the action space consists of 3 DOF, defined as the movement of the robot gripper in 3D space along the xyz axes $(\Delta x, \Delta y, \Delta z)$. We additionally provide a [visualization](https://imgdrop.io/image/g5Yld) of the action definitions for all tasks to illustrate the action space, please check out to [visualization](https://imgdrop.io/image/g5Yld) (d) for details.
>
> In our `Object Sorting` task, the orientation of the pusher rotates around the gravity direction, determined by the starting and ending locations of the pusher (i.e., the pusher is always perpendicular to the pushing direction $(\Delta x, \Delta y)$ in [visualization](https://imgdrop.io/image/g5Yld) \(c\)).
>
> We have updated Appendix D.1 to include more details about the definition of actions in our tasks, and will include the [visualization](https://imgdrop.io/image/g5Yld) in our future revision.
>
> > Regarding the dynamic model training in simulation, what does the current observation space include?
>
> We follow a general design principle of using 3D points as the state representation for training the dynamics models across all tasks, as 3D points can capture both geometric and semantic information and are easy to transfer between simulation and the real world. Specifically:
>
> - For the tasks `Pushing with Obstacles` and `Object Merging`, we assign keypoints on objects to capture the geometric information of the objects (e.g., 4 keypoints for a 'T'-shaped object and 3 keypoints for an 'L'-shaped object).
> - For the task `Object Sorting`, we consider an object-centric representation by using the center point of all objects as the state and model their interactions as a graph using graph neural networks.
> - For the task `Rope Routing`, we sample multiple keypoints (e.g., 10) along the rope as the state to better represent its geometric configuration.
>
> We have **updated Appendix D.1 and Figure A12 to include a visualization** and more details about the definition of the observation spaces in our tasks.
>
> > I appreciate the discussion on limitations and future direction. However, expanding on specific content and including more corner cases would strengthen the section.
>
> Thank you for your suggestions. We have expanded our discussion on limitations and future directions with specific examples and corner cases:
> 1. **Planning performance depends on the prediction errors of neural dynamics models.**
>
>     The neural dynamics model may not perfectly match the real world. As a result, our optimization framework, BaB-ND, may achieve a good objective as predicted by the learned dynamics model but still miss the target (e.g., the model predicts that a certain action reaches the target, but in reality, the pushing action overshoots). While improving model accuracy is not the primary focus of this paper, future research could explore more robust formulations that account for potential errors in neural dynamics models to improve overall performance and reliability.
>
> 2. **Optimality of our solution may be influenced by the underlying searching algorithms.**
>
>     The planning performance of BaB-ND is inherently influenced by the underlying sampling-based searching algorithms (e.g., sampling-based methods may over-exploit or over-explore the objective landscape, resulting in suboptimal solutions in certain domains). Although our branch-and-bound procedure can mitigate this issue by systematically exploring the input space and efficiently guiding the search, incorporating advanced sampling-based searching algorithms with proper parameter scheduling into BaB-ND could improve its ability to tackle more challenging planning problems.
>
> 3. **Improved branching heuristics and strategies are needed for more efficiently guiding the search for more challenging settings.**
>
>     There is still room for improving the branching heuristics and bounding strategies to generalize across diverse tasks (e.g., our current strategy may not always find the optimal axis to branch). Future efforts could focus on developing more advanced branching strategies for challenging applications, potentially leveraging reinforcement learning based approaches.
>
> We hope that we have addressed your follow-up questions and we greatly appreciate your feedback. Please let us know if you have any additional concerns.

---

> ### Author Response · Authors · 2024-12-01
> **Follow-Up on Discussion Period**
>
> Dear Reviewer U2wt,
>
> Thank you again for the opportunity to provide further clarification on our work. We have addressed your follow-up questions in detail and revised the paper accordingly. The changes, especially the new [visualization](https://imgdrop.io/image/g5Yld), further strengthened its clarity.
>
> As tomorrow is the final day for receiving feedback from you, we would greatly appreciate any additional comments you may have and kindly hope you might reconsider the score.
>
> Best Regards,
> The Authors

---

### Official Review · Reviewer_kMxQ · 2024-11-02

**Soundness:** 3
**Presentation:** 2
**Contribution:** 2
**Rating:** 6
**Confidence:** 3

**Summary:**

The paper presents an approach for solving complex long-horizon motion planning tasks using a GPU-accelerated branch-and-bound (BaB) framework. The primary innovation is the integration of bound propagation methods inspired by neural network verification, namely CROWN, to handle the non-linearity of neural dynamics models. This method allows for efficient partitioning of the search space and systematic pruning of sub-domains that cannot yield better solutions, focusing the search on promising areas. The authors demonstrate that BaB-ND outperforms existing sampling-based and mixed-integer programming (MIP) methods in handling contact-rich manipulation tasks like planar pushing and object routing. The framework supports various neural architectures and shows scalability, making it applicable to real-world robotic planning problems.

**Strengths:**

- The paper tackles an important and challenging problems in robot motion planning, especially in scenarios involving long-horizon, contact-rich tasks.
- The paper provides a solid overview of related work in neural dynamics, motion planning, and neural network verification, positioning the contribution well within the context of existing research.
- The use of a branch-and-bound (BaB) framework integrated with bound propagation methods adapted from neural network verification shows potential for enhancing motion planning capabilities.
- The proposed method is designed to support various neural network architectures, allowing applicability to different robotic models and tasks.
- The experiments are complex and challenging, showcasing the method’s applicability to real-world scenarios such as object routing and planar pushing.

**Weaknesses:**

- The method section is highly abstracted, which leaves readers unable to connect theoretical concepts such as the objective function, inputs, bounds, and sub-domains to practical robotic manipulation problems. Explanations remain too abstract and disconnected from practical robotic tasks. Also, the paper does not detail the method sufficiently, particularly the bounding step, making it difficult for readers to understand the full implementation of the approach.
- Understanding the methodology relies heavily on prior familiarity with the CROWN method, making the paper less accessible to readers who are not already experts in neural network verification techniques.
- Although GPU acceleration is mentioned as a key advantage, the paper does not provide sufficient detail or evidence to demonstrate how GPU acceleration is implemented or its specific benefits.
- While the paper compares performance against sampling-based methods, the absence of Mixed-Integer Programming (MIP) as a baseline for evaluating the optimality of solutions is a significant oversight. Additionally, runtime performance is not benchmarked against methods like MPPI and CEM.
- The paper does not include an ablation study to show the importance of individual components or design choices in the proposed method, leaving gaps in understanding the contribution of each element.
- From a robotics perspective the use of a branch-and-bound framework with neural dynamics in contact-rich robotic manipulation seems novel, however, the novelty and contribution from a machine learning perspective is not clear.

**Questions:**

- Can you provide more details on the bounding step of the method? How are bounds estimated in a way that ensures meaningful pruning of sub-domains in practical robotic tasks?
- Could you illustrate the method using a specific example of a robotic manipulation task, detailing how abstract concepts like the input u and bounds map to task-specific elements?
- How is GPU acceleration implemented in your approach, and what specific parts of the framework benefit from it? Could you provide runtime comparisons with and without GPU acceleration?
- Why was Mixed-Integer Programming (MIP) not used as a baseline for comparing the quality of the solutions? Similarly, why were methods like MPPI and CEM not included for runtime performance comparisons?
- Have you considered performing an ablation study to show the contribution of individual components, such as the branching heuristic or specific modifications to the bound propagation method?
- Could you provide more details on the novelty of the approach from a machine learning point of view ?

---

> ### Author Response · Authors · 2024-11-24
> **Author Response to Reviewer kMxQ (Part 1)**
>
> Thank you for acknowledging our innovation in integrating bound propagation methods inspired by neural network verification, as well as the effectiveness and applicability of our BaB-ND on challenging manipulation tasks. We greatly appreciate your thoughtful comments and constructive suggestions. In response, we have included more detailed illustrations connecting our algorithm to a toy robotic task and have conducted **additional experiments** as suggested:
> 1. Runtime analysis for GPU and CPU versions of our BaB-ND;
> 2. Ablation study for our BaB-ND;
> 3. Runtime analysis comparing our BaB-ND with sampling-based methods;
> 4. Search space analysis for our BaB-ND.
>
> > Accessibility of method section, especially to readers not familiar with neural network verification techniques
>
> Thank you for your suggestion. To improve the accessibility of our method, we have added a new example in Appendix A.1 that demonstrates its application to a practical, toy robotic task. This example includes a clear definition of the objective, inputs, bounds, and subdomains.
>
> While we were unable to provide a full introduction to CROWN due to the extensive mathematical exposition it would require, we emphasize one key insight: CROWN provides a lower bound for our objective function, which is used in the branch-and-bound procedure to prune unhelpful subdomains. We have revised the paper to make this point clearer. Additionally, we address your two specific questions regarding our methods below.
>
> > Can you provide more details on the bounding step of the method? How are bounds estimated in a way that ensures meaningful pruning of subdomains in practical robotic tasks?
>
> We first emphasize that it is not essential to understand the bounding algorithm to understand the high-level idea of BaB-ND. From the view of BaB-ND, bounding as a component is used to provide the lower bounds of the objective function in different subdomains. These bounds are used to prune unpromising subdomains and identity promising subdomains. We have improved our writing to highlight its high-level idea and **provided a detailed illustration of the bounding component with a simple example on an MLP in our revised Appendix B.2**.
>
> To understand the effectiveness of the estimated bounds in pruning subdomains for practical robotic tasks, we conducted an additional experiment on task `Pushing with Obstacles`. Below, we report three metrics over the brand-and-bound iterations: (1) the normalized space size of pruned subdomains, (2) the size of the selected subdomains for searching, and (3) the improvements in the objective value.
>
> As iterations progress, **the average total space size of pruned subdomains increases rapidly and then converges**, indicating the effectiveness of our bounding methods. Once the pruned space size reaches a plateau, **the total space size of selected promising subdomains continues to decrease,** showing that the estimated lower bounds remain effective in identifying promising subdomains. The **decreasing objective over iterations** further confirms that BaB-ND focuses on the most promising subdomains, reducing space size to the magnitude of $1 \times 10^{-4}$.
>
>    | Iterations | 0 | 4 | 8 | 12 | 16 | 20 |
>    |-|-|-|-|-|-|-|
>    | Pruned space size (↑) | 0.0000 | 0.7000 | 0.8623 | 0.8725 | 0.8744 | 0.8749 |
>    | Selected space size (↓) | 1.0000 | 0.3000 | 0.0412 | 4.8339e-03 | 5.2795e-04 | 2.8972e-04 |
>    | Objective (↓) | 41.1222 | 36.0511 | 35.5091 | 34.8024 | 33.8991 | 33.3265 |

---

> ### Author Response · Authors · 2024-11-24
> **Author Response to Reviewer kMxQ (Part 2)**
>
> > Could you illustrate the method using a specific example of a robotic manipulation task, detailing how abstract concepts like the input u and bounds map to task-specific elements?
>
> We have provided a simplified example of our manipulation task with an illustration of our method in Figure 3 in the paper. Below, we refer to Figure 3 to introduce abstract concepts:
>
> In Figure 3(a), we first define the configuration of the task, where the robot moves left or right to push an object toward the target. Now we define the inputs, objective, neural dynamics model, subdomains, bounds and the pruning process involved in this task:
>
> - **Inputs**: The inputs in our formulation are the action sequence with a horizon of $H$. In this simplified case, we consider a horizon of 1 and restrict the action to 1D. So in this case, the inputs become a 1D action $u \in \mathcal{C}$ representing the movement of the robot pusher, with $\mathcal{C} = [-l, l]$ as its domain, where $l$ is the maximum movement distance (e.g., $1 \, \text{cm}$ in practice). A value of $u < 0$ means the robot moves left, while $u > 0$ means the robot moves right.
>
> - **Objective**: The objective $f(u)$ measures the distance between the object and the target under a specific action $u$. In this case, $f(u) = d\_1^2 + d\_2^2$, where $d\_1$ is the distance between a keypoint ($P\_1$) on the object and the corresponding keypoint ($P\_{1,T}$) on the target, and $d\_2$ is the distance between another keypoint pair ($P\_2$ and $P\_{2,T}$). For example, if the robot moves left ($u < 0$), $d\_2$ decreases while $d\_1$ increases.
>
> - **Neural dynamics model**: The values of $d\_1$ and $d\_2$ depend on a neural network dynamics model $f\_{\text{dyn}}$. This model takes as input the current positions of $P\_1$ and $P\_2$, along with an action $u$, to predict the next positions of $P\_1$ and $P\_2$. Based on these predictions, $d\_1$ and $d\_2$ are updated accordingly and $f(u)$ may exhibit non-convex behavior.
>
> Our goal in planning is to find the optimal action $u^*$ that minimizes $f(u)$. To achieve this, we propose a branch-and-bound-based method. In Figure 3(b), \(c\), and (d), we illustrate some key concepts and the overall procedure of our method:
>
> - **Subdomains**: A subdomain $\mathcal{C}\_i \in \mathcal{C}$ is a subset of the entire input domain $\mathcal{C}$ (action space in our case). For example, in Figure 3(b), we initially split $\mathcal{C} = [-l, l]$ into two subdomains: $\mathcal{C}\_1 = [-l, 0]$ and $\mathcal{C}\_2 = [0, l]$, to separately analyze left and right movements.
>
> - **Lower and upper bounds**: Each subdomain $\mathcal{C}\_i$ has associated lower and upper bounds: $\underline{f}^*\_{\mathcal{C}\_i}$ and $\overline{f}^*\_{\mathcal{C}\_i}$. These represent the bounds of the best objective in $\mathcal{C}\_i$ (${f}^*\_{\mathcal{C}\_i} := \min\_{u \in \mathcal{C}\_i} f(u)$). For example, if the optimal objective (the sum of the squared distances between keypoint pairs, $d\_1^2 + d\_2^2$) given by the best action in $\mathcal{C}\_i$ is 2, we might estimate $\underline{f}^*\_{\mathcal{C}\_i} = 1$ and $\overline{f}^*\_{\mathcal{C}\_i} = 3$. ($1\leq \min\_{x\in\mathcal{C}\_i}d\_1^2 + d\_2^2 = 2 \leq 3$) Intuitively, $\underline{f}^*\_{\mathcal{C}\_i}$ overestimates the effect of the optimal action on improving $f(u)$, while $\overline{f}^*\_{\mathcal{C}\_i}$ underestimates it.
>
> - **The pruning process**: We compute $\underline{f}^*\_{\mathcal{C}\_i}$ using bounding (Figure 3\(c\)) and $\overline{f}^*\_{\mathcal{C}\_i}$ using searching (Figure 3(d)). These bounds allow us to determine whether a subdomain $\mathcal{C}\_i$ is promising for containing the optimal action $u^*$ or whether it can be pruned as unpromising. For instance, in Figure 3, if $\underline{f}^*\_{\mathcal{C}\_1} = 4$ and $\overline{f}^*\_{\mathcal{C}\_2} = 3$, it means that no objective better than 4 can be achieved in $\mathcal{C}\_1$, while no objective worse than 3 can occur in $\mathcal{C}\_2$. In this case, we can directly prune $\mathcal{C}\_1$ without further exploration.
>
> - **The branch-and-bound process**: Our branch-and-bound process include 3 key components: *branching*, *bounding* and *searching*. Specifically, We can initially *bound* and *search* on the input space $\mathcal{C}$, and then split it into subdomains like $\mathcal{C}\_1$ and $\mathcal{C}\_2$ in Figure 3(b) (*branching*). Then on the new subdomains, we continue to repeat these steps, and we may *prune* any subdomains with lower bounds larger than the best objective we found so far. Iteratively,the algorithm will concentrate on improving the solution in the smaller and more promising subspace, leading to better planning actions.

---

> > ### Author Response · Authors · 2024-11-24
> > **Author Response to Reviewer kMxQ (Part 3)**
> >
> > > How is GPU acceleration implemented in your approach, and what specific parts of the framework benefit from it?
> >
> > All components of our planning framework (*branching*, *bounding*, and *searching*) are computed on the GPU and significantly benefit from GPU acceleration:
> >    - *Branching*: Branching operations for subdomains are batched and executed in parallel on the GPU, enabling efficient processing of multiple subdomains simultaneously.
> >    - *Bounding*: This step leverages the neural network's structure, performing bounding propagation and estimation through a back-propagation process, which is highly parallelizable on the GPU.
> >    - *Searching*: The underlying sampling-based algorithm is parallelized across both samples and subdomains, taking full advantage of GPU capabilities for improved computational performance.
> >
> > > Could you provide runtime comparisons with and without GPU acceleration?
> >
> > Thank you for your suggestion. We evaluate the performance improvement from CPU to GPU using the same test cases as in the Scalability Analysis (Figure 6(a)) in our paper and report the results below. The complete results have been included in the revised Appendix C.1. If the process does not finish within 300 seconds, we report "NaN" in the table.
> >
> >    The results clearly demonstrate that **our implementation benefits significantly from GPU acceleration**, achieving over **10x speedup** compared to the CPU version, even for small planning problems.
> >
> >    | (Model size, Planning horizon) | (0.232K,1) | (0.712K,3) | (2.440K,5) | (34.31K,10) | (134.2K,15) | (530.2K,20) |
> >    |-|-|-|-|-|-|-|
> >    | CPU | 6.9293 | 21.0398 | 48.6264 | NaN | NaN | NaN |
> >    | GPU | **1.4455** | **2.5975** | **4.6148** | **15.2645** | **32.5773** | **90.5053** |
> >
> >
> > > Why was Mixed-Integer Programming (MIP) not used as a baseline for comparing the quality of the solutions?
> >
> > We want to emphasize that MIP and sampling based methods have quite different characteristics, and thus the comparison in some settings are not meaningful or possible. MIP provides high quality solutions but it works only for very small models, and **not practical for any of the real-world experiments we have done**. MIP won’t be able to produce any feasible solution and will timeout with a planning horizon of 3 and model size of 2.4K, while our experiments involve a time horizon of 20 and network size of 134.2K. Thus, for the four real-world robotic tasks, we could only include sampling based methods.
> >
> >
> > > Similarly, why were methods like MPPI and CEM not included for runtime performance comparisons?
> >
> > The runtime performance comparisons with MIP reported in our paper was **conducted with much smaller networks, much shorter time horizons, and easier environments (no obstacles)**, solely serving as a direct comparison to [1]. These are not the settings of our main experiments. Under these much simpler settings, we have conducted comparisons to sampling-based methods as suggested by the reviewer and reported the results in our revised Appendix C.1. We found that in much less challenging settings, all these methods (sampling-based methods, MIP and ours) can perform similarly, though MIP exhibits significantly worse scalability. But we again want to emphasize that this is a **synthetic setting purely serving as a comparison to [1]**, and not the main setting studied in our paper.
> >
> > [1] Z. Liu et al., ‘Model-Based Control with Sparse Neural Dynamics’, NeurIPS, 2023.
> >
> > To fully understand the runtime performance of our method compared to approaches like MPPI and CEM, we conducted an additional experiment to evaluate the scalability in the **realistic setting** of the `Pushing with Obstacles` task. Parameters for all methods were adjusted to ensure similar runtimes for the largest problems.
> >
> > Below are the average runtimes of all methods across six diagonal elements in the heatmap (Figure A 10), ranging from the smallest model size and shortest planning horizon to the largest model size and longest planning horizon:
> >
> >    |Runtime (↓)|(0.232K,1)|(0.712K,3)|(2.440K,5)|(34.31K,10)|(134.2K,15)|(530.2K,20)|
> >    |-|-|-|-|-|-|-|
> >    | GD | 0.6930 | 1.4128 | 2.8380 | 15.5129 | 39.6837 | 117.8663 |
> >    | MPPI | 0.7800 | 1.6758 | 3.4854 | 16.7010 | 34.6054 | 116.5935 |
> >    | CEM | 1.0159 | 1.8450 | 3.5274 | 16.5886 | 34.6403 | 115.8820 |
> >    | Ours | 3.5938 | 6.3131 | 9.3808 | 25.0006 | 45.6266 | 114.6788 |
> >
> > We also report the average objectives for all methods on the largest four planning problems to evaluate their effectiveness:
> >
> > |Objective (↓)|(134.2K,15)|(134.2K,20)|(530.2K,15)|(530.2K,20)|
> > |-|-|-|-|-|
> > | GD | 57.2768 | 64.4789 | 54.7078 | 60.2575 |
> > | MPPI | 47.4451 | 53.7356 | 45.1371 | 45.6338 |
> > | CEM | 47.0403 | 47.6487 | 43.8235 | 38.8712 |
> > | Ours | **46.0296** | **46.1938** | **41.6218** | **34.6972** |
> >
> > These results show that our **BaB-ND achieves similar scalability but noticeably better objective compared with sampling-based baseline methods.**

---

> ### Author Response · Authors · 2024-11-24
> **Author Response to Reviewer kMxQ (Part 4)**
>
> > Ablation study to show the contribution of individual components, such as the branching heuristic or specific modifications to the bound propagation method
>
> We conducted additional ablation studies on the `Object Sorting` and `Pushing with Obstacles` tasks to evaluate how different design choices impact planning performance following three settings below:
>
> 1. **Different heuristics for selecting subdomains to split:**
>     - 1. Select based on both lower and upper bounds ($\underline{f}^*\_{\mathcal{C}\_i}$ and $\overline{f}^*\_{\mathcal{C}\_i}$).
>     - 2. Select based only on upper bounds ($\overline{f}^*\_{\mathcal{C}\_i}$ only).
>     - 3.  Select based only on lower bounds ($\underline{f}^*\_{\mathcal{C}\_i}$ only).
>
>    Among these heuristics, selecting promising subdomains based on both lower and upper bounds achieves better planning performance by **balancing exploitation and exploration effectively** compared to the other strategies.
>
>    || $\underline{f}^*\_{\mathcal{C}\_i}$ and $\overline{f}^*\_{\mathcal{C}\_i}$ | $\overline{f}^*\_{\mathcal{C}\_i}$ only | $\underline{f}^*\_{\mathcal{C}\_i}$ only |
>    |-|-|-|-|
>    | Object Sorting | **31.0482** | 32.1249 | 33.2462 |
>    | Pushing with Obstacles | **31.9839** | 32.2777 | 32.6112 |
>
> 2. **Different heuristics for splitting subdomains:**
>     - 1. Split based on the largest $(\overline{\boldsymbol{u}}\_{j} - \underline{\boldsymbol{u}}\_{j}) \cdot |n\_j^{\text{lo}} - n\_j^{\text{up}}|$.
>     - 2. Split based on the largest $(\overline{\boldsymbol{u}}\_{j} - \underline{\boldsymbol{u}}\_{j})$.
>     - 3. Split based on the largest $|n\_j^{\text{lo}} - n\_j^{\text{up}}|$.
>
>    Our heuristic demonstrates superior planning performance on both tasks by **effectively identifying important input dimensions to split.**
>
>    || $(\overline{\boldsymbol{u}}\_{j} - \underline{\boldsymbol{u}}\_{j}) \cdot \|n\_j^{\text{lo}} - n\_j^{\text{up}}\|$ | $(\overline{\boldsymbol{u}}\_{j} - \underline{\boldsymbol{u}}\_{j})$ | $\|n\_j^{\text{lo}} - n\_j^{\text{up}}\|$ |
>    |-|-|-|-|
>    | Object Sorting | **31.0482** | 34.5114 | 32.8438 |
>    | Pushing with Obstacles | **31.9839** | 32.3869 | 32.6989 |
>
> 3. **Bounding component:**
>     - 1. Use our bounding approach with propagation early-stop and search-integrated bounding.
>     - 2. Use constant zero as trivial lower bounds to disable the bounding component.
>     - 3. Disable both the bounding component and our heuristic for selecting subdomains to split.
>
>    Our bounding component improves planning performance by **obtaining tight bound estimations, helping prune unpromising subdomains to reduce the search space, and prioritizing promising subdomains for searching.**
>
>    || Ours | Disable bounding| Disable bounding and domain selection heuristic|
>    |-|-|-|-|
>    | Object Sorting | **31.0482** | 33.6110 | 34.4535 |
>    | Pushing with Obstacles | **31.9839** | 32.3419 | 34.6227 |
>
> > Could you provide more details on the novelty of the approach from a machine learning point of view ?
>
> From a robotics and motion planning perspective, our approach demonstrates significantly better performance compared to principled optimization tools like MIP and other sampling-based motion planners. This is demonstrated across diverse tasks (e.g., rigid and deformable objects, object piles, and non-convex feasible regions) and model architectures (e.g., MLP- and GNN-based neural dynamics models). As noted by Reviewer ci99, “It might set a new direction in combining machine learning verification methods with motion planning.”
>
> From a machine learning perspective, while our BaB-ND is inspired by alpha-beta-CROWN, a method that has achieved notable success in **formal verification of machine learning**, it **cannot be directly applied to the planning and optimization challenges** we address. To bridge this gap, we completely redesigned the *branching* techniques, extensively adapted the underlying *bounding* algorithm, and integrated it with sampling-based planning methods (*searching*). These technical innovations were essential for achieving superior planning performance across a range of complex robotic manipulation tasks.
>
> We have attempted to address all your questions and made the corresponding changes to the paper. We believe the paper is now much stronger thanks to your suggestions. Please let us know if you have any additional concerns and we will really appreciate it if you can reevaluate our paper.

---

> ### Comment · Area_Chair_sADg · 2024-11-25
>
> Dear Reviewer,
>
> Please provide feedback to the authors before the end of the discussion period, and in case of additional concerns, give them a chance to respond.
>
> Timeline: As a reminder, the review timeline is as follows:
>
> November 26: Last day for reviewers to ask questions to authors.
>
> November 27: Last day for authors to respond to reviewers.

---

> ### Author Response · Authors · 2024-11-26
> **Follow-Up on Discussion Period**
>
> Dear Reviewer kMxQ,
>
> We have provided detailed responses to all your questions and revised our paper accordingly. We believe the paper is now much stronger, thanks to your constructive feedback and suggestions. As the paper revision deadline is less than a day away, we would appreciate it if you could kindly review our response and let us know if you have any additional questions. Thank you!
>
> Best Regards,
> The Authors

---

> > ### Comment · Reviewer_kMxQ · 2024-11-26
> >
> > Thank you for your detailed response. Many of my concerns have been effectively addressed. The revised manuscript now provides greater detail about the proposed method and its capabilities compared to related works. Additionally, the inclusion of new experiments enhances the impact of the paper and further validates the approach.
> >
> > However, I still find the paper somewhat difficult to follow, and a few questions remain unanswered, such as:
> >
> > - What are the exact inputs \( u \) in the robotic experiments? For instance, in the pushing with obstacles task, is the contact point between the pusher and the object determined by BaB-ND? The same question applies to the other tasks.
> >
> > - How does the proposed method manage input domains that contain discontinuities or are subject to constraints?
> >
> > That said, I believe the revisions made are substantial enough to warrant an increase in my score.

---

> ### Author Response · Authors · 2024-11-29
> **Follow-Up Response to Reviewer kMxQ (Part 1)**
>
> Thank you for your feedback. We greatly appreciate your follow-up questions and the opportunity to provide further clarification on our work. We have **revised our paper** to further include experiment details, provided a **new visualization** of the action definitions, as well as included **new experiments** to show our effectiveness under input discontinuities.
>
> > What are the exact inputs ( u ) in the robotic experiments? For instance, in the pushing with obstacles task, is the contact point between the pusher and the object determined by BaB-ND? The same question applies to the other tasks.
>
> For all tasks, the inputs $\boldsymbol{u}$ are defined as a sequence of actions over a planning horizon of $H$. The specific definitions of the actions vary across tasks briefly discussed below, and we’ve also updated detailed descriptions in Appendix D.1 to make our paper more clear. We additionally provide a [visualization](https://imgdrop.io/image/g5Yld) of the action definitions and will include it in our future revision.
>
> - For the `Pushing with Obstacles` and `Object Merging` tasks, the action is defined as the 2D movement of the pusher $(\Delta x, \Delta y)$ in the xy-plane. ([visualization](https://imgdrop.io/image/g5Yld) (a) and (b))
> - For the `Object Sorting` task, the action includes the 2D initial position of the pusher $(x, y)$ and its 2D movement $(\Delta x, \Delta y)$ in the xy-plane. ([visualization](https://imgdrop.io/image/g5Yld) \(c\))
> - For the `Rope Routing` task, the action is the 3D movement of the gripper $(\Delta x, \Delta y, \Delta z)$ in 3D space along the xyz axes. ([visualization](https://imgdrop.io/image/g5Yld) (d))
>
> For all tasks, we do not explicitly determine the contact point. Instead, our BaB-ND framework outputs a sequence of end-effector positions for the robot to follow, which implicitly decides, for instance, which side of the T-shaped object is being pushed.

---

> ### Author Response · Authors · 2024-11-29
> **Follow-Up Response to Reviewer kMxQ (Part 2)**
>
> > How does the proposed method manage input domains that contain discontinuities or are subject to constraints?
>
> There are indeed discontinuities in our input domains. For example, the `Pushing with Obstacles` task contains non-feasible regions occupied by obstacles. Below, we first detail how we consider the **discontinuities in our objective function** and then present **additional empirical results** showcasing the effectiveness of our method in handling discontinuities compared with the baselines. We have updated Appendix D.4 in our paper to make these settings more clear.
>
> **Objective formulation**: We implicitly define the discontinuities or constraints in our objective function by adding large penalty items on the non-feasible regions. Take the example of `Pushing with Obstacles`, we define the cost function at step $t$ as following:
>
> $c(x_t, u_t) = w_t\left\|x_t-x_{\text{target}}\right\|
>     + \lambda \sum_{{o} \in {O}} \left(\text{ReLU}(s_{{o}} - \left\|p_t-p_{{o}}\right\|) + \text{ReLU}(s_{{o}} - \left\|x_t-p_{{o}}\right\|)\right)$
>
> - $x_t$ and $u_t$ are the state and action at step $t$. $x_{\text{target}}$ is the target state. The state is the concatenation of the keypoint positions. The action is the pusher movement at every step.
> - $\left\|x_t-x_{\text{target}}\right\|$ gives the difference between the state at time $t$ and the target, weighted by $w_t$.
> - $p_t$ is the position of the robot pusher.
> - $p_o$ and $s_o$ give the position and size of an obstacle $o\in O$ where $O$ is the set of the obstacles.
> - The term $\text{ReLU}(s_{{o}} - \left\|p_t-p_{{o}}\right\|)$ introduces a positive penalty when the pusher is within the obstacle ${o}$ (i.e., when $s_{{o}} > \left\|p_t-p_{{o}}\right\|$).
> - Similarly, the second $\text{ReLU}$ term, $\text{ReLU}(s_{{o}} - \left\|x_t-p_{{o}}\right\|)$, penalizes distances between the keypoints and the obstacle ${o}$, calculated for each keypoint individually ($\left\|x_t-p_{{o}}\right\|$ is computed keypoint-wise).
> - $\lambda$ is a constant value that penalizes collisions.
>
> **Performance change with varying input discontinuities**: We conducted a follow-up experiment by removing the obstacles (non-feasible regions) in the problem, simplifying the objective function. Below, we report the performance of different methods on the simplified objective function (w/o obstacles) and the original objective function (w/ obstacles).
>
> The results show that in simple cases, although our BaB-ND consistently outperforms baselines, MPPI and CEM provide competitive performance. In contrast, **in cases with constraints and discontinuities, BaB-ND significantly outperforms the baselines**, demonstrating its effectiveness in handling these complex cases.
>
> |Objective (↓, *w/o obstacles*)|(134.2K,15)|(134.2K,20)|(530.2K,15)|(530.2K,20)|
> |-|-|-|-|-|
> | GD | 64.5308 | 64.2956 | 63.0130 | 60.6300 |
> | MPPI | 34.4295 | 26.9970 | 33.8077 | 26.1204 |
> | CEM | 34.3864 | 26.7688 | 33.6669 | 25.9599 |
> | Ours | **34.2347** | **26.4841** | **33.6144** | **25.6603** |
>
> |Objective (↓, *w/ obstacles*)|(134.2K,15)|(134.2K,20)|(530.2K,15)|(530.2K,20)|
> |-|-|-|-|-|
> | GD | 57.2768 | 64.4789 | 54.7078 | 60.2575 |
> | MPPI | 47.4451 | 53.7356 | 45.1371 | 45.6338 |
> | CEM | 47.0403 | 47.6487 | 43.8235 | 38.8712 |
> | Ours | **46.0296** | **46.1938** | **41.6218** | **34.6972** |
>
> We hope that we have addressed your follow-up questions and we greatly appreciate your feedback. Please let us know if you have any additional concerns.

---

### Official Review · Reviewer_bbRD · 2024-11-04

**Soundness:** 3
**Presentation:** 3
**Contribution:** 3
**Rating:** 6
**Confidence:** 3

**Summary:**

The paper presents a GPU-accelerated Branch-and-Bound (BaB) framework for planning in robotic tasks using neural dynamics models, employing α,β-CROWN for bound propagation. The framework supports various neural network architectures, such as MLPs and GNNs, but the model requires customization based on different scenarios, selecting specific architectures for tasks like object sorting and rope routing​.

**Strengths:**

1. The framework leverages bound propagation from neural network verification (α,β-CROWN) to optimize neural dynamics in planning tasks. By structuring the search space into sub-domains and pruning non-promising regions, it shows potential for handling high-dimensional, complex scenarios.
2. BaB-ND effectively focuses on feasible solutions over global optimization, adapting BaB methods for motion planning in complex, contact-rich robotic tasks. It supports diverse neural architectures, such as MLPs and GNNs.
3. With GPU acceleration, BaB-ND hints at the potential to manage extended planning horizons and complex models

**Weaknesses:**

1. The code URL provided in the paper is currently unavailable.
2. Figure 6 has an ambiguous y-axis, making it difficult to understand what is being measured and, therefore, hard to assess the framework’s actual performance or effectiveness based on this chart.
3. Most experiments are conducted in 2D scenarios, and even the 3D rope manipulation task appears relatively simple, which raises questions about the framework's scalability to more complex 3D environments and interactions, especially with deformable objects.
4. Although a generalized pipeline is proposed, the framework’s actual scalability across more diverse and complex tasks remains unproven. The comparisons are mostly limited to similar methods, without broader baselines that might better validate its extensibility and robustness.
5. The current experiments focus on relatively simple tasks, leaving uncertainty about the framework’s effectiveness in more complex or varied scenarios. Custom neural network design for each task could make adaptation time-consuming and resource-intensive.
6. The framework’s success is sensitive to selecting suitable network architectures for each task. Without careful selection, mismatches could lead to substantial inefficiencies or even failure, indicating a high demand for expert design and tuning.

**Questions:**

1. Given that most experiments are based in 2D, what modifications would be necessary to extend the framework to more complex 3D tasks or interactions with deformable objects? Is the focus on 2D a practical choice, or does it reflect inherent limitations in handling higher-dimensional complexity?

2. Would including broader baselines from other methodologies provide a more comprehensive benchmark for evaluating the framework? Stronger comparisons could help clarify its extensibility.

3. How might the framework perform in a wider range of challenging scenarios? Additional experiments with more varied conditions would offer insight into its adaptability to complex tasks or novel object interactions.

4. Since each scenario requires a custom-designed network, how practical is this approach for real-world applications? What are the estimated time and resource requirements for designing and tuning a model for a new scenario, and could this affect the framework’s usability in resource-limited settings?

5. Does the current setup introduce complexity that might hinder practical implementation? If an optimal network is not chosen, how significantly would this impact performance, and are there ways to mitigate this dependency on precise network selection?

---

> ### Author Response · Authors · 2024-11-24
> **Author Response to Reviewer bbRD (Part 1)**
>
> Thank you for acknowledging our novel contributions in adapting branch-and-bound (BaB) for neural dynamics (ND) models and its potential to handle extended planning horizons and complex models. We greatly appreciate your thoughtful comments and constructive suggestions, and we have **included additional experiments** as suggested:
> 1. Broader baselines from other methodologies, specifically more conventional motion planning approaches.
> 2. Evaluating our BaB-ND under dynamic scenarios.
>
> We would also like to note the concern regarding the challenges in training neural dynamics models. However, we would like to clarify that this has been extensively studied in the robot learning community (as discussed in our related work section) but is not the focus of our paper. Instead, our work aims to address **motion planning problems** involving neural dynamics models.
>
> > The code URL provided in the paper is currently unavailable.
>
> The code URL (https://anonymous.4open.science/r/BaB-ND-68C3) works on our end. Could you please try accessing it again with a different network connection?
>
> > Figure 6 has an ambiguous y-axis, making it difficult to understand what is being measured and, therefore, hard to assess the framework’s actual performance or effectiveness based on this chart.
>
> We report the final cost for all experiments except rope routing. For `Rope Routing`, we find that relying solely on the final step cost is insufficient to evaluate trajectory success. A greedy trajectory that horizontally routes the rope might achieve a low final cost but fail to guide the rope into the slot. Therefore, we report the success rate for this task. We have clearly clarified this inconsistency in our revised paper.
>
> > Task diversity and scalability to more complex scenarios
>
> We would like to emphasize that we have already included a diverse set of tasks, encompassing rigid bodies and deformable objects, single objects and object piles, 2D and 3D actions, non-convex feasible regions, and long action sequences. Additionally, we have demonstrated the framework's applicability to both MLP- and GNN-based neural dynamics models.
>
> The tasks included in this paper are significantly more challenging than those in prior work using MIP for planning [1]:
>    - The `Object Pushing` task in [1] only considered 1-step planning. Our `Pushing with Obstacles` task introduces additional obstacles with diverse configurations, making it considerably more challenging and requiring a much longer planning horizon.
>    - The `Rope Manipulation` task in [1] is limited to 2D planar actions and short-horizon planning. In comparison, we extend this task to a 3D scenario where the rope must be routed into a tight-fitting slot, necessitating long-horizon planning.
>
> We agree that incorporating more complex 3D tasks could further enhance the significance of this work. As part of future extensions, we plan to apply our framework to tasks from [2, 3], such as dexterous and cloth manipulation. We are confident in the potential results, as these tasks typically rely on CEM/MPPI for planning, and we have demonstrated that our method consistently outperforms CEM/MPPI in both efficiency and performance.
>
> [1] Z. Liu et al., ‘Model-Based Control with Sparse Neural Dynamics’, NeurIPS, 2023.
> [2] A. Nagabandi et al., ‘Deep dynamics models for learning dexterous manipulation’, CoRL, 2020.
> [3] X. Lin et al., ‘Learning Visible Connectivity Dynamics for Cloth Smoothing’, CoRL, 2021.
>
> > Comparison with broader baselines from other methodologies
>
> For baselines from other methodologies, we identified the following two approaches from more conventional motion planning community (as has also been suggested by reviewer ci99).
> 1. Rapidly-exploring Random Tree (RRT)
> 2. Probabilistic Roadmap (PRM)
>
> We conducted additional experiments on task `Pushing with Obstacle` to compare the planning performance of our BaB-ND method against RRT and PRM. We also include the performance of the original baselines (i.e., GD, MPPI, and CEM) as a reference. The table below presents the final step cost at a planning horizon of $H$:
> ||GD|MPPI|CEM|RRT|PRM|Ours|
> |-|-|-|-|-|-|-|
> |Final step cost (↓)|4.1238|1.5082|1.0427|10.6472|13.2930|**0.2339**|
>
> The results demonstrate that our method significantly outperforms all other approaches. Implementation details for RRT and PRM have been included in Appendix D. The main reasons for the performance gap are as follows:
> 1. The search space in our task is complex and continuous, making it challenging for discrete sampling methods like RRT and PRM to achieve effective coverage.
> 2. These methods are prone to getting stuck on obstacles, often failing to reach the target state.

---

> ### Author Response · Authors · 2024-11-24
> **Author Response to Reviewer bbRD (Part 2)**
>
> > Practical considerations on selecting and training neural dynamics models
>
> Thank you for raising the concerns and these are important considerations. Learning neural dynamics models is a rising technique, and we acknowledge the challenges in building effective models for different scenarios. Prior work ([1-3] and many other works discussed in the related work section) has explored various types of neural dynamics models to address limitations in their design and learning. However, addressing these challenges is not the focus of our work. Instead, our paper focuses on **effective planning using pre-trained neural dynamics models** for robotic manipulation tasks. Our method is compatible with MLPs and GNNs which are widely used in prior work [1,3] and can be integrated with advancements in training methods for these models, further enhancing their potential applications.
>
> In practice, the model doesn’t have to be perfect. We apply **closed-loop feedback control to compensate for the prediction error** — we will constantly obtain feedback from the environment and replan to correct deviations from the original plan.
>
> **Estimated effort to obtain the neural dynamics models:** Below, we provide a breakdown of the time required for data collection, model training, and long-horizon planning across different tasks:
>
>    || Data Collection (min) | Model Training (min) | Long-Horizon Planning (min) |
>    |-|-|-|-|
>    | Pushing with Obstacles | 1 | 5 | 1 |
>    | Object Merging | 8.5 | 20 | 1 |
>    | Object Sorting | 4 | 20 | 1.3 |
>    | Rope Routing | 10 hours | 20 | 0.5 |
>
> Our framework is reasonably efficient: for most tasks shown in the paper, data collection (performed in simulation) takes less than 10 minutes, and model training requires approximately 20 minutes. The exception is the `Rope Routing` task, where data collection takes longer due to simulator inefficiencies. However, this can be addressed using faster simulators in the future.
>
> It is also important to note that data collection is performed offline and only once. As a result, it introduces minimal overhead for the practical deployment of our framework.
>
>
> > Given that most experiments are based in 2D, what modifications would be necessary to extend the framework to more complex 3D tasks or interactions with deformable objects? Is the focus on 2D a practical choice, or does it reflect inherent limitations in handling higher-dimensional complexity?
>
> **No specific modifications are necessary** to extend the framework to more complex 3D tasks or interactions with deformable objects. For example, the planning formulation for the `Rope Routing` task, which involves interacting with a deformable rope in 3D space, is identical to that of the 2D tasks. BaB-ND is designed to **solve planning problems involving neural dynamics without relying on task-specific prior knowledge.**
>
> The focus on 2D tasks in many of the evaluations reflects practical considerations rather than limitations. Specifically, (1) these tasks effectively demonstrate the performance of BaB-ND compared to baseline methods, and (2) the 2D tasks are non-trivial, involving challenges such as object piles, non-convex feasible regions, and long action sequences. For instance, the `Pushing with Obstacles` task, to the best of our knowledge, has not been addressed in prior works. Even the original version of this task introduced in [4] only considered scenarios without obstacles and with a fixed target pose.
>
> [4] C. Chi et al., ‘Diffusion Policy: Visuomotor Policy Learning via Action Diffusion’, RSS, 2023.
>
>
> > How might the framework perform in a wider range of challenging scenarios? Additional experiments with more varied conditions would offer insight into its adaptability to complex tasks or novel object interactions.
>
> Thank you for the suggestion. We conducted additional experiments on varied conditions like the configuration of the target or obstacles may change during the execution. Please refer to the anonymous video links: [Demo video for adding a new obstacle](https://streamable.com/rla3ud) and [Demo video for changing target state](https://streamable.com/852ug2). The results show that our BaB-ND can effectively provide new action sequences to reach the target when the environment changes.
>
> We have attempted to address all your questions and made the corresponding changes to the paper. We believe the paper is now much stronger thanks to your suggestions. Please let us know if you have any additional concerns, and we hope you can reevaluate our paper based on our response.

---

> ### Comment · Area_Chair_sADg · 2024-11-25
>
> Dear Reviewer,
>
> Please provide feedback to the authors before the end of the discussion period, and in case of additional concerns, give them a chance to respond.
>
> Timeline: As a reminder, the review timeline is as follows:
>
> November 26: Last day for reviewers to ask questions to authors.
>
> November 27: Last day for authors to respond to reviewers.

---

> ### Author Response · Authors · 2024-11-26
> **Follow-Up on Discussion Period**
>
> Dear Reviewer bbRD,
>
> We have provided detailed responses to all your questions and revised our paper accordingly. We believe the paper is now much stronger, thanks to your constructive feedback and suggestions. As the paper revision deadline is less than a day away, we would appreciate it if you could kindly review our response and let us know if you have any additional questions. Thank you!
>
> Best Regards,
> The Authors

---

> > ### Comment · Reviewer_bbRD · 2024-11-26
> >
> > Thank you for the detailed response and the additional context provided. While I appreciate the effort to address my concerns, some critical points remain insufficiently clarified.
> >
> > Regarding task generality versus customization, while the breakdown of task-specific data collection and training time is helpful, it does not fully address the concern about the framework's adaptability to diverse, real-world applications without significant manual tuning. There is insufficient discussion on how the framework could handle more complex scenarios or larger-scale tasks with minimal reliance on expert intervention.
> >
> > Although feedback control is mentioned as a way to mitigate errors, there is a lack of concrete evidence or analysis on how design mismatches could impact performance or how such dependency could be reduced in practice.
> >
> > Additionally, on scalability to 3D tasks, while it is stated that no specific modifications are necessary, the theoretical discussion does not address the potential challenges in higher-dimensional settings, such as increased search space complexity or computational demands, especially with deformable objects or non-convex feasible regions. It would be helpful to see a more detailed analysis of how the framework manages such challenges and whether strategies like pruning or bound propagation improvements are specifically designed to address the added difficulty in 3D scenarios. For example, what are the expected impacts on computational costs and solution times when moving from 2D to 3D? These clarifications would provide a stronger foundation for understanding the framework's scalability and robustness.

---

> ### Author Response · Authors · 2024-11-29
> **Follow-Up Response to Reviewer bbRD (Part 1)**
>
> Thank you for your feedback. We greatly appreciate your follow-up questions and the opportunity to provide further clarification on our work. Specifically, we discussed the challenges in the field of neural dynamic models and **added new experiments** to show our scalability potential.
>
> > Regarding task generality versus customization, while the breakdown of task-specific data collection and training time is helpful, it does not fully address the concern about the framework's adaptability to diverse, real-world applications without significant manual tuning. There is insufficient discussion on how the framework could handle more complex scenarios or larger-scale tasks with minimal reliance on expert intervention.
>
> We acknowledge that manual tuning or expert intervention can pose challenges for the framework in more complex scenarios or larger-scale tasks. While neural dynamics modeling is a promising technique for robotic manipulation, there remain many open research questions to address, such as **(1) generating large-scale, high-quality data, (2) generalizing model design/learning across diverse scenarios, and (3) effectively leveraging these models for downstream applications.**
>
> Our work **focuses on the last question** by providing a stronger planner that is broadly applicable to various neural dynamics models. While we cannot solve all these challenges in a single paper, we believe our contributions represent meaningful progress in this direction and lay the groundwork for future exploration of generalizable, large-scale solutions.
>
> > Although feedback control is mentioned as a way to mitigate errors, there is a lack of concrete evidence or analysis on how design mismatches could impact performance or how such dependency could be reduced in practice.
>
> Thank you for raising the issue of design mismatches again.  We acknowledge that there are many challenges associated with learning of neural dynamics models, and design mismatches are indeed one of them (e.g., MLPs can struggle with handling multi-object interactions). However,  it is not the primary focus of our contributions since **we study better utilization of pre-trained models instead of learning of better models.**
>
> That said, **we have made a good effort to reduce the mismatch issue in practice.** We design our neural dynamics models for different tasks based on prior works in this area and further narrow the sim-to-real gap by fine-tuning the models with additional data collected from the real world.
>
> > Additionally, on scalability to 3D tasks, while it is stated that no specific modifications are necessary, the theoretical discussion does not address the potential challenges in higher-dimensional settings, such as increased search space complexity or computational demands, especially with deformable objects or non-convex feasible regions. It would be helpful to see a more detailed analysis of how the framework manages such challenges and whether strategies like pruning or bound propagation improvements are specifically designed to address the added difficulty in 3D scenarios. For example, what are the expected impacts on computational costs and solution times when moving from 2D to 3D? These clarifications would provide a stronger foundation for understanding the framework's scalability and robustness.
>
> Thank you for your question regarding the scalability of our BaB-ND in complex manipulation tasks. To provide a stronger foundation for understanding the scalability and robustness of BaB-ND, we conducted **additional experiments**. These include a synthetic example that scales to hundreds of dimensions and further analysis of the search space in the `Rope Routing` task, which involves both 3D action and a deformable object.

---

> > ### Author Response · Authors · 2024-11-29
> > **Follow-Up Response to Reviewer bbRD (Part 2)**
> >
> > ### 1. Scalability Analysis on the Synthetic Example
> >
> > We extend our experiment on the synthetic example shown in Figure 4, as this allows us to easily scale up the input dimension while knowing the optimal objectives. We vary the input dimension $N$ from 50 to 300 and compare our BaB-ND with MPPI and CEM. Although this synthetic example is simpler than practical cases, it provides valuable insights into the **expected computational cost and solution quality as we scale to high-dimensional problems.** It demonstrates the potential of BaB-ND in handling complex scenarios such as 3D tasks. We report the gaps between the best objective found by different baseline methods and the optimal objective value below.
> >
> > The results show that our BaB-ND much outperforms baselines when the input dimension increases. These results are expected since existing sampling-based methods **search for solutions across the entire input space**, requiring an exponentially increasing number of samples to achieve sufficient coverage. In contrast, our BaB-ND **strategically splits and prunes unpromising regions of the input space**, guiding and improving the effectiveness of existing sampling-based methods.
> >
> > |Gap to $f^*$ (↓)|50|100|150|200|250|300|
> > |-|-|-|-|-|-|-|
> > |MPPI|7.4467|45.1795|105.1584|181.1274|259.1044|357.3273|
> > |CEM|5.1569|15.6328|26.3735|39.3862|61.6739|92.4286|
> > |Ours|**0.0727**|**0.2345**|**0.4210**|**0.6976**|**1.2824**|**1.7992**|
> >
> > We further report the following metrics about our BaB-ND to better understand the behavior of BaB-ND under high-dimensional cases:
> > 1. The gap between the best objective found and the optimal objective value as above,
> > 2. The normalized space size of selected subdomains at the last iteration, and
> > 3. The total runtime.
> >
> > The results demonstrate that our BaB-ND effectively **focuses on small regions to search** for better objectives, while the **runtime increases approximately linearly with input dimension** under GPU acceleration.
> >
> > |Input dim $N$|50|100|150|200|250|300|
> > |-|-|-|-|-|-|-|
> > |Gap to $f^*$ (↓)|0.0727|0.2345|0.4210|0.6976|1.2824|1.7992|
> > |Selected Space Size (↓) |0.0002|0.0017|0.0026|0.0042|0.0064|0.0040|
> > |Runtime (↓)|4.2239|6.5880|9.5357|11.6504|13.7430|15.8053|
> >
> >
> >
> > ### 2. Search Space Analysis on `Rope Routing`
> >
> > To evaluate the effectiveness of bounding estimation and pruning in a 3D task involving deformable objects, we conducted additional experiments on the `Rope Routing` task. We report three metrics over the branch-and-bound iterations:
> > 1. The normalized space size of pruned subdomains,
> > 2. The size of selected subdomains for searching, and
> > 3. Improvements in the objective value.
> >
> > The results show that, as iterations progress, **the average total space size of pruned subdomains increases rapidly and then converges,** demonstrating the effectiveness of our bounding methods. Once the pruned space size stabilizes, **the total space size of selected promising subdomains continues to decrease**, indicating that the estimated lower bounds remain effective in identifying promising subdomains. Finally, **the steadily decreasing objective value over iterations** confirms that BaB-ND focuses on promising subdomains.
> >
> > | Iterations       | 0      | 4      | 8      | 12     | 16     | 20     |
> > | ----------------- | ------ | ------ | ------ | ------ | ------ | ------ |
> > | Pruned Space Size (↑) | 0.0000 | 0.2875 | 0.7453 | 0.7928 | 0.8032 | 0.8060 |
> > | Selected Space Size (↓) | 1.0000 | 0.7125 | 0.1086 | 0.0142 | 0.0059 | 0.0035 |
> > | Objective (↓)    | 67.1343 | 54.4342 | 48.5176 | 47.2982 | 47.0786 | 46.9568 |
> >
> > This experiment shows that our **bounding and pruning strategies work well on 3D tasks and can guide the search effectively**. In complex 3D tasks with deformable objects, our approach can achieve better scalability compared to existing sampling-based methods which search for solutions across the entire input space.
> >
> > We hope that we have addressed your follow-up questions and we greatly appreciate your feedback. Please let us know if you have any additional concerns.

---

### Official Review · Reviewer_ci99 · 2024-11-04

**Soundness:** 3
**Presentation:** 4
**Contribution:** 3
**Rating:** 8
**Confidence:** 3

**Summary:**

This paper presents BaB-ND, a GPU-accelerated Branch-and-Bound framework designed for long-horizon motion planning in robotic manipulation tasks that require trajectory optimization over neural dynamics models. Addressing challenges associated with non-linearity in neural network dynamics, BaB-ND utilizes a branching heuristic to divide the action space into sub-domains and a modified bound propagation method (inspired by neural verification methods like α, β-CROWN) to efficiently prune non-promising regions. This systematic approach allows BaB-ND to outperform some existing methods in tasks like object pushing, sorting, and rope manipulation, where contact-rich dynamics and high-dimensional action spaces are involved. The framework supports diverse neural architectures (e.g., MLPs, GNNs) and demonstrates scalability and superior planning quality in both simulated and real-world settings.

Overall, this work presents a novel idea, with a clear presentation, and I could increase my rating if my questions are covered during the rebuttal phase.

**Strengths:**

**Originality**:
This paper is quite original in its approach to motion planning by adapting the branch-and-bound (BaB) algorithm for neural dynamics (ND) models. Unlike existing methods, which typically rely on sampling-based or gradient descent approaches, this work creatively applies a modified bound propagation technique inspired by neural network verification algorithms, specifically the α,β-CROWN method. This adaptation is novel for addressing the inherent non-linearity and complexity of neural dynamics in long-horizon planning tasks. It might set a new direction in combining machine learning verification methods with motion planning.

**Quality**:
The research demonstrates rigor and of high-quality.
- Good amount of testing across diverse scenarios (planar pushing, object sorting, rope routing)
- Performance advantages over existing methods like Mixed-Integer Programming and sampling-based approaches
- Robust handling of complex, contact-rich environments
- The video covers the experiments and demonstrates the improvement over alternative methods

**Clarity**
The presentation of the paper is good.
- Well-organized structure and methodology explanation
- Effective use of visual aids and diagrams
- The step-by-step example of applying BaB on a 1D problem is particularly helpful
- (mostly) clear differentiation between BaB-ND and traditional verification methods

**Significance**
The work is important in several fronts.
- It addresses a critical challenge in robotic motion planning: managing complex, long-horizon tasks with contact dynamics
- The provided solution is -somewhat- scalable, when compared to an existing method
- Different neural architectures (MLPs to graph neural networks) are used, which demonstrates its applicability on different architectures, and hence different types of problems.

**Weaknesses:**

There are several weaknesses/limitations of the work:

**Limited Task Diversity**:
- Current evaluation focuses primarily on basic manipulation tasks (pick-and-place and pushing, relying on planar motions)
- Framework's applicability remains uncertain for:
  - Complex 3D manipulation with critical contact points and impact forces
  - Highly constrained environments requiring precise 3D motions of the full robot kinematic chain
  - e.g., Real-world scenarios like book reordering across shelves

**scalability**: (somewhat similar to the comment above)
- Despite GPU acceleration, potential bottlenecks exist:
  - Computationally intensive bound propagation
  - Extensive branching requirements
  - Challenges for real-time applications
- Paper would benefit from:
  - Discussion of potential optimizations for real-time use
  - Experiments measuring latency in time-sensitive scenarios
  - Strategies for reducing computational overhead

**Lack of Comparative Analysis**:  I acknowledge that it'd be out of scope, but still some remarks:
- Comparisons with traditional motion planners that use full state information might be interesting, especially considering the somewhat long runtime of BaB-ND + the training time for each task. Overall, that could provide a better context regarding trade-offs between:
  - BaB-ND's computational costs (including training time)
  - Performance of conventional motion planning approaches
  - Overall efficiency in practical applications

**Insufficient Hyperparameter Analysis**: The BaB-ND framework includes various hyperparameters, such as the choice of branching heuristic, bound propagation depth, and sampling rates. However, there is limited discussion or analysis on how these hyperparameters influence performance across different tasks or neural architectures. Since tuning these parameters is likely critical for achieving optimal results in different scenarios, an expanded study on hyperparameter sensitivity would be beneficial. For instance, sensitivity experiments could provide insights into the trade-offs between solution quality and computational efficiency, guiding practical implementation.

**Questions:**

- In the supplementary video, it is stated that planning time / horizon is the same for all methods. How would the performance of the baseline methods change if we increase the planning time? In the experiments, it is shown that BaB-ND outperforms other methods in almost all tasks. May it be the case that other methods have not yet found their optimal solutions?

- Can you prune enough sub-domains so that the algorithm does not end up sampling exhaustively? In which cases (or for which type of objective functions) can you provide an assurance that enough sub-domains will be pruned? Can you give a theoretical insight into it? More generally, since the proposed method relies on sampling, in which cases it performs poorly compared to the non-sampling methods?

- Why isn't there any sampling-based method for scalability comparison? It is expected for a sampling-based method to be more scalable compared to MIP (?).

- How is the close-loop control achieved and feedback received?

- data collection: how are the variations (assuming there are) of the task achieved?

- Fig.6: consistency on the plots is preferred. For rope routing, cost performance is reported for the open-loop scenario, whereas success rate is reported for the closed-loop. Why is it so?

- Could the authors discuss potential modifications for real-time application?

- Could the authors expand on hyperparameter sensitivity?

- Are there potential optimizations for early-stopping propagation in bound calculations? The current early-stopping approach helps avoid excessive error from deep-layer propagation, but it may not achieve the tightest bounds possible. Any chance on adative stopping based on some criteria/threshold?

---

> ### Author Response · Authors · 2024-11-24
> **Author Response to Reviewer ci99 (Part 1)**
>
> Thank you for acknowledging our novel contributions in adapting branch-and-bound (BaB) for neural dynamics (ND) models and its potential to set a new direction in combining ML verification methods with motion planning. We greatly appreciate your thoughtful comments and constructive suggestions, and we have conducted **additional experiments** as you suggested:
> 1. Performance comparison with conventional motion planning approaches;
> 2. Hyperparameter analysis for our BaB-ND;
> 3. Search space analysis for our BaB-ND;
> 4. Runtime analysis for our BaB-ND and sampling-based methods;
> 5. Planning performance comparison under a longer time limit.
>
> > Task Diversity
>
> We would like to emphasize that we have already included a diverse set of tasks, encompassing rigid bodies and deformable objects, single objects and object piles, 2D and 3D actions, non-convex feasible regions, and long action sequences. Additionally, we have demonstrated the framework's applicability to both MLP- and GNN-based neural dynamics models.
>
> The tasks included in this paper are significantly more challenging than those in prior work using MIP for planning [1]:
>    - The `Object Pushing` task in [1] only considered 1-step planning. Our `Pushing with Obstacles` task introduces additional obstacles with diverse configurations, making it considerably more challenging and requiring a much longer planning horizon.
>    - The `Rope Manipulation` task in [1] is limited to 2D planar actions and short-horizon planning. In comparison, we extend this task to a 3D scenario where the rope must be routed into a tight-fitting slot, necessitating long-horizon planning.
>
> We agree that incorporating more complex 3D tasks could further enhance the significance of this work. As part of future extensions, we plan to apply our framework to tasks from [2, 3], such as dexterous and cloth manipulation. We are confident in the potential results, as these tasks typically rely on CEM/MPPI for planning, and we have demonstrated that our method consistently outperforms CEM/MPPI in both efficiency and performance.
>
> [1] Z. Liu et al., ‘Model-Based Control with Sparse Neural Dynamics’, NeurIPS, 2023.
> [2] A. Nagabandi et al., ‘Deep dynamics models for learning dexterous manipulation’, CoRL, 2020.
> [3] X. Lin et al., ‘Learning Visible Connectivity Dynamics for Cloth Smoothing’, CoRL, 2021.
>
> > Scalability (potential bottlenecks, real-time use) and potential modifications for real-time application
>
> **Potential bottlenecks**: We have provided a detailed analysis of the runtime for each component (branching, bounding, and searching) across planning problems of varying sizes (Figure 7(b) in the paper). The results indicate that **branching and bounding add only marginal overhead**, particularly for larger problems, as both branching and bounding can be parallelized and benefit significantly from GPU acceleration.
>
> The primary bottleneck remains the searching process, which is similar to the sampling-based planning methods such as CEM and MPPI. In practice, if a sufficiently good solution can be identified, the runtime of the search process can be reduced by decreasing the number of samples or iterations during sampling.
>
> **Time-sensitive scenarios**: For real-world deployment on robots, planning only needs to be performed once at the start, typically taking 1 minute. During execution, a real-time low-level planner is employed to track the original plan. Consequently, our proposed approach does not pose a bottleneck for real-time manipulation.
>
> If we want to further improve the efficiency of the algorithm, we could explore warm-starting future BaB-ND launches by reusing the branch-and-bound tree from previous solutions wherever possible. Additional strategies include minimizing redundant initialization overhead across multiple launches or splitting multiple dimensions simultaneously during branching.

---

> > ### Author Response · Authors · 2024-11-24
> > **Author Response to Reviewer ci99 (Part 2)**
> >
> > > Comparison with conventional motion planning approaches and overall efficiency
> >
> > We conducted additional experiments on task `Pushing with Obstacle` to compare the planning performance of our BaB-ND method against two conventional motion planning approaches:
> > 1. Rapidly-exploring Random Tree (RRT)
> > 2. Probabilistic Roadmap (PRM)
> >
> > We also include the performance of the original baselines (i.e., GD, MPPI, and CEM) as a reference. The table below presents the final step cost at a planning horizon of $H$:
> > ||GD|MPPI|CEM|RRT|PRM|Ours|
> > |-|-|-|-|-|-|-|
> > |Final step cost (↓)|4.1238|1.5082|1.0427|10.6472|13.2930|**0.2339**|
> >
> > The results demonstrate that our method significantly outperforms all other approaches. Implementation details for RRT and PRM have been included in Appendix D.5. The main reasons for the performance gap are as follows:
> > 1. The search space in our task is complex and continuous, making it challenging for discrete sampling methods like RRT and PRM to achieve effective coverage.
> > 2. These methods are prone to getting stuck on obstacles, often failing to reach the target state.
> >
> > **Overall efficiency**: Below, we provide a breakdown of the time required for data collection, model training, and long-horizon planning across different tasks:
> >
> >    || Data Collection (min) | Model Training (min) | Long-Horizon Planning (min) |
> >    |-|-|-|-|
> >    | Pushing with Obstacles | 1 | 5 | 1 |
> >    | Object Merging | 8.5 | 20 | 1 |
> >    | Object Sorting | 4 | 20 | 1.3 |
> >    | Rope Routing | 10 hours | 20 | 0.5 |
> >
> > Our framework is reasonably efficient: for most tasks shown in the paper, data collection (performed in simulation) takes less than 10 minutes, and model training requires approximately 20 minutes. The exception is the `Rope Routing` task, where data collection takes longer due to simulator inefficiencies. However, this can be addressed using faster simulators in the future.
> >
> > It is also important to note that data collection is performed offline and only once. As a result, it introduces minimal overhead for the practical deployment of our framework.
> >
> > Finally, we emphasize that our novel technical contribution focuses on improving *planning* efficiency of pre-trained neural dynamics; the study on efficient data collection and model training are important future work for the robot learning community.
> >
> > > Hyperparameter Analysis
> >
> > We evaluated the hyperparameter sensitivity of BaB-ND for the `Pushing with Obstacles` and `Object Sorting` tasks by adjusting three key hyperparameters:
> > 1. $\eta = \frac{n_1}{n} \in [0,1]$, the ratio of the number of subdomains picked with the best upper bounds ($n_1$) to the number of all picked subdomains ($n$) in the heuristic for selecting subdomains to split. A larger $\eta$ promotes exploitation, while a smaller $\eta$ encourages exploration.
> > 2. $T \in \mathbb{R}$, the temperature of Softmax sampling in the heuristic for subdomain selection. A larger $T$ results in more uniform and random sampling, whereas a smaller $T$ leads to more deterministic selection of subdomains with the smaller lower bounds.
> > 3. $w \in (0,100]$, the percentage of top samples used in the heuristic for splitting subdomains. A larger $w$ results in more conservative decisions by considering more samples, while a smaller $w$ leads to more aggressive splitting.
> >
> > We report the mean objectives under different hyperparameter configurations. The base hyperparameter configuration is $\eta=0.75$, $T=0.05$, and $w=1$. For benchmarking, we vary at most one hyperparameter at a time while keeping the others fixed at the base configuration.
> >
> > ||$\eta=0.25$|$\eta=0.50$|$\eta=0.75$|$T=0.05$|$T=1$|$T=20$|$w=0.1$|$w=1$|$w=10$|
> > |-|-|-|-|-|-|-|-|-|-|
> > |Pushing with Obstacles|31.8574|31.9828|31.9839|31.9839|32.3990|32.1267|32.0068|31.9839|32.0599|
> > |Object Sorting|30.1760|30.2795|31.0482|31.0482|31.2366|31.8263|30.5953|31.0482|31.1545|
> >
> > The results demonstrate that varying hyperparameter configurations produces only slight differences in objectives, highlighting that **BaB-ND is robust to hyperparameter variations**. In practice, we can use a consistent set of hyperparameters across all experiments with only minor task-specific adjustments.

---

> > > ### Author Response · Authors · 2024-11-24
> > > **Author Response to Reviewer ci99 (Part 3)**
> > >
> > > > How would the performance of the baseline methods change if we increase the planning time? … May it be the case that other methods have not yet found their optimal solutions?
> > >
> > > We conduct the following simulation experiment to evaluate the open-loop planning performance of all methods under a longer time limit. **While all methods continue to improve the objective within the time limit, our BaB-ND consistently outperforms the baselines as planning time increases.**
> > >
> > > || 1 min | 2 mins | 4 mins |
> > > |-|-|-|-|
> > > |GD | 57.7254 ± 3.6434 | 55.3134 ± 3.8945 | 54.0194 ± 2.9153 |
> > > |MPPI | 41.1475 ± 2.7539 | 38.8000 ± 2.9069 | 36.2136 ± 2.7878 |
> > > |CEM | 40.1702 ± 3.0303 | 38.9708 ± 3.6423 | 37.8750 ± 3.5841 |
> > > |Ours | **34.4681 ± 2.9953** | **33.8162 ± 2.7190** | **32.3099 ± 2.0043** |
> > >
> > >
> > > > Can you prune enough subdomains so that the algorithm does not end up sampling exhaustively? In which cases (or for which type of objective functions) can you provide an assurance that enough subdomains will be pruned? Can you give a theoretical insight into it? More generally, since the proposed method relies on sampling, in which cases it performs poorly compared to the non-sampling methods?
> > >
> > > **Theoretical insight**: CROWN is a linear bound propagation algorithm that approximates the objective function with respect to the input as a linear relationship within subdomains. The tightness of these bounds depends on how linear the objective function is. As branching progresses, the objective function in each subdomain asymptotically approaches linearity, resulting in highly accurate bound calculations. Consequently, unpromising subdomains can be effectively pruned.
> > >
> > > **Empirical results**: Our experiments also demonstrate that the bounding procedure effectively prunes unpromising subdomains. Below, we report three metrics over the brand-and-bound iterations: (1) the normalized space size of pruned subdomains, (2) the size of the selected subdomains, and (3) the improvement in the objective value.
> > >
> > > As iterations progress, **the average total space size of pruned subdomains increases rapidly and then converges**, indicating the effectiveness of our bounding methods. Once the pruned space size reaches a plateau, **the total space size of selected promising subdomains continues to decrease,** showing that the estimated lower bounds remain effective in identifying promising subdomains. The **decreasing objective over iterations** further confirms that BaB-ND focuses on the most promising subdomains, reducing space size to the magnitude of $1 \times 10^{-4}$.
> > >
> > >    | Iterations | 0 | 4 | 8 | 12 | 16 | 20 |
> > >    |-|-|-|-|-|-|-|
> > >    | Pruned space size (↑) | 0.0000 | 0.7000 | 0.8623 | 0.8725 | 0.8744 | 0.8749 |
> > >    | Selected space size (↓) | 1.0000 | 0.3000 | 0.0412 | 4.8339e-03 | 5.2795e-04 | 2.8972e-04 |
> > >    | Objective (↓) | 41.1222 | 36.0511 | 35.5091 | 34.8024 | 33.8991 | 33.3265 |
> > >
> > > **Comparison with non-sampling methods**: Sampling-based methods, including ours, are scalable but do not guarantee finding the optimal solution. In contrast, non-sampling methods like MIP can provide such guarantees but require exponentially greater computation time. MIP works well only for very small models in practice (MIP can solve none of our real-world planning experiments).
> > >
> > > > Why isn't there any sampling-based method for scalability comparison? It is expected for a sampling-based method to be more scalable compared to MIP (?).
> > >
> > > The “scalability comparison” in our initial submission solely serves as a direct comparison to [1], the prior work on planning neural dynamics models with MIP. This comparison was conducted with **much smaller networks, much shorter time horizons, and easier environments** (no obstacles), because MIP is **unable to scale** to the more challenging settings. These are not the settings of our main experiments. Under these much simpler settings, we have conducted comparisons to sampling-based methods as suggested by the reviewer and reported the results in our revised Appendix C.1. We found that in much less challenging settings, all these methods (sampling-based methods, MIP and ours) can perform similarly, though MIP exhibits significantly worse scalability. But we again want to emphasize that this is a **synthetic setting purely serving as a comparison to [1]**, and not the main setting studied in our paper.
> > >
> > > [1] Z. Liu et al., ‘Model-Based Control with Sparse Neural Dynamics’, NeurIPS, 2023.

---

> > > > ### Author Response · Authors · 2024-11-24
> > > > **Author Response to Reviewer ci99 (Part 4)**
> > > >
> > > > As suggested, we conducted an **additional experiment to compare the scalability of our BaB-ND with sampling-based methods**. Here we focus on a more challenging and realistic case (`Pushing with Obstacles` task in our main experiment), so MIP cannot be included. Parameters for all methods were adjusted to ensure similar runtimes for the largest problems.
> > > >
> > > > Below are the average runtimes of all methods across six diagonal elements in the heatmap (Figure A 10), ranging from the smallest model size and shortest planning horizon to the largest model size and longest planning horizon:
> > > >
> > > >    |Runtime (↓)|(0.232K,1)|(0.712K,3)|(2.440K,5)|(34.31K,10)|(134.2K,15)|(530.2K,20)|
> > > >    |-|-|-|-|-|-|-|
> > > >    | GD | 0.6930 | 1.4128 | 2.8380 | 15.5129 | 39.6837 | 117.8663 |
> > > >    | MPPI | 0.7800 | 1.6758 | 3.4854 | 16.7010 | 34.6054 | 116.5935 |
> > > >    | CEM | 1.0159 | 1.8450 | 3.5274 | 16.5886 | 34.6403 | 115.8820 |
> > > >    | Ours | 3.5938 | 6.3131 | 9.3808 | 25.0006 | 45.6266 | 114.6788 |
> > > >
> > > > We also report the average objectives for all methods on the largest four planning problems to evaluate their effectiveness:
> > > >
> > > > |Objective (↓)|(134.2K,15)|(134.2K,20)|(530.2K,15)|(530.2K,20)|
> > > > |-|-|-|-|-|
> > > > | GD | 57.2768 | 64.4789 | 54.7078 | 60.2575 |
> > > > | MPPI | 47.4451 | 53.7356 | 45.1371 | 45.6338 |
> > > > | CEM | 47.0403 | 47.6487 | 43.8235 | 38.8712 |
> > > > | Ours | **46.0296** | **46.1938** | **41.6218** | **34.6972** |
> > > >
> > > > These results show that our **BaB-ND achieves similar scalability but noticeably better objective compared with sampling-based baseline methods.**
> > > >
> > > > > How is the close-loop control achieved and feedback received?
> > > >
> > > > At the start of the task, we perform state estimation based on camera observations. Next, we use BaB-ND for long-horizon planning to generate a reference state-action trajectory (the planning takes roughly 1 minute). Closed-loop control is then employed to follow this reference trajectory. At each step of the closed-loop control, camera observations provide object state feedback. A local planner (e.g., MPPI) is used to track the reference trajectory. Details of our real-world experimental setup are provided in Appendix D.4.
> > > >
> > > > > Data collection: how are the variations (assuming there are) of the task achieved?
> > > >
> > > > During data generation, we begin by randomly setting the object's initial state and the end effector's position. Next, we randomly sample actions to encourage interactions between the end effector and the object, ensuring sufficient coverage of scene variations that the neural dynamics model might encounter during testing.
> > > >
> > > >
> > > > > Figure 6: consistency on the plots is preferred. For rope routing, cost performance is reported for the open-loop scenario, whereas success rate is reported for the closed-loop. Why is it so?
> > > >
> > > > We find it is insufficient to use only the final step cost to evaluate the success of a trajectory since a greedy trajectory that horizontally routes the rope may achieve a low final cost but fails to route it into the slot. So we report the success rate directly. We have clearly clarified this inconsistency in our revised paper.
> > > >
> > > > > Are there potential optimizations for early-stopping propagation in bound calculations? The current early-stopping approach helps avoid excessive error from deep-layer propagation, but it may not achieve the tightest bounds possible. Any chance on adative stopping based on some criteria/threshold?
> > > >
> > > > We have implemented an adaptive early-stopping approach in our current implementation. Specifically, we initialize with a small propagation depth so that the bound estimation on large subdomains remains reasonable during the initial iterations. As branching progresses and the subdomains become smaller, we adaptively increase the propagation depth to obtain tight bound estimation. More adaptation techniques to automatically tuning the propagation depth in more general cases can be studied.
> > > >
> > > > We have attempted to address all your questions and made the corresponding changes to the paper. We believe the paper is now much stronger thanks to your suggestions, and we hope you can reevaluate our paper based on our response. Please let us know if you have any additional concerns.

---

> > > > > ### Comment · Reviewer_ci99 · 2024-11-26
> > > > > **Improved paper, updated score**
> > > > >
> > > > > Thank you for the clear and detailed responses to all my questions and concerns. I think the additional experiments and explanations strengthen the paper significantly. Most of my concerns, particularly regarding task diversity, scalability, hyperparameter analysis, and comparisons with baselines, are alleviated. I upgraded my score accordingly.

---

> > > > > > ### Author Response · Authors · 2024-11-26
> > > > > > **Thank You for Reevaluating and Updating the Score**
> > > > > >
> > > > > > Dear Reviewer ci99,
> > > > > >
> > > > > > Thank you for your thoughtful review of our submission and rebuttal. We greatly appreciate your constructive feedback, which has greatly enhanced our work. We are glad the revisions have addressed your concerns.
> > > > > >
> > > > > > Best Regards,
> > > > > > The Authors

---

> ### Comment · Area_Chair_sADg · 2024-11-25
>
> Dear Reviewer,
>
> Please provide feedback to the authors before the end of the discussion period, and in case of additional concerns, give them a chance to respond.
>
> Timeline: As a reminder, the review timeline is as follows:
>
> November 26: Last day for reviewers to ask questions to authors.
>
> November 27: Last day for authors to respond to reviewers.

---

> ### Author Response · Authors · 2024-11-26
> **Follow-Up on Discussion Period**
>
> Dear Reviewer ci99,
>
> We have provided detailed responses to all your questions and revised our paper accordingly. We believe the paper is now much stronger, thanks to your constructive feedback and suggestions. As the paper revision deadline is less than a day away, we would appreciate it if you could kindly review our response and let us know if you have any additional questions. Thank you!
>
> Best Regards,
> The Authors

---

### Author Response · Authors · 2024-11-24
**General Response**

We thank all reviewers for their thoughtful and constructive feedback. In response, we provide detailed individual replies to address the concerns raised by each reviewer and outline the revisions made to our manuscript.

We are pleased that the reviewers acknowledged:
1. The **novelty** of our BaB-ND, which integrates ML verification methods with motion planning involving neural dynamics models. [ci99, kMxQ]
2. The **effectiveness, applicability, and scalability** of BaB-ND in addressing challenging planning problems involving neural dynamics models with diverse architectures. [ci99, bbRD, kMxQ]
3. The clear **presentation** of high-level ideas and formulation. [ci99, U2wt]

We have made the following clarifications to address the reviewers' concerns:
1. Clarified the **diversity and complexity of our benchmarking tasks** compared to prior work and highlighted the effectiveness of BaB-ND in more challenging (realistic, time-sensitive, dynamic) tasks. [ci99, bbRD, U2wt]
2. Emphasized that BaB-ND **focuses on solving complex NN-involved planning problems** in robotic manipulation tasks, assuming a pre-trained neural dynamics model, rather than focusing on **training neural dynamics models** and **data collection**. [ci99, bbRD, U2wt]
3. Clarified the **specific distinctions** between BaB-ND and existing neural network verification techniques [U2wt], as well as our **novelty from a machine learning perspective**. [kMxQ]
4. Explained key concepts and details of BaB-ND using **more illustrative examples** [kMxQ] and provided additional details about our **real-world experiments**. [U2wt]
5. Justified the **selection of different baselines** for comparisons of effectiveness and scalability. [ci99, kMxQ]
6. Clarified the inconsistency in performance metrics for the Rope Routing task. [ci99, U2wt]

Additionally, we have conducted further experiments as suggested by the reviewers:
1. Compared our BaB-ND with two additional conventional motion planning approaches (i.e., RRT and PRM) and demonstrated the superiority of our BaB-ND. [ci99, bbRD]
2. Performed additional ablation studies and hyperparameter analyses to further evaluate the effectiveness of BaB-ND. The results show that our branching heuristics and bounding approaches **effectively and efficiently guide the search**, and BaB-ND is **robust to its hyperparameters**. [kMxQ, ci99]
3. Evaluated BaB-ND under varied and dynamic scenarios. The results indicate that BaB-ND effectively replans and provides **new long-horizon action sequences** to reach the target when the **environment (e.g., obstacles or the target) changes**. [bbRD, U2wt]
4. Compared BaB-ND with sampling-based methods in scalability analysis. The results show that BaB-ND achieves **comparable scalability in runtime** while delivering **superior planning performance** on complex planning problems. [ci99, kMxQ]
5. Conducted a quantitative analysis of the search space to evaluate the critical role of the bounding component in BaB-ND. The results highlight that the bounding component efficiently **provides tight bound estimations**, prunes unpromising subdomains, and identifies promising subdomains to **guide the search**. [ci99, kMxQ]
6. Compared GPU-accelerated BaB-ND with its CPU version. The results show that BaB-ND **benefits significantly from GPU acceleration** and achieves good scalability. [kMxQ]

We have also made revisions to our main paper and reorganized our appendix, with changes highlighted in blue for better clarity:
1. Included the **high-level ideas of the bounding component** to enhance readability and emphasized that understanding our main ideas does not require detailed knowledge of bounding. [kMxQ]
2. Added a paragraph summarizing the **distinctions of our BaB-ND** compared to existing neural network verification techniques. [U2wt]
3. Clarified the inconsistency in performance metrics for the Rope Routing task between Figure 6 and its description. [ci99, U2wt]
4. Provided a detailed illustration of **a simplified robotic manipulation task** to better introduce the key concepts of BaB-ND in Appendix A.1. [kMxQ]
5. Included the discussion about **limitations and potential future directions** of our work in Appendix A.4. [U2wt]
6. Added a **detailed illustration of our bounding component using an MLP** as an example in Appendix B.2. [kMxQ]
7. Offered **additional experimental results** in Appendix C. [ci99, bbRD, kMxQ, U2wt]
8. Included more details about our **real-world experiments** in Appendix D.5. [U2wt]
9. Included implementation details about RRT and PRM in Appendix D.6. [ci99, bbRD]
10. Fixed typos.

Please refer to our individual responses and the revised paper for more details. We sincerely thank the reviewers for their time and efforts, and we hope our responses have effectively addressed all concerns. We hope you can kindly reevaluate our paper based on our new results provided, and we look forward to any additional comments or feedback.

Best regards,
The Authors

---

### Meta-Review · Area_Chair_sADg · 2024-12-23

**Metareview:**

This paper introduces BaB-ND, a GPU-accelerated Branch-and-Bound framework for long-horizon motion planning in robotic tasks, leveraging neural dynamics and bound propagation techniques from neural network verification. The method partitions the action space and prunes non-promising regions, achieving scalability and efficiency in solving complex planning problems. The experimental results across several manipulation tasks demonstrate significant improvements over baselines, with strong scalability and adaptability to different neural architectures.

While the work is novel and rigorously validated, its primary limitations include limited exploration of highly unstructured tasks and the current need for task-specific neural networks. However, the authors’ rebuttal addressed many concerns, including additional experiments and clarifications, which clarified and strengthened the paper’s contributions.

Given its novelty, rigorous empirical evaluation, and potential for future research, I recommend acceptance. Remaining limitations are minor compared to the paper’s contribution in the domain of robot motion planning.

**Additional Comments On Reviewer Discussion:**

The discussion and rebuttal phase highlighted several critical aspects of the paper. Reviewer ci99 noted the paper’s originality in adapting BaB methods for neural dynamics and emphasized the importance of additional experiments and hyperparameter analysis. The authors addressed these points by conducting new experiments, providing insights into task diversity, runtime comparisons, and scalability. This led to an upgraded score from this reviewer.

Reviewer bbRD raised concerns about task generalization, scalability to 3D scenarios, and the framework’s reliance on task-specific networks. The rebuttal clarified that the current evaluations primarily focus on structured tasks, and while the method is theoretically applicable to 3D and deformable-object tasks, the empirical validations remain limited. The authors conducted additional experiments to demonstrate scalability and discussed potential extensions for more complex tasks, partially alleviating these concerns.

Reviewer kMxQ critiqued the accessibility of the method and the limited ablation studies. The authors responded by adding illustrative examples in the appendix and conducting ablations to highlight the contributions of individual components. While the reviewer acknowledged these efforts, they maintained that the explanations for core concepts like bound propagation and GPU acceleration could be more detailed.

Overall, the authors’ thorough and well-structured rebuttal effectively addressed many of the reviewers’ concerns, strengthening the case for acceptance. The remaining limitations, particularly regarding task generalization and method accessibility, are valid but do not outweigh the paper’s contributions to the field, therefore the AC recommends accepting the paper.

---

### Decision · Program_Chairs · 2025-01-22

Accept (Poster)